# Schur Connections:
## Chord Counting, Line Operators, and Indices

Oscar Lewis [1], Mark Mezei [1], Matteo Sacchi[2], Sakura Schäfer-Nameki [1]

[1] *Mathematical Institute, University of Oxford,*
*Andrew Wiles Building, Woodstock Road, Oxford, OX2 6GG, UK*
[2] *Simons Center for Geometry and Physics,*
*Stony Brook University, Stony Brook, NY 11794-3636, USA*

Recently, an intriguing correspondence was conjectured in [1] between Schur half-indices of pure 4d $SU(2)$ $\mathcal{N} = 2$ supersymmetric Yang-Mills (SYM) theory with line operator insertions and partition functions of the double scaling limit of the Sachdev-Ye-Kitaev model (DSSYK). Motivated by this, we explore a generalization to $SU(N)$ $\mathcal{N} = 2$ SYM theories. We begin by deriving the algebra of line operators, $\mathcal{A}_{\text{Schur}}$, representing it both in terms of the $\mathfrak{q}$-Weyl algebra and $\mathfrak{q}$-deformed harmonic oscillators, respectively. In the latter framework, the half-index admits a natural description as an expectation value in the Fock space of the oscillators. This $\mathfrak{q}$-oscillator perspective further suggests an interpretation in terms of generalized colored chord counting, and maps the half-index to a purely combinatorial quantity. Finally, we establish a connection with the quantum Toda chain, which is an integrable model whose commuting Hamiltonians can be identified with the Wilson lines of the $SU(N)$ SYM, and their eigenfunctions correspond to the function basis appearing in the half-index.

# 1  Introduction and Summary

Surprising connections between supersymmetric gauge theories and various quantum systems such as integrable, conformal, topological field theories have led to deep mathematical and physical insights. Classic examples are the AGT correspondence [2], the 3d-3d correspondence [3], or the 4d-2d correspondence [4], for reviews see [5–7]. In each of these, the starting point is the identification of certain physical observables, such as a supersymmetric partition function with a correlation function in the dual quantum system. These correspondences usually are general and hint to some fundamental connections at the core. For the AGT correspondence, the initial formulation related the $S^4$ partition function of $\mathcal{N} = 2$ $SU(2)$ gauge theories and 2d Liouville CFT correlators. This was subsequently shown to generalize to higher rank $SU(N)$ class $\mathcal{S}$ theories [8] and Toda CFTs [9].

Recently, Gaiotto and Verlinde proposed a similarly intriguing connection between the 4d $\mathcal{N} = 2$ pure $SU(2)$ Yang-Mills theory and the Sachdev-Ye-Kitaev (SYK) model in one (time) dimension [1]. More precisely, they provide evidence for the identification of the following two quantities:

- The Schur half-index of 4d $\mathcal{N} = 2$ pure $SU(2)$ SYM with Wilson and dyonic line operator insertions. This is a function of $\mathfrak{q}$, which is keeping track of a combination of R-symmetry and Lorentz spins;

- The expectation value of the Hamiltonian and local operators in the double scaling limit of the SYK model (DSSYK) [10–13], where $\mathfrak{q}$ plays the role of the double scaling parameter.

The main goal of this paper is to consider the natural extension of this to $SU(N)$ pure $\mathcal{N} = 2$ SYM, and to explore the – less obvious – extension of various combinatorial and quantum mechanical systems that can be associated to the Schur half-index in this context.

**Schur Quantization, Half Index and Algebra of Line Operators.**  The starting point here is a 4d $\mathcal{N} = 2$ supersymmetric gauge theory with half-BPS line defects. Although this generalizes to any theory with a Seiberg-Witten description, we will here restrict to pure gauge theories with gauge groups $SU(N)$. The algebra of line operators in such a theory is

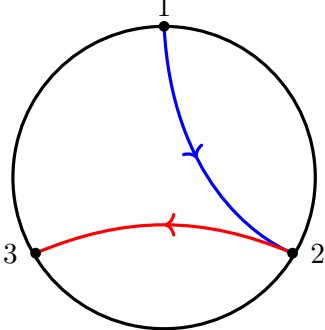
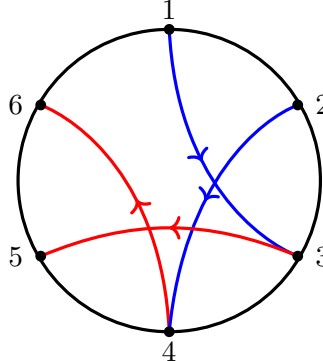

Figure 1: Two examples of chord diagrams arising in the context of evaluating the Schur half-index 4d $\mathcal{N} = 2$ $SU(3)$ SYM, where the elementary chord configuration is the one shown on the left (one blue chord, one red chord with a single intersection). Its gauge theoretic interpretation is the insertion of three fundamental Wilson lines. A diagram contributing to the insertion of six Wilson lines is shown on the right.

best studied in the context of the holomorphic-topological twist [14]. The Schur index [15, 16] can be used to formulate a quantization of the Coulomb branch of the $\mathcal{N} = 2$ gauge theory [5, 17]. Specifically, the set of half-BPS line defects inserted in the Schur index or the Schur half-index [18] forms a non-commutative $\star$-algebra $\mathcal{A}_{\text{Schur}}$ that depends on a deformation parameter $\mathfrak{q}$. As discussed in [17], for $0 < \mathfrak{q} < 1$ one can associate a Hilbert space $\mathcal{H}_{\text{Schur}}$ via the Gelfand-Naimark-Segal (GNS) construction, thanks to the fact that the Schur half-index is positive definite and thus induces an inner product. This is what is usually referred to as *Schur quantization*. The Schur algebra also connects to a multitude of interesting problems, such as quantization of complex Chern-Simons theory and representations of $U_{\mathfrak{q}}(SL(N, \mathbb{C}))$. Moreover, in the limit $\mathfrak{q} \to 1$ the algebra $\mathcal{A}_{\text{Schur}}$ becomes commutative and reduces to the algebra of holomorphic functions over the Coulomb branch of the 3d $\mathcal{N} = 4$ obtained by $S^1$ compactification. Hence, $\mathcal{A}_{\text{Schur}}$ can also be understood as a quantization (in the mathematical sense) of this algebra.

**DSSYK, Chord Counting and Liouville.** Let us now turn to the DSSYK part of the correspondence. The DSSYK quantities have several alternative descriptions: they can be computed combinatorially from counting certain chord diagrams [19–21] which is equivalent to a transfer matrix acting on an auxiliary chord Hilbert space, or as $\mathfrak{q}$-deformed Liouville quantum mechanics [22], or as a timelike Liouville CFT on the Möbius strip with (non-conformal) Dirichlet boundary conditions [23]. In the $\mathfrak{q} \to 1$ limit (and simultaneously going to low energies) these descriptions simplify to the well-studied Schwarzian theory holographically dual to Jackiw-Teitelboim (JT) gravity [24, 25]: e.g. the Liouville quantum mechanics describes

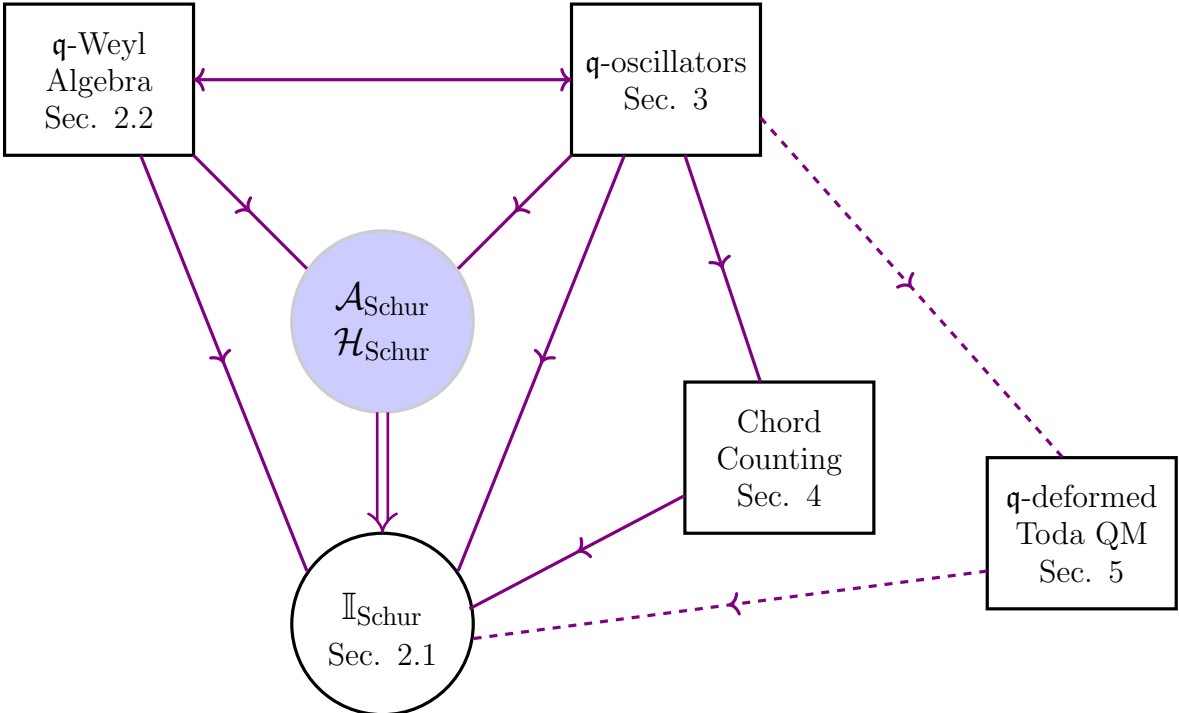

Figure 2: Schur Connections: The central object is the algebra $\mathcal{A}_{\text{Schur}}$ and its representation on a Hilbert space $\mathcal{H}_{\text{Schur}}$. This is the algebra of line operators of a 4d $\mathcal{N} = 2$ supersymmetric gauge theory (for us: the pure $SU(N)$ SW theory) in the holomorphic-topological twist. From it we can compute the Schur half-index $\mathbb{I}_{\text{Schur}}$. The core of the paper is about connecting these to various different formulations: in terms of a $\mathfrak{q}$-Weyl algebra representation, or in terms of $\mathfrak{q}$-oscillators. The latter connects directly in the case of Wilson and dyonic lines to generalized chord counting. It can also be identified in terms of $\mathfrak{q}$-deformed Toda quantum mechanics in the case of Wilson lines only. Solid lines are relations that hold for both Wilson and dyonic line operators, dashed lines indicate connections that we find for Wilson lines only.

the quantum gravity dynamics of the length of the Einstein-Rosen wormhole connecting the two sides of the eternal black hole. In looking for extensions, having so many equivalent descriptions is invaluable in view of generalizations.

In this paper, we explore correspondences for the Schur half-indices of $\mathcal{N} = 2$ pure gauge theories based on an $SU(N)$ gauge group. We find several alternative descriptions: they can be equivalently computed using colored chord counting (see figure 1 for an example), in terms of a transfer matrix acting on an auxiliary chord Hilbert space. Equivalently, they are related to the relativistic open Toda chain [26] based on the Lie algebra $su(N)$, see figure 2. However, so far we were unable to find a generalized SYK model, whose double-scaling limit would give rise to this structure, nor a Toda CFT description.

**Summary of Results.** Let us explain the strategy that led to these findings, summarized in figure 2. The central object is the algebra of line operators in the holomorphic-topological twist of $\mathcal{N} = 2$ SYM with gauge group $SU(N)$: $\mathcal{A}_{\text{Schur}}$. The representation of this algebra on a Hilbert space $\mathcal{H}_{\text{Schur}}$, gives a direct way to determine the Schur index $\mathbb{I}_{\text{Schur}}$. We find two ways of achieving this: using the $\mathfrak{q}$-Weyl algebra (or quantum torus algebra), and $\mathfrak{q}$-deformed harmonic oscillators, respectively.

We start from the $\mathfrak{q}$-Weyl algebra side: The recent work [17] connected the Schur index to the algebra of lines through the procedure called 'Real Schur Quantization'. They showed that the lines can be built as composites of a convenient set of operators that obey the relatively tractable $\mathfrak{q}$-Weyl algebra. We extend and develop the $\mathfrak{q}$-Weyl algebra representation of line operators for $SU(N)$ gauge theories in section 2. That is, we determine the Wilson lines and dyonic lines of minimal $SU(N)$ magnetic flux in terms of the $\mathfrak{q}$-Weyl algebra generators. We derive the algebra $\mathcal{A}_{\text{Schur}}$ in detail for $SU(3)$ and we provide some extensions for general $N$ in section 2.3. For general $N > 3$ we only discuss a subset of dyonic lines and leave the general analysis for the future.

The perhaps more surprising representation of the $\mathcal{A}_{\text{Schur}}$ algebra is in terms of $\mathfrak{q}$-deformed harmonic oscillators that we obtain in section 3. For $SU(N)$ we can represent the Wilson line operators in terms of mutually commuting sets of oscillators $a_i$ and their conjugates $a_i^\dagger$ for $i = 1, \cdots, N - 1$, satisfying

$$\left[a_i, a_i^\dagger\right]_{\mathfrak{q}} = a_i a_i^\dagger - \mathfrak{q}\, a_i^\dagger a_i = 1 \,. \tag{1.1}$$

This representation of the Wilson lines maps the computation of the Schur half-index with Wilson line operator insertions to a simple evaluation of the vacuum expectation value (VEV) on the Fock space of these $\mathfrak{q}$-deformed harmonic oscillators! We also show explicitly for $SU(3)$ how one can construct $\mathfrak{q}$-oscillator representations of the dyonic line operators, thus making it

possible to evaluate the index with arbitrary Wilson line and dyonic line insertions as oscillator VEVs. In particular for dyonic lines, the computation of the index had not been done prior to this work using standard index methods.

The representation of line operators in terms of $\mathfrak{q}$-oscillators motivates the third connection, namely to generalized chord counting, which we develop in section 4. The observation in [1] is to note that the fundamental Wilson line for $SU(2)$ SYM is represented in terms of the $\mathfrak{q}$-oscillators as $W_{[1]} = (1-\mathfrak{q})^{\frac{1}{2}}(a+a^{\dagger})$, which matches the transfer matrix of DSSYK (up to an overall $(1-\mathfrak{q})^{\frac{1}{2}}$ factor). The expectation values of the powers of the fundamental Wilson line $\langle W_{[1]}^k \rangle$ – which are the Schur half-indices with these lines inserted – then have an immediate interpretation as the moments of the DSSYK Hamiltonian! This in turn can be obtained from an effective chord counting (weighted by $\mathfrak{q}$-powers).

In section 4 we provide the generalized chord counting construction which reproduces the Schur half-index with any insertion of Wilson lines and dyonic line operators. We explain the construction of the chord diagrams for the Wilson lines in any representation of $SU(N)$, and how dyonic lines can be realized using the insertion of so called *matter chords* for the case of $SU(3)$. Key to this construction is the representation of a line in terms of the $\mathfrak{q}$-oscillators, which we refer to in this context as the *transfer matrix*, that maps naturally to the construction of chord counting rules. Each term in the transfer matrix associated to a line operator can be understood as a boundary vertex rule for some multi-valency creation and annihilation process involving colored chords. Following this line of reasoning gives a purely combinatorial derivation of the Schur half-index for $SU(N)$ SYM for any line operator insertion!

Finally in section 5 we provide a concrete proof for generic $SU(N)$ of the statement that the Schur half-indices with insertions of Wilson lines match the VEVs of the corresponding $\mathfrak{q}$-oscillator operators. This proof will allow us to establish an intriguing connection to the quantum Toda chain, the last link of the diagram in figure 2. Specifically, the subalgebra of $\mathcal{A}_{\text{Schur}}$ consisting of only the Wilson lines is generated for $SU(N)$ by those in the $N-1$ irreducible representations associated to each node of the Dynkin diagram. Their $\mathfrak{q}$-oscillator representations are $N-1$ commuting operators that we can diagonalize simultaneously. We show that the resulting spectral problem is the same one of the $N-1$ Hamiltonians of the $\mathfrak{q}$-Toda chain of type $\mathfrak{su}(N)$ [26], or equivalently that the Wilson lines are isospectral to the Toda Hamiltonians. The eigenfunctions of the $\mathfrak{q}$-Toda chain, and thus also of the $SU(N)$ Wilson lines, are known to be a class of orthogonal polynomials called $\mathfrak{q}$-*Whittaker polynomials* (see e.g. [26–30]). For $N = 2$ these polynomials reduce to the $\mathfrak{q}$-Hermite polynomials, which appear in a similar way in the study of DSSYK [20]. The knowledge of these eigenfunctions will allow

us to prove various non-trivial results, such as the aforementioned identity between Schur half-indices and $\mathfrak{q}$-oscillator VEVs for the case of the Wilson lines. We then further discuss how to properly take the $\mathfrak{q} \to 1$ limit of the Wilson line operators so to recover the conserved charges of the classical Toda chain, generalizing a similar analysis done for $SU(2)$ in [22] in the context of DSSYK. We should mention that the appearence of Toda integrable systems in the study of four-dimensional SYM theories has been observed before (see e.g. [4, 31]). It would be interesting to understand whether the way it arises from our study of Schur indices is somehow related to these previously known constructions. We conclude in section 6 and discuss some further questions and extensions.

## 2 Schur Half-Index and $\mathfrak{q}$-Weyl Representation of $\mathcal{A}_{\text{Schur}}$

We start with an overview of Schur (half-)indices and the algebra $\mathcal{A}_{\text{Schur}}$ of lines in the holomorphic-topological twisted theories. We then derive the algebra $\mathcal{A}_{\text{Schur}}$ for $SU(N)$ pure SYM, and discuss its representation.

### 2.1 Schur Indices and the Schur Algebra $\mathcal{A}_{\text{Schur}}$

For any 4d $\mathcal{N} = 2$ supersymmetric quantum field theory, the *Schur index* can be defined as a refined Witten index [15, 16] (see also Appendix A)

$$\mathcal{I} = \text{Tr} \, (-1)^F \mathfrak{q}^{j_2 - j_1 + R} \,, \tag{2.1}$$

where $j_1$, $j_2$ are the Lorentz spins, $R$ is the generator of the Cartan of the $SU(2)_R$ R-symmetry and the trace is taken over all the Schur operators which are defined as those annihilated by the supercharges $Q_{1+}$ and $\tilde{Q}_{1\dot{-}}$. Notice that the Schur index only requires the theory to possess an $SU(2)_R$ R-symmetry, but not a $U(1)_r$ R-symmetry. This enables the definition also for non-conformal theories, like the $SU(N)$ SYM theories that we consider in this paper, for which $U(1)_r$ is anomalous. Moreover, we will assume that $\mathfrak{q}$ is real and $0 < \mathfrak{q} < 1$ so that the index satisfies certain positivity properties [17] that we will review momentarily.

The Schur index can also be understood in two other ways. The first one is as a partition function over $S^3 \times S^1$ with a particular choice of background for the $SU(2)_R$ R-symmetry in order to preserve enough supersymmetry for it to be computable via supersymmetric localization. From this perspective, the parameter $\mathfrak{q}$ is related to the ratio of the radii of $S^3$ and $S^1$ (in such a way that $\mathfrak{q} \to 1$ corresponds to the $S^1$ being much smaller than the $S^3$) and to the R-symmetry background field. However, to the best of our knowledge such a localization computation has not yet been carried out explicitly. The second one is as the Euler character

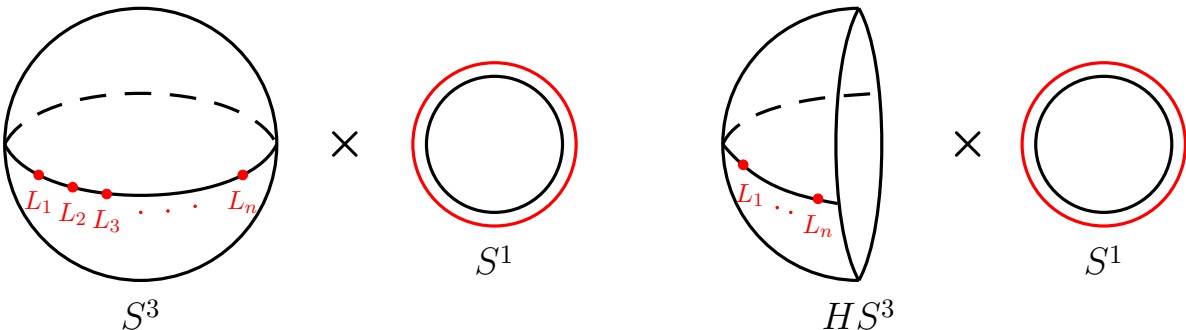

Figure 3: Schematic representation of the Schur index as a partition function on $S^3 \times S^1$ (on the left) and of the Schur half-index as a partition function on $HS^3 \times S^1$ (on the right). These indices can be decorated with the insertion of $\frac{1}{2}$-BPS lines represented in red, which wrap the $S^1$ and sit on the same equator of $S^3$ or half equator of $HS^3$.

of the complex of local operators in the holomorphic-topological (HT) twist $\mathbb{R}^2 \times \mathbb{C}$ [14, 32], where we twist by the $SU(2)_R$ R-symmetry. From this perspective, the parameter $\mathfrak{q}$ is related to a grading of the complex by a combination of the rotations on $\mathbb{C}$ and of the R-symmetry. In the present work, we will mainly use the index interpretation.

The Schur index admits the insertion of $\frac{1}{2}$-BPS line operators [18, 33–35].[1] We will in particular consider half-line defects, although also full lines are admitted. From the $S^3 \times S^1$ partition function perspective we can insert a collection of lines $L_i$ with $i = 1, \cdots, n$ by wrapping them on the $S^1$ and placing them along the equator of $S^3$, see the picture on the left of figure 3 (a full line would occupy antipodal points on this equator). What we are computing is then the correlation function of such operators in the $S^3 \times S^1$ background, sometimes called the Schur correlation function. From the HT twist perspective, they extend radially from the origin of $\mathbb{R}$. The index perspective is instead that we are counting operators which can live at the end of the line. The two perspectives are related by conformal mapping of the plane with the half-line starting from the origin to the cylinder $S^3 \times \mathbb{R}$ and subsequent compactification of $\mathbb{R}$ to $S^1$.

Because all line operators should be placed along the same equator of $S^3$, it is not possible to move them and exchange their order without having them collide with each other. We then need to consider the fusion of lines, which is described by a non-commutative $\star$-algebra over $\mathbb{Z}[\mathfrak{q}^{\frac{1}{2}}, \mathfrak{q}^{-\frac{1}{2}}]$ called the *Schur algebra* $\mathcal{A}_{\text{Schur}}$ [17, 33, 34, 51–56].

Supersymmetric partition functions on suitable manifolds $M$ are known to be factorizable into building blocks that are usually called *holomorphic blocks* [57–62] or, in the case of indices, *half-indices* [34]. Such blocks can themselves be understood as partition functions

---

[1]Various aspects of (full) Schur indices with the insertion of lines have been investigated in [36–50].

on a manifold $\widetilde{M}$ with a boundary, from which one can obtain $M$ via some gluing operation. In spite of this, rather than by applying directly localization on $\widetilde{M}$, they were originally computed by explicitly evaluating the integral of the full partition function on $M$ and then by showing that the result takes the form of a sum of contributions that holomorphically factorize. Moreover, the integrand of the full partition function on $M$ usually also factorizes into objects that correspond to the integrand of the half partition function over $\widetilde{M}$. Schematically, one has

$$\mathcal{Z}_M = \oint \|\Upsilon\|_g^2 = \sum_{i:\,\text{vac}} \left\|\mathcal{B}_{\widetilde{M}}^{(i)}\right\|_g^2, \qquad \mathcal{B}_{\widetilde{M}}^{(i)} = \oint_{\gamma_i} \Upsilon, \tag{2.2}$$

where $\|\cdot\|_g^2$ is a suitable square operation that depends on the type of gluing that relates $M$ and $\widetilde{M}$, and the sum is over certain supersymmetric vacua of the theory to each of which corresponds an integration contour $\gamma_i$ that is related to the different types of boundary conditions we can choose on $\partial\widetilde{M}$. From the index perspective, in which the contour is simply taken to be the unit circle, one counts certain gauge invariant protected operators on the space with the boundary. The bulk degrees of freedom that are given Dirichlet boundary conditions will not contribute to the index, implying that the integrand of the half-index is, roughly speaking, the square root of the one of the full index.

The Schur index is no exception. The space $S^3 \times S^1$ can be factorized in two copies of $HS^3 \times S^1$, where $HS^3$ is the hemisphere (see e.g. [18, 34, 61, 63, 64]). We will choose Neumann boundary conditions for the 4d $\mathcal{N} = 2$ vector multiplet, which in $\mathcal{N} = 1$ notation corresponds to Neumann for the $\mathcal{N} = 1$ vector multiplet and Dirichlet for the $\mathcal{N} = 1$ adjoint chiral multiplet. These will be the only boundary conditions that we need to specify, since SYM does not contain any other fields. Moreover, for most of our discussion we will not add any 3d boundary degrees of freedom (except in subsection 4.4). By our previous discussion, the resulting quantity can be called the *Schur holomorphic block* since it should be obtainable as the Schur limit of the holomorphic blocks worked out in [61] or the *Schur half-index* [18] if we employ the index perspective. We will mostly use the latter name, since the integration contour that we will use is the unit circle, which is better understood from the index perspective. In fact, the resulting Schur half-index counts certain gauge invariant protected operators, but in the presence of the boundary

$$\mathbb{I} = \text{Tr}_\partial (-1)^F \mathfrak{q}^{j_2 - j_1 + R}. \tag{2.3}$$

Also the Schur half-index can be decorated by line defects as the full Schur index, see the picture on the right of figure 3, however in this case we can only insert half lines due to the factorization of the geometry. For the case of our interest of 4d $\mathcal{N} = 2$ $SU(N)$ SYM with the insertion of Wilson lines we can express the Schur half-index as a matrix integral. Considering

a configuration with $k_1$ Wilson lines in a representation $\mathcal{R}_1$ of $SU(N)$, $k_2$ in a representation $\mathcal{R}_2$ and so on, such matrix integral is (see Appendix A for more details)

$$\mathbb{I}^{(N)}_{W^{k_1}_{\mathcal{R}_1}, W^{k_2}_{\mathcal{R}_2}, \cdots} = \frac{(\mathfrak{q};\mathfrak{q})^{N-1}_\infty}{N!} \oint_{\mathbb{T}^{N-1}} \prod_{a=1}^{N-1} \frac{\mathrm{d}z_a}{2\pi i z_a} \prod_{a<b}^{N} \left( (z_a z_b^{-1})^{\pm 1} ; \mathfrak{q} \right)_\infty \prod_i \left( \chi_{\mathcal{R}_i}(\vec{z}) \right)^{k_i} \Bigg|_{\prod_a z_a = 1} , \qquad (2.4)$$

where each variable $z_a$ is integrated over the unit circle, $(x;\mathfrak{q})_\infty = \prod_{k=0}^\infty (1 - x\mathfrak{q}^k)$ is the $\mathfrak{q}$-Pochhammer symbol, and the $\pm 1$ notation is a shorthand for taking the product of both sign choices. In this expression, the product of $\mathfrak{q}$-Pochhammer symbols encodes the contribution of the vector multiplet, while the contribution of the Wilson lines is expressed in terms of the character $\chi_{\mathcal{R}}(\vec{z})$ of the associated representation $\mathcal{R}$ of $SU(N)$. Moreover, we will work in a basis where for the fundamental representation we have

$$\chi_{[1,0,\cdots,0]}(\vec{z}) = z_1 + \sum_{a=2}^n z_a z_{a+1}^{-1} + z_{N-1}^{-1} , \qquad (2.5)$$

where we use Dynkin labels for the representations. An equivalent way of expressing the integral (2.4) is by performing the change of variables $z_a = \mathrm{e}^{i\theta_a}$

$$\mathbb{I}^{(N)}_{W^{k_1}_{\mathcal{R}_1}, W^{k_2}_{\mathcal{R}_2}, \cdots} = \frac{(\mathfrak{q};\mathfrak{q})^{N-1}_\infty}{N(2\pi)^{N-1}} \int_{\mathcal{D}_N} \prod_{a=1}^{N-1} \mathrm{d}\theta_a \prod_{a<b}^{N} \left( \mathrm{e}^{\pm i(\theta_a - \theta_b)} ; \mathfrak{q} \right)_\infty \prod_i \left( \chi_{\mathcal{R}_i}\left( \mathrm{e}^{i\vec{\theta}} \right) \right)^{k_i} \Bigg|_{\sum_a \theta_a = 0} , \qquad (2.6)$$

where the integration domain is (see e.g. (4.9) of [65])

$$\mathcal{D}_N = \left\{ (\theta_1, \cdots, \theta_{N-1}) \, \middle| \, -\pi \le \theta_1 \le \cdots \le \theta_{N-1} \le \pi \right\} . \qquad (2.7)$$

From the above expressions it is evident that the index is independent of the order of the insertion of the Wilson lines. The Wilson lines indeed fuse following the tensor products of the corresponding representations and so, even though the full Schur algebra $\mathcal{A}_{\text{Schur}}$ is non-commutative, the subalgebra generated by the Wilson lines is commutative. For $SU(N)$ any representation can be obtained by taking tensor products of the $N-1$ representations corresponding to one of the nodes of the Dynkin diagram each. We will denote these representations by

$$\mathcal{R}_r = [\underbrace{0, \cdots, 0}_{r-1}, 1, \underbrace{0, \cdots, 0}_{N-r-1}] , \qquad r = 1, \cdots, N-1 . \qquad (2.8)$$

The corresponding Wilson lines $W_{\mathcal{R}_r}$ then constitute a set of $N-1$ commuting generators of such subalgebra. Some standard fusion rules satisfied by the Wilson lines are for example

those involving the fundamental and antifundamental representations

$$W_{[1,0,\cdots,0]}W_{[n_1,\cdots,n_{N-1}]} = W_{[n_1+1,n_2,\cdots,n_{N-1}]} + \sum_{i=1}^{N-2} W_{[n_1,\cdots,n_i-1,n_{i+1}+1,\cdots,n_{N-1}]} + W_{[n_1,\cdots,n_{N-2},n_{N-1}-1]}\,,$$

$$W_{[0,\cdots,0,1]}W_{[n_1,\cdots,n_{N-1}]} = W_{[n_1-1,n_2,\cdots,n_{N-1}]} + \sum_{i=1}^{N-2} W_{[n_1,\cdots,n_i+1,n_{i+1}-1,\cdots,n_{N-1}]} + W_{[n_1,\cdots,n_{N-2},n_{N-1}+1]}\,.$$
$$(2.9)$$

The index is much more complicated with the insertion of more general line operators, such as 't Hooft or dyonic lines. However one can still study the Schur algebra $\mathcal{A}_{\text{Schur}}$ quite explicitly and in the rest of this section we will employ the techniques of [17] to analyze it in the 4d $\mathcal{N} = 2$ $SU(N)$ SYM.

## 2.2 $\mathfrak{q}$-Weyl Algebra Description of $\mathcal{A}_{\text{Schur}}$ for $SU(N)$

In this section we will derive the Schur algebra $\mathcal{A}_{\text{Schur}}$ for 4d $\mathcal{N} = 2$ pure $SU(N)$ SYM, representing the lines in terms of the $\mathfrak{q}$-Weyl algebra (i.e. the quantum torus algebra).

### Auxiliary Variables and $\mathfrak{q}$-Weyl Algebra

The spectrum of line operators of a 4d $\mathcal{N} = 2$ theory can be characterized in terms of a *quantum torus algebra* (see e.g. [18, 66, 67]), which is described by the integral lattice $\Gamma$ of electric and magnetic charges equipped with a Dirac pairing $\langle \, , \, \rangle$ [2]

$$X_\gamma X_{\gamma'} = \mathfrak{q}^{\langle \gamma, \gamma' \rangle} X_{\gamma'} X_\gamma\,, \qquad \forall \gamma, \gamma' \in \Gamma\,. \tag{2.10}$$

This quantum torus algebra is used to compute the Schur index of the 4d $\mathcal{N} = 2$ theory in terms of its IR description as a $U(1)^r$ abelian gauge theory on a generic point of the Coulomb branch [67], where $r$ is the rank of the theory, i.e. the complex dimension of its Coulomb branch.[3] From this perspective, the variables $X_\gamma$ are associated to the abelian lines of such IR theory. In order to study the lines of the original UV theory, one has to express the UV lines in terms of the abelian IR lines $X_\gamma$ and then use the quantum torus algebra relations to analyze their properties [18].

To determine the $SU(N)$ Schur algebra, we can start from the one for $U(N)$ in [17], which we denote by $\mathcal{A}'_{\text{Schur}}$, and find a suitable restriction. For $U(N)$ SYM the quantum torus algebra is particularly simple: it is given by two sets of $N$ variables $U_i$, $V_i$ for $i = 1, \cdots, N$ that

---

[2]We use a slightly different convention, which is related to the standard $q$-deformed Weyl algebra by $\mathfrak{q} = q^2$. This will ease comparison with $\mathfrak{q}$-deformed oscillators and the Schur half-index.

[3]The Coulomb branch low energy $U(1)^r$ theory is not an absolute theory, thus its lines are not necessarily mutually local. This fact is encoded in the quantum torus algebra (2.10).

satisfy the $\mathfrak{q}$-Weyl algebra

$$U_i V_j = \mathfrak{q}^{\delta_{ij}} V_j U_i\,,$$
$$U_i U_j = U_j U_i\,, \qquad\qquad i,j = 1,\cdots,N\,, \qquad\qquad (2.11)$$
$$V_i V_j = V_j V_i\,.$$

We will then describe the $U(N)$ Schur algebra $\mathcal{A}'_{\text{Schur}}$ in terms of these auxiliary variables $U_i$, $V_i$.

For this purpose it is also useful to define

$$U_{i,+} = \frac{\mathfrak{q}^{\frac{N-1}{2}}}{\prod_j^{j\neq i}\left(1 - V_j V_i^{-1}\right)} U_i\,,$$
$$U_{i,-} = \frac{1}{\prod_j^{j\neq i}\left(V_i V_j^{-1} - 1\right)} U_i^{-1}\,, \qquad\qquad (2.12)$$

which instead satisfy

$$U_{i,\pm} V_j = \mathfrak{q}^{\pm\delta_{ij}} V_j U_{i,\pm}\,. \qquad\qquad (2.13)$$

The commutation relations $U_i U_j = U_j U_i$ instead lead to very non-trivial relations between the $U_{i,\pm}$. For example, one can show that

$$U_{i,+} U_{j\neq i,-} = \frac{(-1)^N}{(1 - V_j V_i^{-1})(\mathfrak{q}^{-1} V_j V_i^{-1} - 1)} \frac{\mathfrak{q}^{\frac{N-1}{2}}}{\prod_{m,n\neq i,j}(1 - V_m V_i^{-1})(1 - V_j V_n^{-1})} U_i U_j^{-1}\,,$$
$$U_{j,-} U_{i\neq j,+} = \frac{(-1)^N}{(1 - V_j V_i^{-1})(\mathfrak{q}^{-1} V_j V_i^{-1} - 1)} \frac{\mathfrak{q}^{\frac{N-1}{2}}}{\prod_{m,n\neq i,j}(1 - V_n V_i^{-1})(1 - V_j V_m^{-1})} U_j^{-1} U_i\,. \qquad (2.14)$$

In subsection 2.3 we will derive the full set of relations satisfied by the variables $U_{i,\pm}$, $V_i$ for the $N = 3$ case.

## $\mathfrak{q}$-Weyl Representation of Line Operators

The line operators of the $U(N)$ Schur algebra $\mathcal{A}'_{\text{Schur}}$ can be expressed in terms of the auxiliary variables $U_i$, $V_i$ and we can thus compute relations among them by using the $\mathfrak{q}$-Weyl algebra (2.11). We can then restrict to combinations of $U(N)$ lines that are also $SU(N)$ lines, so to obtain the relations of the $SU(N)$ Schur algebra $\mathcal{A}_{\text{Schur}}$ we are interested in.

The line operators of SYM can be either Wilson, 't Hooft or dyonic lines [68]. The Wilson lines only depend on the variables $V_i$ and are expressed as the characters of the corresponding representation

$$W_{\mathcal{R}}(\vec{V}) = \chi_{\mathcal{R}}(\vec{V})\,. \qquad\qquad (2.15)$$

Notice that as a consequence of the $\mathfrak{q}$-Weyl algebra (2.11) these commute, as expected. Moreover, they have charge $q_e$ under the electric $U(1)_e$ 1-form symmetry equal to the $N$-ality of

the representation and trivial charge $q_m = 0$ under the magnetic $U(1)_m$ 1-form symmetry [69]. For example for the fundamental representation we have

$$W_{[1,0,\cdots,0]} = \sum_{i=1}^{N} V_i \,, \tag{2.16}$$

which has $q_e = 1$. The most basic Wilson lines for $U(N)$ are the $N-1$ in (2.8) that are also Wilson lines of $SU(N)$, for which $q_e = r$, plus the one in the determinant representation, for which $q_e = N$. In order to get the $SU(N)$ Wilson lines we should then not consider the relations involving the Wilson line in the determinant representation. Moreover, we should impose the $SU(N)$ constraint that trivializes the determinant Wilson line

$$\prod_{i=1}^{N} V_i = 1 \,. \tag{2.17}$$

We stress however that this combination of the $V_i$ variables does not commute with the $U_i$ variables, which is crucial to compute the correct algebra relations. Hence, one should impose the $SU(N)$ constraint only at the end of every computation.

Furthermore, we can construct $U(N)$ dyonic lines with minimal magnetic charge $q_m = \pm 1$ and electric charge $q_e = n$ such as

$$H_{\pm 1,n} = \sum_{i=1}^{N} \mathfrak{q}^{\pm \frac{n}{2}} V_i^n U_{i,\pm} \,. \tag{2.18}$$

However these operators are not $SU(N)$ operators. This is because they correspond to operators with the minimal possible $U(N)$ magnetic fluxes[4]

$$\mathcal{F}_+ = (0,\cdots,0,1) \,, \qquad \mathcal{F}_- = (-1,0,\cdots,0) \,, \tag{2.19}$$

which however are not $SU(N)$ fluxes, i.e. they are not part of the $SU(N)$ co-weight lattice $\lambda_w^\vee(SU(N))$.[5] In order to get actual $SU(N)$ operators we should instead consider the combinations

$$h_{n,m} = H_{1,n} H_{-1,m} \,. \tag{2.20}$$

These indeed have a magnetic flux that is the minimal possible $SU(N)$ flux[6]

$$\mathcal{F} = \mathcal{F}_+ + \mathcal{F}_- = (-1,0,\cdots,0,1) \in \Lambda_w^\vee(SU(N)) \,. \tag{2.21}$$

---

[4]Magnetic fluxes for $U(N)$ are $N$-dimensional vectors in the co-weight lattice $\Lambda_w^\vee(U(N)) \cong \mathbb{Z}^N/S_N$ of $U(N)$, where $S_N$ is the Weyl group of $U(N)$. We denote them by $\mathcal{F} = (\mathcal{F}_1,\cdots,\mathcal{F}_N) \in \mathbb{Z}^N/S_N$ and we choose to work in the Weyl chamber $-\infty \leq \mathcal{F}_1 \leq \cdots \leq \mathcal{F}_N \leq +\infty$.

[5]$SU(N)$ magnetic fluxes are those $U(N)$ magnetic fluxes that satisfy the condition $\sum_{i=1}^{N} \mathcal{F}_i = 0$.

[6]Here the choice of global structure is crucial. Indeed for $PSU(N)$ the minimal allowed flux is $\left(-\frac{1}{N},\cdots,\frac{1}{N}\right)$. We do not consider such case here.

These dyonic lines have electric charge $q_e = n + m$ and trivial magnetic charge $q_m = 0$. This is consistent with the fact that the 4d $\mathcal{N} = 2$ $SU(N)$ SYM has an electric $\mathbb{Z}_N$ 1-form symmetry but no magnetic 1-form symmetry.

We should note that there are in principle other dyonic lines that one could consider that have the same magnetic flux. The fluxes (2.19) indeed break the gauge group as

$$U(N) \to U(N-1) \times U(1) \,. \tag{2.22}$$

In the dyonic lines (2.18) we have that for each $i$ the $i$-th Cartan $U(1)$ of $U(N)$ is singled out and the sum over $i$ is to obtain Weyl invariants. We then dress by the $U(1)$ characters $V_i^n$ so to give an electric charge $q_e = n$ to the lines. However, one could also consider other lines where we dress by characters of the $U(N-1)$ part.

Taking products of these other lines one would then obtain different $SU(N)$ dyonic lines than the $h_{n,m}$ we introduced before. These still have the same magnetic flux (2.21) that breaks the gauge group as

$$SU(N) \to SU(N-2) \times U(1) \times U(1) \,. \tag{2.23}$$

The $h_{n,m}$ operators correspond to dressing by charges $n$ and $m$ for the two $U(1)$'s, but one could also consider dressing by the $SU(N-2)$ part. However, notice that for $SU(3)$ this part of the gauge group trivializes and one only has $U(1) \times U(1)$. This indicates that for $SU(3)$ the $h_{n,m}$ are enough to describe all the dyonic lines with magnetic flux (2.21) and any electric charge. In appendix C.2 we show explicitly that other dyonic lines that one might possibly consider are not actually independent from the $h_{n,m}$ introduced here.

In the next section we will use the $\mathfrak{q}$-Weyl algebra to compute the relations among these line operators for the case $N = 3$. One crucial feature of these relations involving product of Wilson and dyonic lines is that they should be homogeneous in the electric charge modulo $N$, compatibly with the fact that the electric 1-form symmetry of $SU(N)$ SYM is $\mathbb{Z}_N$.

## 2.3  Example: $\mathcal{A}_{\mathbf{Schur}}$ for $SU(3)$

Let us illustrate this general structure in terms of the concrete case of the $SU(3)$ Schur algebra $\mathcal{A}_{\text{Schur}}$. We start from the $U(3)$ $\mathfrak{q}$-Weyl algebra

$$
\begin{aligned}
U_i V_j &= \mathfrak{q}^{\delta_{ij}} V_j U_i \,, \\
U_i U_j &= U_j U_i \,, \qquad\qquad i,j = 1, 2, 3 \,, \\
V_i V_j &= V_j V_i \,.
\end{aligned}
\tag{2.24}
$$

Furthermore let

$$U_{1,+} = \frac{\mathfrak{q}}{\left(1 - V_2 V_1^{-1}\right)\left(1 - V_3 V_1^{-1}\right)} U_1 \qquad U_{1,-} = \frac{1}{\left(1 - V_1 V_2^{-1}\right)\left(1 - V_1 V_3^{-1}\right)} U_1^{-1}$$

$$U_{2,+} = \frac{\mathfrak{q}}{\left(1 - V_1 V_2^{-1}\right)\left(1 - V_3 V_2^{-1}\right)} U_2 \qquad U_{2,-} = \frac{1}{\left(1 - V_2 V_1^{-1}\right)\left(1 - V_2 V_3^{-1}\right)} U_2^{-1} \qquad (2.25)$$

$$U_{3,+} = \frac{\mathfrak{q}}{\left(1 - V_1 V_3^{-1}\right)\left(1 - V_2 V_3^{-1}\right)} U_3 \qquad U_{3,-} = \frac{1}{\left(1 - V_3 V_1^{-1}\right)\left(1 - V_3 V_2^{-1}\right)} U_3^{-1} \,,$$

so that

$$U_{i,\pm} V_j = \mathfrak{q}^{\pm \delta_{ij}} V_j U_{i,\pm} \,. \qquad (2.26)$$

Moreover, one finds that whenever $i \neq j$

$$U_{i,+} U_{j,+} = \frac{V_i - \mathfrak{q} V_j}{\mathfrak{q} V_i - V_j} U_{j,+} U_{i,+} \,,$$

$$U_{i,-} U_{j,-} = \frac{\mathfrak{q} V_i - V_j}{V_i - \mathfrak{q} V_j} U_{j,-} U_{i,-} \,, \qquad (2.27)$$

$$U_{i,+} U_{j,-} = U_{j,-} U_{i,+} \,,$$

$$U_{i,-} U_{j,+} = U_{j,+} U_{i,-} \,,$$

while for $i = j$ ($U_{i+}$ and $U_{i-}$ of course commute with themselves)

$$U_{i,+} U_{i,-} = \mathfrak{q} V_i^2 \prod_{j \neq i} \frac{V_j}{(V_i - V_j)(\mathfrak{q} V_i - V_j)} \,,$$

$$U_{i,-} U_{i,+} = \mathfrak{q} V_i^2 \prod_{j \neq i} \frac{V_j}{(V_i - V_j)(V_i - \mathfrak{q} V_j)} \,. \qquad (2.28)$$

These are all the relations that are needed in order to derive the ones satisfied by the line operators. We give the explicit proof of some of them in Appendix C.1.

Remember that for $SU(3)$ the only two independent representations are the fundamental $[1,0]$ and the anti-fundamental $[0,1]$. The relations that involve Wilson lines only correspond to the decomposition of tensor products of representations of $SU(3)$. These are given by (2.9) only, which specialized for $N = 3$ become

$$W_{[1,0]} W_{[n_1,n_2]} = W_{[n_1+1,n_2]} + W_{[n_1-1,n_2+1]} + W_{[n_1,n_2-1]} \,,$$

$$W_{[0,1]} W_{[n_1,n_2]} = W_{[n_1-1,n_2]} + W_{[n_1+1,n_2-1]} + W_{[n_1,n_2+1]} \,. \qquad (2.29)$$

More non-trivial are the relations involving the dyonic lines. We start by giving relations for the product of Wilson lines with the dyonic lines defined in (2.20). For example for the fundamental Wilson line we have (provided that $\prod_{i=1}^{3} V_i = 1$)

$$h_{n,m} W_{[1,0]} = \mathfrak{q}^{\frac{1}{2}} h_{n+1,m} + \mathfrak{q}^{-\frac{1}{2}} h_{n,m+1} + h_{n-1,m-1} - \mathfrak{q}^{\frac{n+m}{2}} W_{[n+m-2,0]}$$

$$W_{[1,0]} h_{n,m} = \mathfrak{q}^{-\frac{1}{2}} h_{n+1,m} + \mathfrak{q}^{\frac{1}{2}} h_{n,m+1} + h_{n-1,m-1} - \mathfrak{q}^{\frac{n+m}{2}} W_{[n+m-2,0]} \,, \qquad (2.30)$$

where $W_{[n,m]}$ is the Wilson line in the $SU(3)$ representation with Dynkin labels $[n,m]$ (which is taken to be zero if one of the labels is negative). We can also construct analogous expressions that allow us to lower the electric charge by involving the anti-fundamental Wilson line (again for $\prod_{i=1}^{3} V_i = 1$)

$$W_{[0,1]}h_{-n,-m} = \mathfrak{q}^{\frac{1}{2}}h_{-n-1,-m} + \mathfrak{q}^{-\frac{1}{2}}h_{-n,-m-1} + h_{-n+1,-m+1} - \mathfrak{q}^{\frac{n+m}{2}}W_{[0,n+m-2]}$$

$$h_{-n,-m}W_{[0,1]} = \mathfrak{q}^{-\frac{1}{2}}h_{-n-1,-m} + \mathfrak{q}^{\frac{1}{2}}h_{-n,-m-1} + h_{-n+1,-m+1} - \mathfrak{q}^{\frac{n+m}{2}}W_{[0,n+m-2]} \,. \tag{2.31}$$

In Appendix C.3 we give the proof for the first equation in (2.30), with the other relations being proven analogously.

Using (2.30) and (2.31), we can derive commutation relations for $h_{n,m}$ with the fundamental Wilson line

$$\left[h_{n,m}, W_{[1,0]}\right] = \left(\mathfrak{q}^{\frac{1}{2}} - \mathfrak{q}^{-\frac{1}{2}}\right)\left(h_{n+1,m} - h_{n,m+1}\right) \,, \tag{2.32}$$

and with the antifundamental Wilson line

$$\left[h_{n,m}, W_{[0,1]}\right] = \left(\mathfrak{q}^{\frac{1}{2}} - \mathfrak{q}^{-\frac{1}{2}}\right)\left(h_{n,m-1} - h_{n-1,m}\right) \,. \tag{2.33}$$

We can also use (2.30)-(2.31) to build recursion relations for the higher dyonic line operators $h_{n,m}$ in terms of the lower ones and of the Wilson lines, which will turn out to be very useful. Specifically, from (2.30) we obtain

$$h_{n+1,m} = \frac{\mathfrak{q}^{\frac{1}{2}}}{(1-\mathfrak{q})(1+\mathfrak{q})}\left(W_{[1,0]}h_{n,m} - \mathfrak{q}h_{n,m}W_{[1,0]} + (1-\mathfrak{q})\left(\mathfrak{q}^{\frac{n+m}{2}}W_{[n+m-2,0]} - h_{n-1,m-1}\right)\right)$$

$$h_{n,m+1} = \frac{\mathfrak{q}^{\frac{1}{2}}}{(1-\mathfrak{q})(1+\mathfrak{q})}\left(h_{n,m}W_{[1,0]} - \mathfrak{q}W_{[1,0]}h_{n,m} + (1-\mathfrak{q})\left(\mathfrak{q}^{\frac{n+m}{2}}W_{[n+m-2,0]} - h_{n-1,m-1}\right)\right) \,, \tag{2.34}$$

while from (2.31) we obtain

$$h_{-m-1,-n} = \frac{\mathfrak{q}^{\frac{1}{2}}}{(1-\mathfrak{q})(1+\mathfrak{q})}(h_{-m,-n}W_{[0,1]} - \mathfrak{q}W_{[0,1]}h_{-m,-n}$$

$$+ (1-\mathfrak{q})(\mathfrak{q}^{\frac{m+n}{2}}W_{[0,m+n-2]} - h_{-m+1,-n+1}))$$

$$h_{-m,-n-1} = \frac{\mathfrak{q}^{\frac{1}{2}}}{(1-\mathfrak{q})(1+\mathfrak{q})}(W_{[0,1]}h_{-m,-n} - \mathfrak{q}h_{-m,-n}W_{[0,1]}$$

$$+ (1-\mathfrak{q})(\mathfrak{q}^{\frac{m+n}{2}}W_{[0,m+n-2]} - h_{-m+1,-n+1})) \,. \tag{2.35}$$

Finally, we wish to construct algebraic relations for the dyonic line operators $h_{n,m}$. Here we will exemplify only a few of them, but more relations can be obtained in a similar way. In particular, we can prove that

$$\mathfrak{q}h_{n,0}h_{0,1} = h_{n,1}h_{0,0} \,,$$

$$\mathfrak{q}h_{n,0}h_{1,1} = h_{n,1}h_{1,0} \,, \qquad n \in \mathbb{Z} \tag{2.36}$$

$$\mathfrak{q}h_{n,1}h_{0,2} = h_{n,2}h_{0,1} \,.$$

Other relations that one can derive are the following:

$$\mathfrak{q}h_{1,0}h_{0,n} = h_{0,0}h_{1,n}\,,$$

$$\mathfrak{q}h_{1,1}h_{0,n} = h_{0,1}h_{1,n}\,, \qquad n \in \mathbb{Z} \qquad (2.37)$$

$$\mathfrak{q}h_{2,0}h_{1,n} = h_{1,0}h_{2,n}\,,$$

however these are not independent from (2.36) since they can be derived from them upon acting with a certain charge conjugation operation that we will describe momentarily. In Appendix C.4 we present the proof of the first equation in (2.36), with those of the remaining ones working in a similar manner.

## 2.4 Algebras, Modules, and Hilbert Spaces

In this subsection we will briefly review the results of [17] on the representation theory of the Schur algebra $\mathcal{A}_{\text{Schur}}$. We then focus on the 4d $\mathcal{N} = 2$ $SU(N)$ SYM theory that we are interested in.

As mentioned in subsection 2.1, the Schur algebra $\mathcal{A}_{\text{Schur}}$ is a $\star$-algebra on $\mathbb{Z}[\mathfrak{q}^{\frac{1}{2}}, \mathfrak{q}^{-\frac{1}{2}}]$. We recall that a $\star$-algebra is an algebra $\mathcal{A}$ over a field $\mathbb{F}$, in this current situation $\mathbb{Z}[\mathfrak{q}^{\frac{1}{2}}, \mathfrak{q}^{-\frac{1}{2}}]$, equipped with a map $\star \colon \mathcal{A} \to \mathcal{A}$ with the following properties for any $a, b \in \mathcal{A}$ and $\lambda \in \mathbb{F}$:[7]

- involution: $(a^\star)^\star = a$,

- anti-linearity: $(\lambda a + b)^\star = \bar{\lambda} a^\star + b^\star$,

- anti-multiplicativity: $(ab)^\star = b^\star a^\star$.

In the case of 4d $\mathcal{N} = 2$ $SU(N)$ SYM, the $\star$ operation is naturally interpreted as the action of the outer-automorphism of $SU(N)$, which for brevity we will call *charge conjugation*

$$\tau \colon \quad \begin{aligned} W_{\mathcal{R}} &\to \tau(W_{\mathcal{R}}) = W_{\overline{\mathcal{R}}} \\ h_{n,m} &\to \tau(h_{n,m}) = h_{-n,-m}\,. \end{aligned} \qquad (2.38)$$

It is not immediately obvious from this definition that $\tau$ indeed satisfies the properties listed above of the $\star$ operation. In the next section we will show that this is true by using the representation of $\mathcal{A}_{\text{Schur}}$ in terms of $\mathfrak{q}$-oscillators, where $\tau$ will be implemented by the Hermitian conjugation and so it will automatically satisfy those properties. Notice also that $\tau$ inverts the sign of the charge $q_e$ under the electric 1-form symmetry of all the line operators, as expected from charge conjugation.

---

[7]We assume that the field $\mathbb{F}$ possesses an involutive anti-automorphism, that is a map $\sigma \colon \mathbb{F} \to \mathbb{F}$ such that $\sigma^2 = \text{id}$. In the text we denoted $\bar{\lambda} \equiv \sigma(\lambda)$.

The operation defined in (2.38) simplifies in the $SU(2)$ case [1, 17], since $\tau$ acts trivially on the Wilson lines and the non-trivial action is only on the dyonic lines. The reason is that $SU(2)$ is pseudo-real and so the action of charge conjugation on its representations is trivial. However, in the presence of a magnetic flux the gauge group is broken to the subgroup $U(1) \subset SU(2)$ and charge conjugation acts by changing the sign of the charge of an operator under the $U(1)$, which explains the action on the dyonic lines.

Thanks to the fact that in the Schur half-index we choose Neumann boundary conditions, we have the possibility of inserting Wilson lines that live on the boundary of the hemisphere $HS^3$, which we will denote by $W^\partial$. These organize into a module $\mathcal{M}_{\text{Schur}}$ of $\mathcal{A}_{\text{Schur}}$ that encodes the fusion of bulk and boundary lines. Endowing this module with an inner product defined by the Schur half-index provides a positive-definite inner product, promoting the module to a Hilbert space $\mathcal{H}_{\text{Schur}}$ [17].

**GNS construction of the Hilbert Space.** More precisely, the way one can associate a Hilbert space $\mathcal{H}_{\text{Schur}}$ to the algebra $\mathcal{A}_{\text{Schur}}$ is via the Gelfand-Naimark-Segal (GNS) construction [70, 71] in the context of algebraic quantum field theory. Given a $\star$-algebra $\mathcal{A}$, a state is defined as a linear map $\omega : \mathcal{A} \to \mathbb{C}$ that is positive,[8] in the sense that $\omega(a^*a) \geq 0$ for any $a \in \mathcal{A}$. The GNS construction allows us to associate to any such state the following data:

- a Hilbert space $\mathcal{H}_\omega$,

- a representation $\pi_\omega$ of $\mathcal{A}$ on $\mathcal{H}_\omega$, namely an operator $\pi_\omega(a)$ on $\mathcal{H}_\omega$ for any $a \in \mathcal{A}$,

- a state $|\Omega_\omega\rangle$, from which the full $\mathcal{H}_\omega$ can be constructed by $\pi_\omega(\mathcal{A})|\Omega_\omega\rangle$, such that

$$\omega(a) = \langle \Omega_\omega | \pi_\omega(a) | \Omega_\omega \rangle , \qquad a \in \mathcal{A} . \tag{2.39}$$

In the case of $\mathcal{A}_{\text{Schur}}$, the map $\omega$ is given by the Schur half-index with the insertion of boundary Wilson lines $W^\partial$

$$\omega(L) = \mathbb{I}_{\tau(W^\partial) \, L \, W^\partial} , \qquad L \in \mathcal{A}_{\text{Schur}} , \tag{2.40}$$

where on the r.h.s. we mean the Schur half-index with the insertion of $W^\partial$ both on the left and on the right boundaries (the two points where the equator on which the lines are placed intersects the boundary of $HS^3$) and of $L$ in the bulk, where $L$ could also be a composite line made from a collection of lines. Indeed, the quantity

$$\omega(\tau(L)L) = \mathbb{I}_{\tau(W^\partial) \, \tau(L) \, L \, W^\partial} = \mathbb{I}_{\tau(L W^\partial) \, L \, W^\partial} , \tag{2.41}$$

---

[8]For simplicity we are assuming that $\mathbb{F} = \mathbb{C}$, but this applies also for the case $\mathbb{F} = \mathbb{Z}[\mathfrak{q}^{\frac{1}{2}}, \mathfrak{q}^{-\frac{1}{2}}]$ of $\mathcal{A}_{\text{Schur}}$.

was argued in [17] to be positive definite for $0 < \mathfrak{q} < 1$. Such a state $\omega$ can be constructed for any choice of boundary Wilson line $W^\partial$. Hence, by the GNS construction we obtain a Hilbert space

$$\mathcal{H}_{\text{Schur}} = \left\{ |W^\partial\rangle \,\middle|\, W^\partial \in \mathcal{A}_{\text{Schur}} \right\}, \tag{2.42}$$

together with a representation $\pi(L)$ of any line $L \in \mathcal{A}_{\text{Schur}}$ and a positive definite inner product

$$\langle W^\partial | \tilde{W}^\partial \rangle = \mathbb{I}_{\tau(W^\partial)\tilde{W}^\partial}. \tag{2.43}$$

In particular the vacuum of $\mathcal{H}_{\text{Schur}}$ corresponds to having no boundary Wilson line and is denoted by $|1^\partial\rangle$.

$\mathcal{H}_{\textbf{Schur}}$ **for** $SU(N)$ **SYM.** In the case of 4d $\mathcal{N} = 2$ $SU(N)$ SYM, the vacuum $|1^\partial\rangle$ of $\mathcal{H}_{\text{Schur}}$ satisfies the important property of being annihilated by some of the dyonic lines. Explicitly in the $SU(3)$ case, we find that

$$h_{0,1}|1^\partial\rangle = h_{-1,0}|1^\partial\rangle = 0. \tag{2.44}$$

These relations can be obtained as follows. We first recall that the Wilson lines, including the boundary ones, can be expressed in terms of the auxiliary variables $V_i$ as the characters of the corresponding representations, see (2.15). The dyonic lines $h_{n,m}$ are instead written in terms of both $V_i$ and $U_j$. The $U_i$ acts on the corresponding $V_i$ as difference operators $V_i \to \mathfrak{q}V_i$, as encoded in the $\mathfrak{q}$-Weyl algebra (2.11). Hence, to determine the action of a dyonic line $h_{n,m}$ on a boundary Wilson line we first re-express $h_{n,m}$ by moving all the $U$'s to the right and all the $V$'s to the left for each of the terms using (2.11). Schematically

$$h_{n,m} = \sum_k F_k(\vec{V})G_k(\vec{U}). \tag{2.45}$$

Then we can act on the boundary Wilson line by applying the difference operators $G_k(\vec{U})$ on the corresponding character. In the case of the trivial Wilson line $1^\partial$ the character is simply 1 and so the action of $G_k(\vec{U})$ is trivial, that is

$$G_k(\vec{U})|1^\partial\rangle = |1^\partial\rangle. \tag{2.46}$$

Hence, in order to check (2.44) we should show that $\sum_k F_k(\vec{V})|1^\partial\rangle = 0$, or simply

$$\sum_k F_k(\vec{V}) = 0, \tag{2.47}$$

since the Wilson lines, both boundary and bulk ones, simply fuse following the standard multiplication. Carrying out this computation for $h_{0,1}$ and $h_{-1,0}$ one can verify that (2.44)

hold true. Other remarkable relations that one can find in a similar way are

$$h_{1,0}|1^\partial\rangle = \mathfrak{q}^{\frac{3}{2}}|W^\partial_{[1,0]}\rangle\,, \qquad h_{0,-1}|1^\partial\rangle = \mathfrak{q}^{\frac{3}{2}}|W^\partial_{[0,1]}\rangle\,. \tag{2.48}$$

In Appendix C.5 we provide more details on the derivation of the relations (2.44)-(2.48). This structure hints at a more algebraic description. In the next section we will give an explicit realization of $\mathcal{H}_{\mathrm{Schur}}$ in terms of $\mathfrak{q}$-deformed quantum harmonic oscillators.

# 3 $\mathfrak{q}$-oscillator Representation of $\mathcal{A}_{\mathbf{Schur}}$

The observations regarding the modules of the $\mathcal{A}_{\mathrm{Schur}}$ algebra of 4d $\mathcal{N} = 2$ $SU(N)$ SYM that we made in the last section in fact hint at a more algebraic realization of the algebra and the associated half-index. Indeed, we will show that 4d $\mathcal{N} = 2$ $SU(N)$ SYM can be represented in terms of $N - 1$ decoupled $\mathfrak{q}$-deformed harmonic oscillators, acting on a standard Fock space. This will furthermore allow us to compute the Schur half-index in terms of simple oscillator algebra.

More precisely, we are going to provide a unitary representation of $\mathcal{A}_{\mathrm{Schur}}$, that is a representation on the $\mathfrak{q}$-oscillators Hilbert space $\mathcal{H}_{\mathrm{osc}}$ where for each $L \in \mathcal{A}_{\mathrm{Schur}}$ we have a corresponding operator $\pi(L)$ acting on $\mathcal{H}_{\mathrm{osc}}$ and such that the $\star$ operation is represented by Hermitian conjugation $\pi(L^\star) = \pi(L)^\dagger$. As we will discuss, the Schur half-indices with the insertion of lines can be computed in this representation as expectation values of the corresponding operators acting on $\mathcal{H}_{\mathrm{osc}}$.

## 3.1 $\mathfrak{q}$-oscillators for $SU(N)$

For generic $SU(N)$ we introduce $N - 1$ decoupled $\mathfrak{q}$-oscillators

$$\begin{aligned}
\left[a_i, a_i^\dagger\right]_{\mathfrak{q}} &= 1\,, & i &= 1, \cdots, N-1\,, \\
\left[a_i, a_j^\dagger\right]_1 &= 0\,, & i &\neq j = 1, \cdots, N-1\,,
\end{aligned} \tag{3.1}$$

with the $\mathfrak{q}$-commutator is defined as $[\mathcal{O}_1, \mathcal{O}_2]_{\mathfrak{q}} \equiv \mathcal{O}_1\mathcal{O}_2 - \mathfrak{q}\mathcal{O}_2\mathcal{O}_1$. Since the oscillators commute with each other, we can introduce a basis of states which are given by the tensor product of the states of each oscillator

$$|n_1, \cdots, n_{N-1}\rangle = |n_1\rangle \otimes \cdots \otimes |n_{N-1}\rangle\,, \tag{3.2}$$

on which the creation and annihilation operators act as

$$\begin{aligned}
a_i^\dagger|n_1, \cdots, n_i, \cdots, n_{N-1}\rangle &= |n_1, \cdots, n_i + 1, \cdots, n_{N-1}\rangle\,, \\
a_i|n_1, \cdots, n_i, \cdots, n_{N-1}\rangle &= \frac{1 - \mathfrak{q}^{n_i}}{1 - \mathfrak{q}}|n_1, \cdots, n_i - 1, \cdots, n_{N-1}\rangle = [n_i]_{\mathfrak{q}}|n_1, \cdots, n_{i-1}, \cdots, n_{N-1}\rangle\,.
\end{aligned} \tag{3.3}$$

The ground state $|0,\ldots 0\rangle$ is defined to be the state annihilated by the oscillators $a_i$

$$a_i|0,\ldots 0\rangle = 0\,, \qquad i = 1,\cdots, N-1\,. \tag{3.4}$$

The $\mathfrak{q}$-oscillators Hilbert space is then

$$\mathcal{H}_{\text{osc}} = \left\{ |n_1,\cdots, n_{N-1}\rangle \,\middle|\, n_i \in \mathbb{Z}_+ \right\}\,. \tag{3.5}$$

We also endow the Hilbert space of $\mathfrak{q}$-oscillators with an inner product between states such that $a_i$ and $a_i^\dagger$ are Hermitian conjugates

$$\langle m_1,\ldots,m_{N-1}|n_1,\ldots,n_{N-1}\rangle = \prod_{i=1}^{N-1} \delta_{m_i,n_i} [n_i]_{\mathfrak{q}}!\,, \tag{3.6}$$

where the $\mathfrak{q}$-deformed factorial and $\mathfrak{q}$-Pochhammer symbol are defined as

$$[n]_{\mathfrak{q}}! = \frac{(\mathfrak{q};\mathfrak{q})_n}{(1-\mathfrak{q})^n}\,, \qquad (\mathfrak{q};\mathfrak{q})_n = \prod_{k=1}^{n}(1-\mathfrak{q}^k)\,. \tag{3.7}$$

This is a basis on which the number operators $\mathfrak{n}_i$ act diagonally

$$\mathfrak{n}_i|n_1,\cdots, n_i,\cdots, n_{N-1}\rangle = n_i|n_1,\cdots, n_i,\cdots, n_{N-1}\rangle\,. \tag{3.8}$$

Such number operators satisfy the relations

$$\begin{aligned}
\left[a_i, \mathfrak{q}^{\mathfrak{n}_i}\right] &= (\mathfrak{q}-1)\mathfrak{q}^{\mathfrak{n}_i} a_i \\
\left[a_i^\dagger, \mathfrak{q}^{\mathfrak{n}_i}\right] &= (\mathfrak{q}^{-1}-1)\mathfrak{q}^{\mathfrak{n}_i} a_i^\dagger\,,
\end{aligned} \tag{3.9}$$

or more succinctly

$$\begin{aligned}
a_i \mathfrak{q}^{\mathfrak{n}_i} &= \mathfrak{q}^{\mathfrak{n}_i+1} a_i \\
a_i^\dagger \mathfrak{q}^{\mathfrak{n}_i} &= \mathfrak{q}^{\mathfrak{n}_i-1} a_i^\dagger\,.
\end{aligned} \tag{3.10}$$

We should note at this point that, as will be discussed in Section 4, $\mathfrak{q}$-deformed oscillators naturally arise in the context of chord counting, and we will be able to connect the half-index to certain generalized chord counting problems.

## 3.2 The $\mathfrak{q}$-oscillator Representation of $\mathcal{A}_{\text{Schur}}$ for $SU(3)$

### 3.2.1 Map from $\mathfrak{q}$-oscillators to $\mathcal{A}_{\text{Schur}}$

We will start by studying in detail the $\mathfrak{q}$-oscillator representation of $\mathcal{A}_{\text{Schur}}$ for the case of $SU(3)$. In this case, it is possible to introduce the $\mathfrak{q}$-oscillators $a_i^\dagger$, $a_i$ for $i = 1,2$ as the following combinations of $\mathcal{A}_{\text{Schur}}$ operators:

$$\begin{aligned}
a_1^\dagger &= \mathfrak{q}^{\frac{1}{2}}(1-\mathfrak{q})^{-\frac{1}{2}} h_{0,1} h_{0,0}^{-1}\,, \\
a_1 &= \mathfrak{q}^{-\frac{1}{2}}(1-\mathfrak{q})^{-\frac{1}{2}} h_{0,-1} h_{0,0}^{-1}\,, \\
a_2^\dagger &= \mathfrak{q}^{\frac{1}{2}}(1-\mathfrak{q})^{-\frac{1}{2}} h_{-1,0} h_{0,0}^{-1}\,, \\
a_2 &= \mathfrak{q}^{-\frac{1}{2}}(1-\mathfrak{q})^{-\frac{1}{2}} h_{1,0} h_{0,0}^{-1}\,.
\end{aligned} \tag{3.11}$$

Notice that we can express this map also in terms of the $\mathfrak{q}$-Weyl variables $U_{i,\pm}$ and $V_i$ by using the definitions of the dyonic lines $h_{n,m}$, however we prefer to express it in terms of the $h_{n,m}$ to avoid cluttering the notation. Using the definition of the number operators $a_i^\dagger a_i = [\mathfrak{n}_i]_\mathfrak{q}$, we can use the above (3.11) to derive similar expressions for the number operators

$$\mathfrak{q}^{\mathfrak{n}_1} = 1 - h_{0,1} h_{0,0}^{-1} h_{0,-1} h_{0,0}^{-1},$$
$$\mathfrak{q}^{\mathfrak{n}_2} = 1 - h_{-1,0} h_{0,0}^{-1} h_{1,0} h_{0,0}^{-1}.$$
(3.12)

It is possible to verify that with the above definitions the $\mathfrak{q}$-oscillators relations (3.1) and (3.10) hold true by using the relations satisfied by the line operators of $\mathcal{A}_{\mathrm{Schur}}$. In particular, some relations that can be proven in $\mathcal{A}_{\mathrm{Schur}}$ using the machinery developed in the previous section and that are imporant for this purpose are

$$h_{0,0}^{-1} h_{0,1} = \mathfrak{q} h_{0,1} h_{0,0}^{-1},$$
$$h_{0,0}^{-1} h_{1,0} = \mathfrak{q}^{-1} h_{1,0} h_{0,0}^{-1},$$
$$h_{0,0}^{-1} h_{0,-1} = \mathfrak{q}^{-1} h_{0,-1} h_{0,0}^{-1},$$
$$h_{0,0}^{-1} h_{-1,0} = \mathfrak{q} h_{-1,0} h_{0,0}^{-1}.$$
(3.13)

Using these identities, we can show that

$$\left[a_1, a_1^\dagger\right]_\mathfrak{q} = a_1 a_1^\dagger - \mathfrak{q} a_1^\dagger a_1 = (1 - \mathfrak{q})^{-1} \left( h_{0,-1} h_{0,0}^{-1} h_{0,1} h_{0,0}^{-1} - \mathfrak{q} h_{0,1} h_{0,0}^{-1} h_{0,-1} h_{0,0}^{-1} \right),$$
$$= (1 - \mathfrak{q})^{-1} \left( \mathfrak{q} h_{0,-1} h_{0,1} h_{0,0}^{-2} - h_{0,1} h_{0,-1} h_{0,0}^{-2} \right) = 1,$$
(3.14)

where in the last step we also made use of the $\mathcal{A}_{\mathrm{Schur}}$ identity

$$\mathfrak{q} h_{0,-1} h_{0,1} - h_{0,1} h_{0,-1} = (1 - \mathfrak{q}) h_{0,0}^2,$$
(3.15)

which can be verified using the $\mathfrak{q}$-Weyl algebra. We can similarly check that $\left[a_2, a_2^\dagger\right]_\mathfrak{q} = 1$ using in this case the $\mathcal{A}_{\mathrm{Schur}}$ relation

$$h_{-1,0} h_{1,0} - \mathfrak{q} h_{1,0} h_{-1,0} = -(1 - \mathfrak{q}) h_{0,0}^2.$$
(3.16)

Furthermore, we should check that the $a_1$ and $a_2$ oscillators commute, namely $[a_1, a_2]_1 = [a_1, a_2^\dagger]_1 = [a_1^\dagger, a_2]_1 = [a_1^\dagger, a_2^\dagger]_1 = 0$. These equations are equivalent to the following four $\mathcal{A}_{\mathrm{Schur}}$ identities:

$$h_{0,-1} h_{1,0} - h_{1,0} h_{0,-1} = 0,$$
$$h_{-1,0} h_{0,-1} - \mathfrak{q}^2 h_{0,-1} h_{-1,0} = 0,$$
$$h_{0,1} h_{1,0} - \mathfrak{q}^2 h_{1,0} h_{0,1} = 0,$$
$$h_{0,1} h_{-1,0} - h_{-1,0} h_{0,1} = 0,$$
(3.17)

which have again been verified using the $\mathfrak{q}$-Weyl algebra. Finally, we need to verify that (3.12) define number operators. For example, $a_1^\dagger \mathfrak{q}^{\mathfrak{n}_1} = \mathfrak{q}^{\mathfrak{n}_1 - 1} a_1^\dagger$ can be shown to be equivalent to the following $\mathcal{A}_{\text{Schur}}$ identity:

$$h_{0,1} h_{0,0}^2 - \mathfrak{q}^{-1} h_{0,1}^2 h_{0,-1} = \mathfrak{q}^{-1} h_{0,1} h_{0,0}^2 - h_{0,1} h_{0,-1} h_{0,1} \,. \tag{3.18}$$

Similarly, the relations $a_1 \mathfrak{q}^{\mathfrak{n}_2} = \mathfrak{q}^{\mathfrak{n}_2 + 1} a_1$, $a_2^\dagger \mathfrak{q}^{\mathfrak{n}_2} = \mathfrak{q}^{\mathfrak{n}_2 + 1} a_2^\dagger$, and $a_2 \mathfrak{q}^{\mathfrak{n}_2} = \mathfrak{q}^{\mathfrak{n}_2 - 1} a_2$ are equivalent to the $\mathcal{A}_{\text{Schur}}$ identities

$$\begin{aligned}
h_{0,-1} h_{0,0}^2 - \mathfrak{q}^{-1} h_{0,-1} h_{0,1} h_{0,-1} &= \mathfrak{q} h_{0,-1} h_{0,0}^2 - \mathfrak{q}^{-2} h_{0,1} h_{0,-1}^2 \,, \\
h_{-1,0} h_{0,0}^2 - \mathfrak{q}^{-1} h_{-1,0}^2 h_{1,0} &= \mathfrak{q}^{-1} h_{-1,0} h_{0,0}^2 - h_{-1,0} h_{1,0} h_{-1,0} \,, \\
h_{1,0} h_{0,0}^2 - \mathfrak{q}^{-1} h_{1,0} h_{-1,0} h_{1,0} &= \mathfrak{q} h_{1,0} h_{0,0}^2 - \mathfrak{q}^{-2} h_{-1,0} h_{1,0}^2 \,.
\end{aligned} \tag{3.19}$$

Since all of these required equations do indeed hold in $\mathcal{A}_{\text{Schur}}$, we can conclude that the map (3.11) is genuinely a map to $\mathfrak{q}$-deformed harmonic oscillators. The power of this map will become useful later, as it will allow us to check that the candidate representation for a Wilson or dyonic line is indeed mapped to the corresponding line in $\mathcal{A}_{\text{Schur}}$.

### 3.2.2 Representation of Wilson Lines for $SU(3)$

Our next task is to determine how the line operators of $\mathcal{A}_{\text{Schur}}$ are represented in terms of the $\mathfrak{q}$-oscillators. We begin by discussing the Wilson lines and in the next subsection we will instead study the dyonic lines $h_{n,m}$.

For $SU(3)$ we only have two independent Wilson lines (2.8) corresponding to the fundamental $[1,0]$ and the antifundamental $[0,1]$ representations, which we recall that in terms of the $\mathfrak{q}$-Weyl variables $V_i$ read

$$\begin{aligned}
W_{[1,0]} &= V_1 + V_2 + V_3 \\
W_{[0,1]} &= V_1^{-1} + V_2^{-1} + V_3^{-1} \,.
\end{aligned} \tag{3.20}$$

We find that these Wilson lines can be represented in terms of $\mathfrak{q}$-oscillators as follows:[9]

$$\begin{aligned}
W_{[1,0]} &= (1 - \mathfrak{q})^{\frac{1}{2}} \left( a_1^\dagger + (1 - \mathfrak{q})^{\frac{1}{2}} a_1 a_2^\dagger + a_2 \right) \\
W_{[0,1]} &= (1 - \mathfrak{q})^{\frac{1}{2}} \left( a_2^\dagger + (1 - \mathfrak{q})^{\frac{1}{2}} a_1^\dagger a_2 + a_1 \right) \,.
\end{aligned} \tag{3.21}$$

One can indeed check that the expressions (3.20) are reproduced upon application of the map (3.11). For a proof, we expand out the definition of the Wilson lines in terms of the dyonic lines, use the relations (3.13) to commute the $h_{0,0}^{-1}$ terms to the right, and multiply both sides

---

[9]So far we have denoted the representation of a line operator $L \in \mathcal{A}_{\text{Schur}}$ by $\pi(L)$. From now on we simplify the notation and still denote by $L$ the representation of the line operator on the $\mathfrak{q}$-oscillators Hilbert space.

of the resulting equation by $h_{0,0}^2$ from the right. For the fundamental Wilson line for example, we get in this way the equation

$$W_{[1,0]} h_{0,0}^2 = \mathfrak{q}^{\frac{1}{2}} h_{0,1} h_{0,0} + \mathfrak{q} h_{0,-1} h_{-1,0} + \mathfrak{q}^{-\frac{1}{2}} h_{1,0} h_{0,0} \,. \tag{3.22}$$

A quick check using the $\mathfrak{q}$-Weyl algebra verifies this equation is true (provided $\prod_{i=1}^3 V_i = 1$), hence showing the Wilson line in the $\mathfrak{q}$-oscillators representation is indeed mapped to the Wilson line in the $\mathfrak{q}$-Weyl representation. A similar proof can be performed for the antifundamental Wilson line.

Another non-trivial check is that the Wilson line operators commute

$$\left[ W_{[1,0]}, W_{[0,1]} \right] = 0 \,, \tag{3.23}$$

as expected since the Wilson lines commute in $\mathcal{A}_{\text{Schur}}$. We also note that Hermitian conjugation acts on these operators as

$$W_{[1,0]}^\dagger = W_{[0,1]} \,. \tag{3.24}$$

This indicates that hermitian conjugation in the $\mathfrak{q}$-oscillators representation is implementing the $\star$ operation of $\mathcal{A}_{\text{Schur}}$ that we discussed in subsection 2.4.

We can check explicitly the following identity between the Schur half-index with the insertion of $k_1$ fundamental and $k_2$ antifundamental Wilson lines and the VEVs of the corresponding $\mathfrak{q}$-oscillators representation for low values of $k_1$ and $k_2$:

$$\mathbb{I}_{W_{[1,0]}^{k_1}, W_{[0,1]}^{k_2}}^{(N)} = \left\langle W_{[1,0]}^{k_1} W_{[0,1]}^{k_2} \right\rangle \,. \tag{3.25}$$

For example if we insert only fundamental or only antifundamental Wilson lines, we have

$$\begin{aligned}
\left\langle W_{[1,0]}^3 \right\rangle = \left\langle W_{[0,1]}^3 \right\rangle = \mathbb{I}_{W_{[1,0]}^3}^{(3)} = \mathbb{I}_{W_{[0,1]}^3}^{(3)} = (1 - \mathfrak{q})^2 \,, \\
\left\langle W_{[1,0]}^6 \right\rangle = \left\langle W_{[0,1]}^6 \right\rangle = \mathbb{I}_{W_{[1,0]}^6}^{(3)} = \mathbb{I}_{W_{[0,1]}^6}^{(3)} = (1 - \mathfrak{q})^4 (5 + 4\mathfrak{q} + \mathfrak{q}^2) \,,
\end{aligned} \tag{3.26}$$

while all the powers that are not a multiple of 3 vanish.

Notice that the indices with the insertion of fundamental or antifundamental Wilson lines are identical. This is a general feature of the Schur half-index of $SU(N)$ SYM, that is the index with the insertion of a Wilson line in a representation $\mathcal{R}$ is identical to the one with a line in the conjugate representation $\overline{\mathcal{R}}$

$$\mathbb{I}_{W_{\mathcal{R}}}^{(N)} = \mathbb{I}_{W_{\overline{\mathcal{R}}}}^{(N)} \,. \tag{3.27}$$

This can be understood at the level of the integral expression (2.4), where the two indices are related by the change of variables $z_a \to z_a^{-1}$. Such a transformation is implementing the action

of charge conjugation on the theory. The part of the integral given by the $\mathfrak{q}$-Pochhammers is left invariant, which encodes the fact that the theory is charge conjugation invariant. On the other hand, the contribution of the Wilson line is not invariant and in particular we have

$$\chi_{\mathcal{R}}(z_1^{-1}, \cdots, z_{N-1}^{-1}) = \chi_{\overline{\mathcal{R}}}(z_1, \cdots, z_{N-1}) \, , \tag{3.28}$$

from which (3.27) follows.

If instead we insert the same number $k_1 = k_2 = k$ of fundamental and antifundamental Wilson lines, we have

$$
\begin{aligned}
\langle W_{[1,0]} W_{[0,1]} \rangle &= \mathbb{I}^{(3)}_{W_{[1,0]}, W_{[0,1]}} = (1 - \mathfrak{q})^1 \, , \\
\langle \left( W_{[1,0]} W_{[0,1]} \right)^2 \rangle &= \mathbb{I}^{(3)}_{W^2_{[1,0]}, W^2_{[0,1]}} = 2(1 - \mathfrak{q})^2 \, , \\
\langle \left( W_{[1,0]} W_{[0,1]} \right)^3 \rangle &= \mathbb{I}^{(3)}_{W^3_{[1,0]}, W^3_{[0,1]}} = (1 - \mathfrak{q})^3 (6 + \mathfrak{q} - \mathfrak{q}^2) \, .
\end{aligned}
\tag{3.29}
$$

Later on in this section we will give a more abstract motivation of the identity (3.25) between the Schur half-indices with the insertion of Wilson lines and the VEVs of the corresponding Wilson lines in the $\mathfrak{q}$-oscillator representation by showing that the vacuum states of $\mathcal{H}_{\text{Schur}}$ and $\mathcal{H}_{\text{osc}}$ are related. In Section 5 we will instead give a more direct proof that the VEVs of the Wilson lines admit the same integral expression (2.4) as the Schur half-indices by diagonalizing the Wilson lines in their oscillator representation.

**Other Irreducible Representations.** From the fundamental and antifundamental representations of $SU(3)$ we can construct any other irreducible representation by considering tensor products. As mentioned in subsection 2.1, the Wilson lines fuse following the decomposition into irreducible representations of these tensor products. We can thus use this property to determine the $\mathfrak{q}$-oscillator expressions for a generic Wilson line in any irreducible representation of $SU(3)$ by starting from the ones in (3.21) for the fundamental and antifundamental Wilson lines.

Let us consider for example the tensor product decomposition $[1,0] \otimes [0,1] = [1,1] \oplus [0,0]$. The singlet Wilson line is expected to be trivial, since the index with its insertion actually coincides with the index with no insertion of line operators at all

$$W_{[0,0]} = 1 \, . \tag{3.30}$$

Now, knowing the oscillator representation of the Wilson lines for the $[1,0]$ and $[0,1]$ representations, we can extract $W_{[1,1]}$ using

$$W_{[1,1]} = W_{[1,0]} W_{[0,1]} - 1 \, , \tag{3.31}$$

which is given explicitly by

$$W_{[1,1]} = -\mathfrak{q} + (1-\mathfrak{q})\left(a_1 a_2 + a_1^\dagger a_2^\dagger + a_1^\dagger a_1 + a_2^\dagger a_2 + (1-\mathfrak{q})^{\frac{1}{2}}\left((a_1^\dagger)^2 a_2 + a_1^\dagger a_2^2 + a_1^2 a_2^\dagger + a_1(a_2^\dagger)^2\right)\right)$$
$$+ (1-\mathfrak{q})^2 \mathfrak{q}\, a_1^\dagger a_1 a_2^\dagger a_2\,.$$

(3.32)

Another example, which will be useful for computing the oscillator representations of the dyonic lines $h_{n,m}$, is given by $[1,0] \otimes [1,0] = [2,0] \oplus [0,1]$. Using this, one finds

$$W_{[2,0]} = W_{[1,0]} W_{[1,0]} - W_{[0,1]}$$
$$= (1-\mathfrak{q})^{\frac{1}{2}}\left(-(a_1 + a_2^\dagger)\mathfrak{q} + (1-\mathfrak{q})^{\frac{1}{2}}\left(a_1^\dagger a_1^\dagger + a_1^\dagger a_2 + a_2 a_2\right)\right)$$
$$+ (1-\mathfrak{q})^{\frac{3}{2}}\left(a_1 a_1 a_2^\dagger a_2^\dagger + (1+\mathfrak{q})\left(a_1 a_2^\dagger a_2 + a_1^\dagger a_1 a_2^\dagger\right)\right)\,.$$

(3.33)

Following this procedure, one can construct the oscillator representation for any Wilson line in an irreducible representation of $SU(3)$. To validate our proposal further, we can verify for low $k$ (and indeed we will prove later in subsection 3.2.4) that the VEVs of these Wilson lines agree with the Schur half-index

$$\langle W_{[1,1]} \rangle = \mathbb{I}^{(3)}_{W_{[1,1]}} = -\mathfrak{q}\,,$$
$$\langle W^2_{[1,1]} \rangle = \mathbb{I}^{(3)}_{W^2_{[1,1]}} = 1 - 2\mathfrak{q} + 2\mathfrak{q}^2\,,$$
$$\langle W^3_{[1,1]} \rangle = \mathbb{I}^{(3)}_{W^3_{[1,1]}} = 3 - 10\mathfrak{q} + 9\mathfrak{q}^2 - 4\mathfrak{q}^4 + \mathfrak{q}^5\,,$$

(3.34)

and similarly for the $[2,0]$ representation we have

$$\langle W_{[2,0]} \rangle = \mathbb{I}^{(3)}_{W_{[2,0]}} = 0\,,$$
$$\langle W^2_{[2,0]} \rangle = \mathbb{I}^{(3)}_{W^2_{[2,0]}} = 0\,,$$
$$\langle W^3_{[2,0]} \rangle = \mathbb{I}^{(3)}_{W^3_{[2,0]}} = 1 - 2\mathfrak{q} + 2\mathfrak{q}^2 - 2\mathfrak{q}^4 + \mathfrak{q}^6\,.$$

(3.35)

We will show in subsection 4.2 how the oscillator representation of the Wilson lines naturally gives rise to chord counting.

### 3.2.3 Representation of Dyonic Lines for $SU(3)$

We will now discuss how to realize the dyonic lines for $SU(3)$ on the $\mathfrak{q}$-oscillators Hilbert space. It is convenient at this point to introduce variables in which the oscillator representation of the Wilson lines takes a more compact form. We define

$$x_i = (1-\mathfrak{q})^{\frac{1}{2}} a_i^\dagger\,, \qquad y_i = (1-\mathfrak{q})^{\frac{1}{2}} a_i\,, \qquad i = 1,2$$

(3.36)

with respect to which the Wilson lines become

$$W_{[1,0]} = x_1 + x_2 y_1 + y_2$$
$$W_{[0,1]} = x_2 + x_1 y_2 + y_1\,.$$

(3.37)

This will allow us to also have more compact expressions for the $\mathfrak{q}$-oscillator representation of the dyonic lines $h_{n,m}$.

## Construction of the $h_{n,m}$ Operators

We will focus on the $h_{n,m}$ lines defined in (2.20). Moreover, we will consider the case of $SU(3)$ for which we have worked out in details various relations that hold in the Schur algebra $\mathcal{A}_{\text{Schur}}$, but the same strategy can be applied with some effort for general $SU(N)$.

We proceed as follows. Consider the relations (2.32)-(2.33), which we repeat here for convenience

$$
\begin{aligned}
[h_{n,m}, W_{[1,0]}] &= (\mathfrak{q}^{\frac{1}{2}} - \mathfrak{q}^{-\frac{1}{2}})(h_{n+1,m} - h_{n,m+1}), \\
[h_{n,m}, W_{[0,1]}] &= (\mathfrak{q}^{\frac{1}{2}} - \mathfrak{q}^{-\frac{1}{2}})(h_{n,m-1} - h_{n-1,m}).
\end{aligned}
\tag{3.38}
$$

Since we have already determined the $\mathfrak{q}$-oscillators representation of the Wilson lines, we can use these equations to determine the representations of the $h_{n,m}$ for higher (or lower) values of $n,m$. To do this, we just need to fix the representation of the 't Hooft line which we propose to be $h_{0,0} = \mathfrak{q}^{-n_1-n_2-1}$. Then we determine the first few other operators for $|n| + |m| < 3$ to be

$$
\begin{aligned}
h_{0,0} &= \mathfrak{q}^{-n_1-n_2-1}, & h_{0,-1} &= y_1 \mathfrak{q}^{-n_1-n_2-\frac{1}{2}}, \\
h_{1,0} &= y_2\, \mathfrak{q}^{-n_1-n_2-\frac{1}{2}}, & h_{-1,0} &= x_2 \mathfrak{q}^{-n_1-n_2-\frac{3}{2}}, \\
h_{0,1} &= x_1\, \mathfrak{q}^{-n_1-n_2-\frac{3}{2}}, & h_{-1,-1} &= y_1 x_2 \mathfrak{q}^{-n_1-n_2-1}, \\
h_{1,1} &= x_1 y_2\, \mathfrak{q}^{-n_1-n_2-1}, & h_{1,-1} &= y_1 y_2 \mathfrak{q}^{-n_1-n_2}, \\
h_{2,0} &= y_2^2 \mathfrak{q}^{-n_1-n_2} - y_1\, \mathfrak{q}^{-n_1}, & h_{-1,1} &= x_1 x_2 \mathfrak{q}^{-n_1-n_2-2}, \\
h_{0,2} &= x_1^2 \mathfrak{q}^{-n_1-n_2-2} - x_1\, \mathfrak{q}^{-n_2-1}, & h_{-2,0} &= x_2^2 \mathfrak{q}^{-n_1-n_2-2} - x_1 \mathfrak{q}^{-n_1-1}, \\
& & h_{0,-2} &= y_1^2 \mathfrak{q}^{-n_1-n_2} - y_2 \mathfrak{q}^{-n_2},
\end{aligned}
\tag{3.39}
$$

where $x_i$, $y_i$ are those defined in (3.36). The higher or lower $h_{n,m}$ operators can be determined from these ones using the recursive relations (2.34)-(2.35), for example

$$
\begin{aligned}
h_{2,1} &= \mathfrak{q}^{\frac{1}{2}} - \mathfrak{q}^{-n_1-\frac{1}{2}} + x_1 {y_2}^2 \mathfrak{q}^{-n_1-n_2-\frac{1}{2}}, \\
h_{1,2} &= \mathfrak{q}^{\frac{1}{2}} - \mathfrak{q}^{-n_2-\frac{1}{2}} + x_1^2 y_2 \mathfrak{q}^{-n_1-n_2-\frac{3}{2}}, \\
h_{2,2} &= \mathfrak{q} x_1 - x_1 \mathfrak{q}^{-n_1-1} + \mathfrak{q} y_2 - y_2 \mathfrak{q}^{-n_2} + \mathfrak{q} x_2 y_1 + x_1^2 y_2^2 \mathfrak{q}^{-n_1-n_2-1}.
\end{aligned}
\tag{3.40}
$$

One can check explicitly that all of these operators satisfy all the relations of $\mathcal{A}_{\text{Schur}}$ that we derived in subsection 2.3, such as (2.36) and (2.37).

Moreover, we can use the map (3.11) to verify the expressions in (3.39). Let us consider $h_{2,0}$ for example. First, we note that under the map (3.11) this is mapped to

$$
h_{2,0} = \mathfrak{q}^{-1} h_{1,0} h_{0,0}^{-1} h_{1,0} - \mathfrak{q}^{-\frac{1}{2}} h_{0,-1} h_{0,0}^{-1} \frac{1}{1 - \mathfrak{q}^{-1} h_{0,1} h_{0,-1} h_{0,0}^{-2}}.
\tag{3.41}
$$

We can multiply the entire equation by the denominator, then move all factors of $h_{0,0}^{-1}$ to the right of each term in the expression, and finally multiply by $h_{0,0}^3$ from the right. This results in the following equation:

$$h_{2,0}h_{0,0}^3 - \mathfrak{q}^{-1}h_{2,0}h_{0,1}h_{0,-1}h_{0,0} = \mathfrak{q}^{-1}h_{1,0}^2 h_{0,0}^2 - \mathfrak{q}^{-2}h_{1,0}^2 h_{0,1}h_{0,-1} - \mathfrak{q}^{-\frac{1}{2}}h_{0,-1}h_{0,0}^2 . \qquad (3.42)$$

This equation can be checked in $\mathcal{A}_{\text{Schur}}$ using the $\mathfrak{q}$-Weyl algebra, provided that $\prod_{i=1}^3 V_i = 1$. Hence, the one in (3.39) is the correct $\mathfrak{q}$-oscillators representation of $h_{2,0}$.

We can perform similar computations also for all the other $h_{n,m}$ in (3.39). Moreover, as mentioned before, any other $h_{n,m}$ can be obtained from the ones in (3.39) by using the recursive relations (2.34)-(2.35). Hence, we have obtained the $\mathfrak{q}$-oscillator representation of any $SU(3)$ dyonic line $h_{n,m}$.

**Conjugation Properties of the Dyonic Lines**

We have seen in (3.24) that Hermitian conjugation induces a transformation on the $\mathfrak{q}$-oscillator representation of the Wilson lines that is the same as the action of charge conjugation of $SU(3)$, which corresponds to the $\star$ operation defined in (2.38) for $\mathcal{A}_{\text{Schur}}$. We are now going to prove that the same is true also for the dyonic lines, namely

$$h_{n,m}^\dagger = h_{-n,-m} . \qquad (3.43)$$

This can be proven by employing the recursive relations (2.34)-(2.35), and proceeding by a two-step induction. Let us consider $n, m \geq 0$ first. The basis of the induction corresponds to checking that (3.43) holds for $n, m \in \{0, 1\}$. This can be done by using explicitly the expressions in (3.39), for example

$$(h_{1,0})^\dagger = (1-\mathfrak{q})^{\frac{1}{2}}\mathfrak{q}^{-\mathfrak{n}_1-\mathfrak{n}_2-\frac{1}{2}}a_2^\dagger = (1-\mathfrak{q})^{\frac{1}{2}}a_2^\dagger \mathfrak{q}^{-\mathfrak{n}_1-\mathfrak{n}_2-\frac{3}{2}} = h_{-1,0} , \qquad (3.44)$$

where we used (3.10). In the inductive step, we assume that (3.43) is true for $h_{n,m}$ and $h_{n-1,m-1}$. Then from (2.34) we can verify that (3.43) is true also for $h_{m+1,n}$

$$
\begin{aligned}
h_{n+1,m}^\dagger &= \frac{\mathfrak{q}^{\frac{1}{2}}}{(1-\mathfrak{q})(1+\mathfrak{q})}\left(W_{[1,0]}h_{n,m}^\dagger - \mathfrak{q}h_{n,m}^\dagger W_{[1,0]} + (1-\mathfrak{q})(\mathfrak{q}^{\frac{n+m}{2}}W_{[n+m-2,0]} - h_{n-1,m-1}^\dagger)\right) \\
&= \frac{\mathfrak{q}^{\frac{1}{2}}}{(1-\mathfrak{q})(1+\mathfrak{q})}\left(W_{[1,0]}h_{-n,-m} - \mathfrak{q}h_{-n,-m}W_{[1,0]} + (1-\mathfrak{q})(\mathfrak{q}^{\frac{n+m}{2}}W_{[n+m-2,0]} - h_{-n+1,-m+1})\right) \\
&= h_{-n-1,-m}
\end{aligned}
$$

$$(3.45)$$

and likewise for $h_{n,m+1}$.

We can extend the above proof, that was valid for $n, m > 0$, to a proof for all values of $m, n$. The result for $n, m < 0$ is proven similarly, but using the recursion relations (2.35) instead. We are then left with the cases where $n$ and $m$ have opposite sign, like $n > 0$ and $m < 0$. Consider for example $h_{1,-1}$. Using (2.34) this can be re-expressed in terms of $h_{0,-1}$ and $h_{-1,-2}$. However we have already shown (3.43) for the former as the basis of the induction and for the latter because it is a case with $n, m < 0$. More generally, the validity of the result for $n, m > 0$, for $n, m < 0$ and for the basis of the induction $n, m \in \{0, 1\}$ allows us to prove it also for $n > 0$, $m < 0$ and $n < 0$, $m > 0$ via the recursion relations.

In summary, the $\star$ operation of $\mathcal{A}_{\text{Schur}}$ which acts as charge conjugation (2.38) for $SU(3)$ (and more generally $SU(N)$) corresponds to the Hermitian conjugation in the q-oscillator representation.

### 3.2.4 Identification of the Vacuum States

So far we have shown how to realize the lines of $\mathcal{A}_{\text{Schur}}$ in terms of operators built from the q-oscillators. However, we can further argue for an equivalence between the Hilbert space $\mathcal{H}_{\text{Schur}}$ introduced in subsection 2.4 and the q-oscillators Hilbert space $\mathcal{H}_{\text{osc}}$ defined in (3.5). More precisely, $\mathcal{H}_{\text{Schur}}$ is isomorphic to the *dual* q-oscillator Hilbert space

$$\mathcal{H}_{\text{Schur}} \cong \mathcal{H}_{\text{osc}}^* . \tag{3.46}$$

Let us consider for example the vacuum state of $\mathcal{H}_{\text{osc}}$, which for $N = 3$ is defined by

$$a_1 |0, 0\rangle = a_2 |0, 0\rangle = 0 . \tag{3.47}$$

On the other hand, we saw in (2.44) that the vacuum state of $\mathcal{H}_{\text{Schur}}$ satisfies

$$h_{0,1} |1^\partial\rangle = h_{-1,0} |1^\partial\rangle = 0 , \tag{3.48}$$

which using (3.11) and $h_{0,0} |1^\partial\rangle \sim |1^\partial\rangle$ implies that

$$a_1^\dagger |1^\partial\rangle = a_2^\dagger |1^\partial\rangle = 0 . \tag{3.49}$$

We then see that the vacuum of $\mathcal{H}_{\text{Schur}}$ is identified with the dual of the vacuum of $\mathcal{H}_{\text{osc}}$

$$|1^\partial\rangle = |0, 0\rangle^\dagger . \tag{3.50}$$

This relationship between the vacua is enough to obtain the identity between the Schur half-index and the VEV of the corresponding q-oscillator operator not just for the Wilson lines

as in (3.25) but for any line operator. Namely, given a line $L \in \mathcal{A}_{\text{Schur}}$ and its representation $\pi(L)$ on the $\mathfrak{q}$-oscillators Hilbert space, we have

$$\mathbb{I}_L^{(N)} = \langle 1^\partial | L | 1^\partial \rangle = \langle 0, \cdots, 0 | \pi(L) | 0, \cdots, 0 \rangle = \langle \pi(L) \rangle . \tag{3.51}$$

One useful application of this relation is to the computation of the Schur half-index with the insertion of dyonic lines, since it can be translated into a very simple one in the $\mathfrak{q}$-oscillators representation. For example for $SU(3)$

$$\begin{aligned}
\langle h_{0,0}^k \rangle &= \mathfrak{q}^{-k} , \\
\langle h_{2,1}^k \rangle &= \langle h_{1,2}^k \rangle = (\mathfrak{q}^{\frac{1}{2}} - \mathfrak{q}^{-\frac{1}{2}})^k .
\end{aligned} \tag{3.52}$$

## 3.3 Representation of the Wilson Lines for $SU(N)$

The representation of the Wilson lines in terms of $\mathfrak{q}$-oscillators that we have found for $SU(3)$ in (3.21) has a natural $SU(N)$ generalization. In this case, we make use of the $N-1$ decoupled $\mathfrak{q}$-oscillators $a_i^\dagger$, $a_i$ introduced in subsection 3.1. We also introduce the auxiliary operators

$$x_i = (1-\mathfrak{q})^{\frac{1}{2}} a_i^\dagger , \qquad y_i = (1-\mathfrak{q})^{\frac{1}{2}} a_i , \qquad i = 1, \cdots, N-1 . \tag{3.53}$$

Consider a Wilson line in one of the representations $\mathcal{R}_r$ in (2.8). We propose that in order to realize the Wilson line $W_{\mathcal{R}_r}$ in terms of the $\mathfrak{q}$-oscillators we should consider the associated character $\chi_{\mathcal{R}_r}(\vec{v})$ written in terms of $SU(N)$ parameters $v_i$ with $i = 1, \cdots, N-1$, and replace $v_i$ with $x_i$ and $v_i^{-1}$ with $y_i$. We shall refer to this as the character to oscillator map.

Let us consider for example the fundamental and the antifundamental representations of $SU(N)$. Their characters are

$$\begin{aligned}
\chi_{[1,0,\cdots,0]}(\vec{v}) &= v_1 + \sum_{i=2}^{N-1} v_i v_{i+1}^{-1} + v_{N-1}^{-1} , \\
\chi_{[0,\cdots,0,1]}(\vec{v}) &= v_1^{-1} + \sum_{i=2}^{N-1} v_i^{-1} v_{i+1} + v_{N-1} .
\end{aligned} \tag{3.54}$$

Following the above rule of the character to oscillator map, we are then led to define

$$\begin{aligned}
W_{[1,0,\cdots,0]} &= x_1 + \sum_{i=2}^{N-1} x_i y_{i-1} + y_{N-1} \\
&= (1-\mathfrak{q})^{\frac{1}{2}} \left( a_1^\dagger + (1-\mathfrak{q})^{\frac{1}{2}} \sum_{i=2}^{N-1} a_i^\dagger a_{i-1} + a_{N-1} \right) , \\
W_{[0,\cdots,0,1]} &= y_1 + \sum_{i=2}^{N-1} y_i x_{i-1} + x_{N-1} \\
&= (1-\mathfrak{q})^{\frac{1}{2}} \left( a_1 + (1-\mathfrak{q})^{\frac{1}{2}} \sum_{i=2}^{N-1} a_i a_{i-1}^\dagger + a_{N-1}^\dagger \right) .
\end{aligned} \tag{3.55}$$

As an important consistency check of our proposal, one can show that all operators constructed in this way commute

$$\left[W_{\mathcal{R}_r}, W_{\mathcal{R}_{r'}}\right] = 0 \qquad r, r' = 1, \cdots, N - 1. \tag{3.56}$$

Remember that this is an expected property since the Wilson lines in $\mathcal{A}_{\text{Schur}}$ commute, in particular the index is independent of the order of the insertions.

Another important property of these operators is that

$$\left(W_{\mathcal{R}}\right)^{\dagger} = W_{\overline{\mathcal{R}}}, \tag{3.57}$$

which one can immediately observe for the fundamental and antifundamental representations from (3.55). In other words, we see that also for $SU(N)$ the action (2.38) of charge conjugation on the Wilson lines is realized as the Hermitian conjugation in the $\mathfrak{q}$-oscillator representation.

Finally, we claim that the vacuum expectation values of the $\mathfrak{q}$-oscillator operators that realize the Wilson lines match with the corresponding Schur half-indices

$$\mathbb{I}^{(N)}_{W_{\mathcal{R}_1}^{k_1}, \cdots, W_{\mathcal{R}_{N-1}}^{k_{N-1}}} = \langle W_{\mathcal{R}_1}^{k_1} \cdots W_{\mathcal{R}_{N-1}}^{k_{N-1}} \rangle, \tag{3.58}$$

where we compactly denoted $\langle \mathcal{O} \rangle = \langle 0, \cdots, 0 | \mathcal{O} | 0, \cdots, 0 \rangle$. In the following we will consider more in detail some low rank examples and check this identity for low values of $k_r$, while in section 5 we will provide a general proof.

**Example:** $SU(4)$. We have already discussed in detail the case of $SU(3)$ in subsection 3.2, so we will now consider $SU(4)$. First, we have the fundamental and antifundamental Wilson lines (3.55)

$$\begin{aligned}
W_{[1,0,0]} &= x_1 + x_2 y_1 + x_3 y_2 + y_3 = (1 - \mathfrak{q})^{\frac{1}{2}} \left( a_1^{\dagger} + (1 - \mathfrak{q})^{\frac{1}{2}} \left( a_2^{\dagger} a_1 + a_3^{\dagger} a_2 \right) + a_3 \right), \\
W_{[0,0,1]} &= y_1 + y_2 x_1 + y_3 x_2 + x_3 = (1 - \mathfrak{q})^{\frac{1}{2}} \left( a_1 + (1 - \mathfrak{q})^{\frac{1}{2}} \left( a_2 a_1^{\dagger} + a_3 a_2^{\dagger} \right) + a_3^{\dagger} \right),
\end{aligned} \tag{3.59}$$

but we also have the antisymmetric Wilson line. The character of the antisymmetric representation of $SU(4)$ is

$$\chi^{SU(4)}_{[0,1,0]}(\vec{v}) = v_2 + v_2^{-1} + v_1 v_3^{-1} + v_1^{-1} v_3 + v_1 v_2^{-1} v_3 + v_1^{-1} v_2 v_3^{-1}, \tag{3.60}$$

which applying the character to oscillator map gives

$$\begin{aligned}
W_{[0,1,0]} &= x_2 + y_2 + x_1 y_3 + y_1 x_3 + x_1 y_2 x_3 + y_1 x_2 y_3 \\
&= (1 - \mathfrak{q})^{\frac{1}{2}} \left( a_2^{\dagger} + a_2 + (1 - \mathfrak{q})^{\frac{1}{2}} \left( a_1 a_3^{\dagger} + a_1^{\dagger} a_3 \right) + (1 - \mathfrak{q}) \left( a_1^{\dagger} a_2 a_3^{\dagger} + a_1 a_2^{\dagger} a_3 \right) \right).
\end{aligned} \tag{3.61}$$

We can explicitly verify using (3.1) that all these operators commute

$$\left[W_{[1,0,0]}, W_{[0,1,0]}\right] = \left[W_{[1,0,0]}, W_{[0,0,1]}\right] = \left[W_{[0,1,0]}, W_{[0,0,1]}\right] = 0 \tag{3.62}$$

and also that charge conjugation acts as hermitian conjugation, so in particular $W_{[0,1,0]}$ is self-adjoint.

Moreover, one can check that the index with the insertion of $k_1$ fundamental, $k_2$ antisymmetric and $k_3$ antifundamental Wilson lines matches with the VEV of the operator $W_{[1,0,0]}^{k_1} W_{[0,1,0]}^{k_2} W_{[0,0,1]}^{k_3}$ by explicit computations for low values of $k_r$. For example we find

$$\begin{aligned}
\langle W_{[0,1,0]}^2 \rangle &= \mathbb{I}^{(4)}_{W_{[0,1,0]}^2} = 1 - \mathfrak{q}\,, \\
\langle W_{[0,1,0]}^4 \rangle &= \mathbb{I}^{(4)}_{W_{[0,1,0]}^4} = (1-\mathfrak{q})^2\left(3 - \mathfrak{q} + \mathfrak{q}^2\right)\,, \\
\langle W_{[0,1,0]}^6 \rangle &= \mathbb{I}^{(4)}_{W_{[0,1,0]}^6} = (1-\mathfrak{q})^3\left(16 - 12\mathfrak{q} + 6\mathfrak{q}^2 + 5\mathfrak{q}^3\right)\,,
\end{aligned} \tag{3.63}$$

with all the odd powers vanishing instead.

**Example: $SU(5)$.** The Wilson lines in the fundamental and antifundamental representations of $SU(5)$ are as usual given by

$$\begin{aligned}
W_{[1,0,0,0]} &= (1-\mathfrak{q})^{\frac{1}{2}}\left(a_1^\dagger + (1-\mathfrak{q})^{\frac{1}{2}}\left(a_2^\dagger a_1 + a_3^\dagger a_2 + a_4^\dagger a_3\right) + a_4\right)\,, \\
W_{[0,0,0,1]} &= (1-\mathfrak{q})^{\frac{1}{2}}\left(a_4^\dagger + (1-\mathfrak{q})^{\frac{1}{2}}\left(a_3^\dagger a_4 + a_2^\dagger a_3 + a_1^\dagger a_2\right) + a_1\right)\,.
\end{aligned} \tag{3.64}$$

Similarly, we have the Wilson line in the rank-1 antisymmetric representation

$$\begin{aligned}
W_{[0,1,0,0]} &= (1-\mathfrak{q})^{\frac{1}{2}}\left(a_2^\dagger + a_3 + (1-\mathfrak{q})^{\frac{1}{2}}\left(a_3^\dagger a_1 + a_4^\dagger a_2 + a_1^\dagger a_4\right)\right) \\
&\quad + (1-\mathfrak{q})\left(a_1^\dagger a_3^\dagger a_2 + a_1^\dagger a_4^\dagger a_3 + a_2^\dagger a_1 a_4 + a_3^\dagger a_2 a_4\right) + (1-\mathfrak{q})^{\frac{3}{2}} a_2^\dagger a_4^\dagger a_1 a_3\right)
\end{aligned} \tag{3.65}$$

and the one in the rank-2 antisymmetric representation

$$\begin{aligned}
W_{[0,0,1,0]} &= (1-\mathfrak{q})^{\frac{1}{2}}\left(a_3^\dagger + a_2 + (1-\mathfrak{q})^{\frac{1}{2}}\left(a_1^\dagger a_3 + a_2^\dagger a_4 + a_4^\dagger a_1\right)\right) \\
&\quad + (1-\mathfrak{q})\left(a_2^\dagger a_1 a_3 + a_3^\dagger a_1 a_4 + a_1^\dagger a_4^\dagger a_2 + a_2^\dagger a_4^\dagger a_3\right) + (1-\mathfrak{q})^{\frac{3}{2}} a_1^\dagger a_3^\dagger a_2 a_4\right).
\end{aligned} \tag{3.66}$$

Once again we can check that all of these operators commute

$$\begin{aligned}
&\left[W_{[1,0,0,0]}, W_{[0,1,0,0]}\right] = \left[W_{[1,0,0,0]}, W_{[0,0,1,0]}\right] = \left[W_{[1,0,0,0]}, W_{[0,0,0,1]}\right] \\
&= \left[W_{[0,1,0,0]}, W_{[0,0,1,0]}\right] = \left[W_{[0,1,0,0]}, W_{[0,0,0,1]}\right] = \left[W_{[0,0,1,0]}, W_{[0,0,0,1]}\right] = 0
\end{aligned} \tag{3.67}$$

and that Hermitian conjugation acts on them exactly as charge conjugation does on the Wilson lines. Moreover, the VEVs for any of their combinations can be checked to match with the corresponding indices by explicit evaluation. For example, we find

$$\langle W_{[0,0,1,0]}^5 \rangle = \mathbb{I}^{(5)}_{W_{[0,1,0]}^2} = (1-\mathfrak{q})^4(6+\mathfrak{q}^2)\,. \tag{3.68}$$

# 4  Generalized Chord Counting

The connection between the line operator algebra $\mathcal{A}_{\text{Schur}}$ and its representation in terms of a system of $N-1$ decoupled $\mathfrak{q}$-deformed harmonic oscillators leads naturally to an interpretation in terms of (generalized) chord counting.

Indeed, it is well-known in the SYK literature that in the double scaling limit, the moments of the SYK Hamiltonian $H$ can be computed in terms of $\mathfrak{q}$-deformed harmonic oscillators by identifying $H$ with a transfer matrix $T = a^\dagger + a$. In turn the expectation values $\langle T^k \rangle$ match exactly the SYK moments $\langle \text{Tr}\, H^k \rangle_J$, where we trace over the Majorana fermions and perform a statistical averaging over the random Gaussian coupling $J$. These moments naturally have an interpretation in terms of summing all chord diagrams on a disc with a fixed number of chords $\frac{k}{2}$, where each diagram is weighted by its number of bulk intersections.

From a gauge theory point of view, $T$ is nothing other than the $SU(2)$ fundamental Wilson line in the $\mathfrak{q}$-oscillator representation (up to a normalization of $(1 - \mathfrak{q})^{\frac{1}{2}}$). This is precisely how [1] connected the double scaled SYK to chords, to $\mathfrak{q}$-oscillators, and to the Schur half-index of pure 4d $\mathcal{N} = 2$ $SU(2)$ SYM with the insertion of Wilson lines.

For $SU(3)$ we have seen that Wilson lines, dyonic lines, and their Schur half-indices have an interpretation in terms of $\mathfrak{q}$-oscillators and associated expectation values. Naturally, we are led to explore the generalization of the DSSYK chord counting that one can extract from our oscillator representation of these line operators. Indeed, we propose concrete chord counting rules that reproduces the Schur half-index with the insertion of any number of Wilson lines in any representation of $SU(3)$. We also use matter chords to give a chord realization of the dyonic lines $h_{n,m}$. More generally, we give the chord realization of any $SU(N)$ Wilson line, thus providing a third way of constructing Schur half-indices, this time as a combinatorial sum over chord configurations.

We will demonstrate our proposal first for $SU(3)$ Wilson lines, explain how it extends to $SU(N)$ Wilson lines, and then explain how we use matter chords to realize dyonic lines. To round off the section, we discuss how certain matter chords are also related to 3d $\mathcal{N} = 2$ domain walls inside the 4d $\mathcal{N} = 2$ $SU(N)$ SYM. All chord diagrams in this section are to be read in a clockwise manner.

## 4.1  Review: DSSYK and Chord Counting

The SYK model is a one-dimensional quantum system with random $p$-body interactions, drawn from a sample of $N$ Majorana fermions obeying $\{\psi_i, \psi_j\} = 2\delta_{ij}$. The Hamiltonian for

this system is given by

$$H = i^{p/2} \sum_{i_1 < \cdots < i_p} J_{i_1 \cdots i_p} \psi_{i_1} \cdots \psi_{i_p} = i^{p/2} \sum_I J_I \psi_I \,, \tag{4.1}$$

where the random couplings $J_I$ are drawn from a Gaussian ensemble

$$\langle J_I \rangle_J = 0 \,, \qquad \langle J_I J_K \rangle_J = \frac{N}{2p^2} \binom{N}{p}^{-1} \delta_{IK} \,. \tag{4.2}$$

The double scaling limit of the SYK model is the one in which we take

$$p, N \to \infty \,, \qquad \lambda = \frac{p^2}{N} \quad \text{fixed} \,. \tag{4.3}$$

In this double scaling limit the model can be solved exactly by recasting it into the language of chords [19]. The chords represent Wick contractions between the random couplings $J_I$ in the expansion of $\langle \operatorname{Tr} H^k \rangle_J$, and represent the connection between two distinct points on the thermal circle. The weight of a bulk intersection point is the average factor one gets from commuting the fermions within the trace. This factor is Poisson distributed, with average value $e^{-\lambda} = \mathfrak{q}$. One then sums over all possible chord diagrams with $k/2$ chords, and weights each diagram by its number of intersections, yielding the partition function

$$Z_{\text{DSSYK}}(\beta) = \langle \operatorname{Tr} e^{-\beta H} \rangle_J = \sum_{k=0}^{\infty} \frac{(-\beta)^k}{k!} \langle \operatorname{Tr} H^k \rangle_J = \sum_{k=0}^{\infty} \frac{(-\beta)^k}{k!} \sum_{\text{diagrams with } k/2 \text{ chords}} \mathfrak{q}^{\# \text{ intersections}} \,. \tag{4.4}$$

The counting of chord diagrams can be recast in the language of $\mathfrak{q}$-deformed harmonic oscillators, similar to those we discussed in section 3 but for the special case of $N = 2$. The idea is to cut open the diagram at a generic point, which is then declared to have no open chords. The $\mathfrak{q}$-oscillator Hilbert space is interpreted as the Hilbert space of open chords. A chord diagram with $k$ external vertices is then constructed by starting from the vacuum $|0\rangle$, acting on it $k$ times with the transfer matrix $T = a^\dagger + a$, and projecting back to the vacuum $\langle 0|$. From this perspective, the $k$-th moment of the SYK Hamiltonian can be computed as a VEV of the transfer matrix

$$\langle \operatorname{Tr} H^k \rangle_J = \langle T^k \rangle \,. \tag{4.5}$$

It is possible to re-express the VEVs of the transfer matrix as an integral by diagonalizing it [20, 21] (we will see this computation for our $SU(N)$ generalization in section 5). One finds in this case that the eigenfunctions of the transfer matrix are given by the $\mathfrak{q}$-Hermite polynomials, and the partition function of the theory is

$$Z_{\text{DSSYK}}(\beta) = \int_0^{\pi} \frac{d\theta}{\pi} e^{\frac{-2\beta \cos(\theta)}{\sqrt{\lambda(1-\mathfrak{q})}}} (\mathfrak{q}; \mathfrak{q})_\infty (e^{2i\theta}; \mathfrak{q})_\infty (e^{-2i\theta}; \mathfrak{q})_\infty \,, \tag{4.6}$$

where the integral appearing in the above expression is a generating function for the 4d $\mathcal{N} = 2$ $SU(2)$ SYM Schur half-index with the insertion of fundamental Wilson lines [1], which in our notation corresponds to $W_{[1]}$. Equivalently, the integral expression for the moment $\langle \mathrm{Tr}\, H^k \rangle_J$ coincides with the Schur half-index (2.6) for $N = 2$ with the insertion of $k$ fundamental Wilson lines (up to some overall $(1 - \mathfrak{q})^{\frac{k}{2}}$ normalization).

As elluded to earlier, the moments of the Hamiltonian $\langle \mathrm{Tr}\, H^k \rangle_J$ coincide (up to an overall $(1 - \mathfrak{q})^{\frac{k}{2}}$ normalization) with the powers of a transfer matrix $\langle T_{[1]}^k \rangle$, where we define $T_{[1]} = (1 - \mathfrak{q})^{\frac{1}{2}}(a + a^\dagger)$.[10] It may seem pathological to introduce the overall normalization into the transfer matrix $T_{[1]}$, since this means it no longer matches exactly with the DSSYK moments, but this has the advantage that the VEVs $\langle T_{[1]}^k \rangle$ now match exactly with the index with the insertion of $k$ fundamental Wilson lines. That is, with this choice of normalization

$$(1 - \mathfrak{q})^{-\frac{k}{2}} \langle \mathrm{Tr}\, H^k \rangle_J = \langle T_{[1]}^k \rangle = \mathbb{I}_{W_{[1]}^k}^{(2)} \,. \tag{4.7}$$

Moreover, under this correspondence, we see that the transfer matrix is exactly identified with the oscillator representation of the fundamental Wilson line $W_{[1]}$. From the transfer matrix $T_{[1]}$, we extract chord rules that allow us to recast the computation of $\langle T_{[1]}^k \rangle$ in terms of a combinatorial sum over weighted chord diagrams.

Let us briefly recap how this works in our new normalization. Note that here we use the terminology *bulk weight* to refer to the factor of $\mathfrak{q}^{\# \text{ intersections}}$ of a chord diagram, and the term *boundary weight* to refer to the factors that one picks up on the boundary each time we create or annihilate a chord. Boundary weights are not a feature of traditional DSSYK chord counting, but they necessarily arise when one considers our higher rank chord rules. We associate the insertion of $(1 - \mathfrak{q})^{\frac{1}{2}} a^\dagger$ in the VEV with creating a chord on the thermal circle with a boundary weight of $(1 - \mathfrak{q})^{\frac{1}{2}}$, and the insertion of $(1 - \mathfrak{q})^{\frac{1}{2}} a$ with annihilating a chord on the thermal circle with boundary weight $(1 - \mathfrak{q})^{\frac{1}{2}}$.

For example, we see from figure 4 that evaluating $\langle T_{[1]}^4 \rangle$ using the chord diagrams with the explicit inclusion of boundary weights gives us

$$\langle T_{[1]}^4 \rangle = (1 - \mathfrak{q})^2 (2 + \mathfrak{q}) \,. \tag{4.8}$$

This matches exactly the DSSYK computation of $\langle \mathrm{Tr}\, H^4 \rangle_J$, up to the overall $(1 - \mathfrak{q})^2$ factor, which we stress is included here so that our computations match exactly with the index $\mathbb{I}_{W_{[1]}^4}^{(2)}$. It is precisely this correspondence between terms in the transfer matrix and weighted chord

---

[10]We use the subscript $[1]$ to align with our notation that this is the transfer matrix associated to the fundamental Wilson line of $SU(2)$. This is to be contrasted to the standard DSSYK literature in which the transfer matrix is just called $T$. We also normalize this new transfer matrix $T_{[1]}$ so that it coincides with the oscillator representation of the Wilson line $W_{[1]}$.

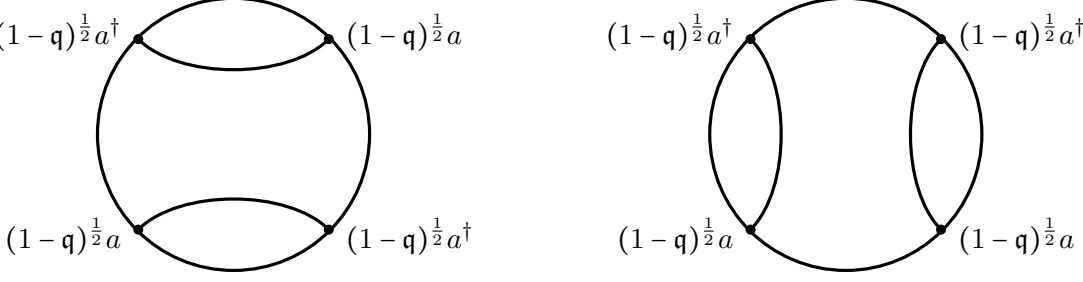

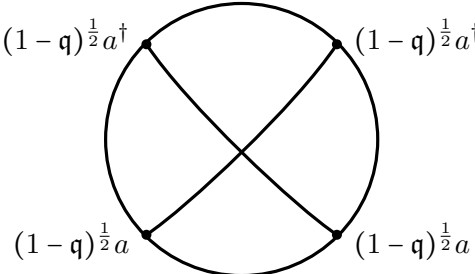

Figure 4: The three chord diagrams contributing at order $\langle T_{[1]}^4 \rangle$ in the standard SYK chord diagram expansion. The first diagram arises from evaluating $(1-\mathfrak{q})^2 \langle aa^\dagger aa^\dagger \rangle$, and the second and third diagrams come from evaluating $(1-\mathfrak{q})^2 \langle a^2 (a^\dagger)^2 \rangle$. We included here explicitly the vertex boundary weights on the thermal circle. The first two diagrams have bulk weight $\mathfrak{q}^0 = 1$ since they have no bulk intersections, and the third diagram has bulk weight $\mathfrak{q}$, since it has one bulk intersection.

rules on the thermal circle that will allow us to interpret for generic $SU(N)$ our $\mathfrak{q}$-oscillators representation of line operators in terms of generalized chords.

In analogy with the DSSYK presentation, in this section we will call the $\mathfrak{q}$-oscillator representation of the Wilson lines *transfer matrices*, and implement the notation $T_\mathcal{R} \equiv W_\mathcal{R}$, where $T_\mathcal{R}$ is the transfer matrix associated to a Wilson line in a representation $\mathcal{R}$ of $SU(N)$. We will also refer to the DSSYK chords as bivalent chords, anticipating an $N$-valency generalization.

Finally, we will present two very insightful limits of the Schur half-index, $\mathfrak{q} \to 1$ and $\mathfrak{q} \to 0$, which will allow us to construct the chord counting problem associated to the fundamental Wilson lines for higher $SU(N)$. The purpose of this is so that one can in principle derive the chord rules purely from the limits of the index, without *a priori* knowledge of the associated transfer matrix. The two limits we present are such that the Schur half-index becomes a generating function for two different associated combinatorial problems. For the $SU(2)$ index, these limits are the given by the following.

The $\mathfrak{q} \to 0$ limit of the Schur half-index with the insertion of $2k$ fundamental Wilson lines counts the number of walks of length $2k$ in $\mathbb{N}_0$ starting and ending at the origin that one can

take given the allowed steps $\{(1),(-1)\}$ [72].[11] If one takes this limit in the language of chords, all chord diagrams with bulk intersections are sent to 0, leaving only the non-intersecting chord diagrams. One can go even further and reconstruct all non-intersecting chord diagrams from the walking problem, by assigning a $(1)$ step to opening a chord, and a $(-1)$ step to closing the first available chord.

The $\mathfrak{q} \to 1$ limit of the Schur half-index with the insertion of $2k$ fundamental Wilson lines counts, up to the $(1 - \mathfrak{q})^k$ prefactor, the number of distinct chord diagrams we can draw with $k$ chords.[12] This is just because in this limit each chord diagram contributes with bulk intersection weight 1.

To summarize, the $\mathfrak{q} \to 1$ limit tells us the total number of chord diagrams, while the $\mathfrak{q} \to 0$ limit allows us to distinguish between those that have trivial and non-trivial bulk intersection. Investigating these limits is what will allow us to figure out what generalized chord counting problem is associated to the higher $SU(N)$ Schur half-index.

Note that our ability to write the DSSYK partition function in the closed form integral expression (4.6) is due to the fact that the SYK model becomes integrable in the double scaling limit. In this limit, we recover $\mathfrak{q}$-deformed Liouville quantum mechanics, and in the $\mathfrak{q} \to 1$ limit of the DSSYK (commonly referred to as the triple scaling limit) we recover standard Liouville quantum mechanics [22].

Since the SYK model has a clear gravitational interpretation, the construction of generalized chord counting problems that are solved by the Schur half-index of $SU(N)$ SYM suggests a (higher spin) gravitational interpretation of the correspondence. The correct higher rank SYK model remains thus far elusive. One expects a random many body interaction analogous to the SYK model, with the feature that in the double scaling limit, the model recovers the relativistic open Toda chain, and in the triple scaling limit reduces to simply the classical open Toda chain (we will show how the Toda chain arises in the study of $SU(N)$ Schur half-indices in section 5.1). In the following, we discuss the generalized chord counting problems for higher rank $SU(N)$ and leave the question of finding a generalization of the SYK model that has a limit in which it is solved by these chord diagrams for future investigation.

## 4.2 Generalized Chord Rules for $SU(3)$ Line Operators

For clarity, let us first focus on $SU(3)$ line operators and propose the associated chord counting. We start with the fundamental Wilson line and then extend to other representations.

---

[11]These are also the 2-dimensional Catalan numbers.

[12]This is also the total number of partitions of the set $\{1, 2, \ldots 2k\}$ into sets of size 2. Since the partition is into sets of size 2, we see that each chord in a diagram corresponds to one of these two-element sets within a given partition, in that it connects two points on the thermal circle.

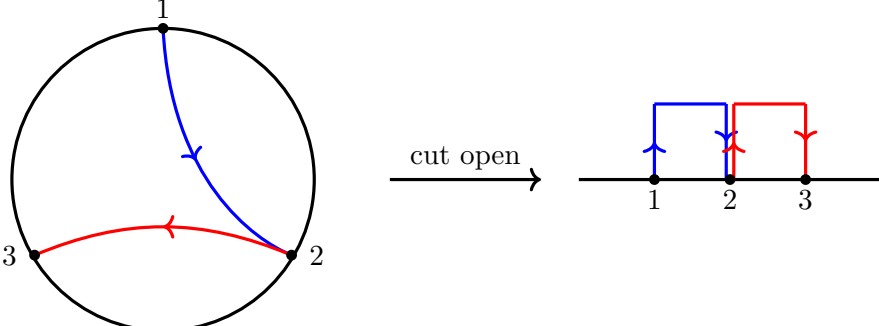

Figure 5: The minimal diagram associated to the fundamental representation of $SU(3)$ has three external vertices joined simultaneously with two types of chord, which we color in blue and red. The diagram also has an intrinsic orientation encoded in the arrows. On the left we represent the closed diagram, while on the right we represent the opened version.

**Fundamental Wilson Lines**

We can associate a counting problem of generalized chord diagrams to the $SU(3)$ fundamental Wilson line transfer matrix

$$T_{[1,0]} = (1-\mathfrak{q})^{\frac{1}{2}} \left( a_1^\dagger + (1-\mathfrak{q})^{\frac{1}{2}} a_1 a_2^\dagger + a_2 \right), \tag{4.9}$$

which coincides with the $\mathfrak{q}$-oscillators representation of the fundamental Wilson line. The counting problem of chord diagrams with $k$ vertices that we are going to discuss will thus reproduce the Schur half-index with the insertion of $k$ fundamental Wilson lines.

The chord diagrams are constructed as follows. Each diagram has $k$ external vertices, which should be joined in triples using two different types of chords that we will distinguish using blue and red colors (see figure 5). At each of the three connected points on the thermal circle, one of the following can occur:

1. a blue chord is created with boundary weight $(1-\mathfrak{q})^{\frac{1}{2}}$;

2. a blue chord is destroyed and a red chord is created with boundary weight $(1-\mathfrak{q})$;

3. a red chord is destroyed with boundary weight $(1-\mathfrak{q})^{\frac{1}{2}}$.

Notice that a natural ordering of the three points is required, since a chord cannot be destroyed before being created. Hence, point 1. has to preceed point 2. which has to preceed point 3. In figure 5 we represent this ordering following the clockwise direction, and we stress the creation/annihilation of a chord with an outgoing/ingoing arrow.

On the right of figure 5 we also represent the opened version of the diagram. A chord diagram can be cut open only at a point where there is not an open chord. Hence, the

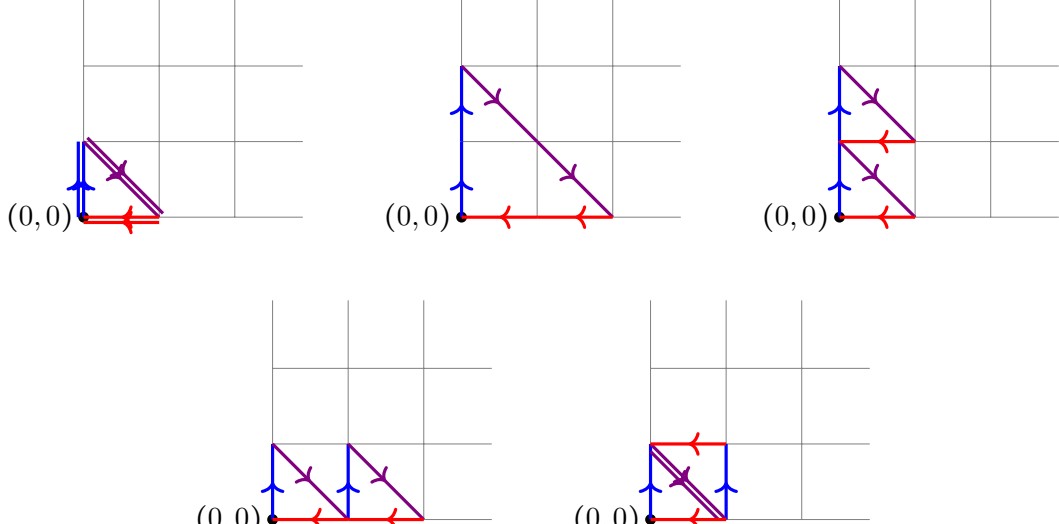

Figure 6: The five walks of length 6 starting and ending at the origin in $\mathbb{N}_0^2$. The diagrams depict the walks **123123**, **112233**, **112323**, **121233**, and **121323**. Blue is used for a step **1**, violet is used for **2**, and red is used for **3**.

diagram in figure 5 can only be cut between points 3. and 1. This has to be contrasted with the $SU(2)$ case corresponding to the ordinary DSSYK chord counting, where we can cut open anywhere and just declare that at that point there are no open chords.

One can check that the number of inequivalent diagrams with $k$ external vertices one can draw is exactly equal (up to an overall $(1-\mathfrak{q})^{\frac{2}{3}k}$) to the Schur half-index of $SU(3)$ SYM with the insertion of $k$ fundamental Wilson lines in the $\mathfrak{q} \to 1$ limit, confirming that this is the right counting problem we are after. Explicitly for low values of $k$ we find[13]

$$
\begin{aligned}
\lim_{\mathfrak{q}\to 1}(1-\mathfrak{q})^{-2}\mathbb{I}^{(3)}_{W^3_{[1,0]}} &= 1\,, \\
\lim_{\mathfrak{q}\to 1}(1-\mathfrak{q})^{-4}\mathbb{I}^{(3)}_{W^6_{[1,0]}} &= 10\,, \\
\lim_{\mathfrak{q}\to 1}(1-\mathfrak{q})^{-6}\mathbb{I}^{(3)}_{W^9_{[1,0]}} &= 280\,, \\
\lim_{\mathfrak{q}\to 1}(1-\mathfrak{q})^{-8}\mathbb{I}^{(3)}_{W^{12}_{[1,0]}} &= 15400\,,
\end{aligned}
\tag{4.10}
$$

while the index vanishes for $k \neq 0 \bmod 3$.

To reproduce the $\mathfrak{q}$-refined index we should find suitable intersection rules that allow us to weight each diagram in the right way. To understand where these chord intersection rules come from, it is useful to consult the second limit of the Schur half-index given by $\mathfrak{q} \to 0$. for

---

[13]These are also the number of partitions of the set $\{1, 2, \ldots, 3k\}$ into sets of size 3 [73].

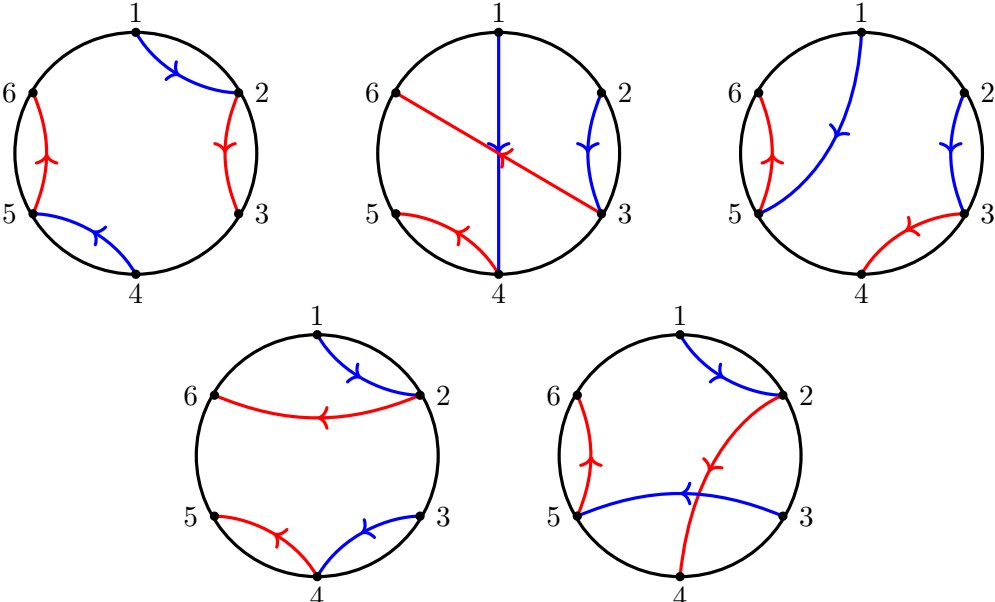

Figure 7: The five chord diagrams with trivial bulk contribution surviving the $\mathfrak{q} \to 0$ limit, corresponding to the five walks in figure 6.

low $k$ we get

$$
\begin{aligned}
\lim_{\mathfrak{q}\to 0}(1-\mathfrak{q})^{-2}\mathbb{I}^{(3)}_{W^3_{[1,0]}} &= 1\,, \\
\lim_{\mathfrak{q}\to 0}(1-\mathfrak{q})^{-4}\mathbb{I}^{(3)}_{W^6_{[1,0]}} &= 5\,, \\
\lim_{\mathfrak{q}\to 0}(1-\mathfrak{q})^{-6}\mathbb{I}^{(3)}_{W^9_{[1,0]}} &= 42\,, \\
\lim_{\mathfrak{q}\to 0}(1-\mathfrak{q})^{-8}\mathbb{I}^{(3)}_{W^{12}_{[1,0]}} &= 462\,.
\end{aligned}
\tag{4.11}
$$

In this limit, the Schur half-index with the insertion of $k = 3n$ fundamental Wilson lines coincides with the number of walks of length $3n$ in $\mathbb{N}_0^2$ starting and ending at the origin $(0,0)$ using the allowed steps[14]

$$
\{\mathbf{1} = (-1,0),\, \mathbf{2} = (0,1),\, \mathbf{3} = (1,-1)\}\,.
\tag{4.12}
$$

In analogy to the $SU(2)$ case of DSSYK chords, we assign the step $(0,1)$ to opening a blue chord, the step $(1,-1)$ to closing a blue chord and opening a red chord, and the step $(-1,0)$ to closing a red chord. For $k = 3n = 6$, there are five possible walks, shown in figure 6.

Using the rules outlined above, we can associate each of these five walks to a chord diagram, which we depict in figure 7. Observe that of all the diagrams, there are three obviously non-intersecting ones, given by the first, third, and fourth. However, there are also two diagrams

---

[14]These are also the 3-dimensional Catalan numbers or equivalently the number of standard tableaux of the form $(n,n,n)$.

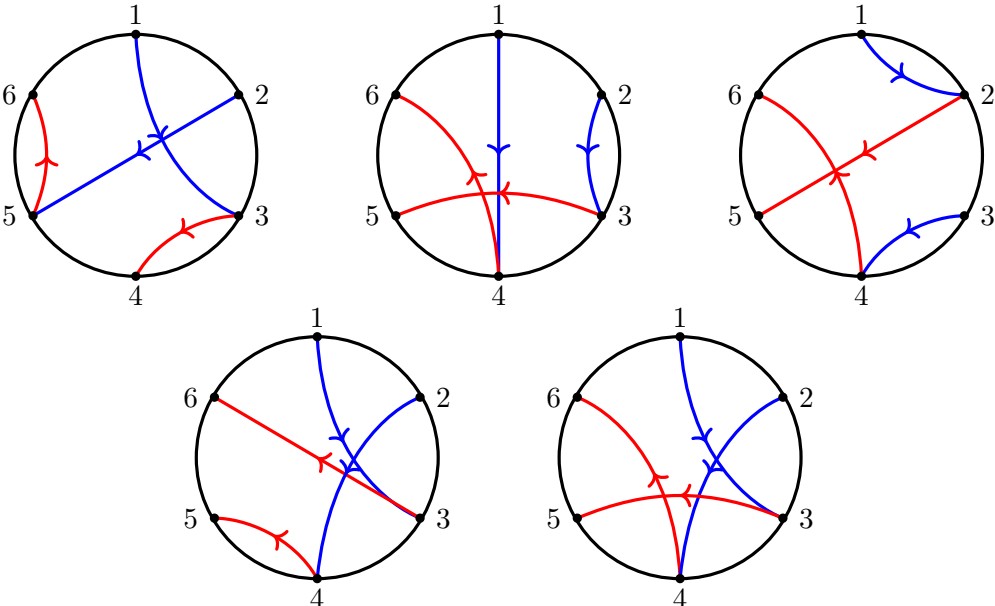

Figure 8: The five chord diagrams with non-trivial bulk intersection weight. The first four diagrams are weighted with weight $(1-\mathfrak{q})^4\,\mathfrak{q}$, and the final diagram has weight $(1-\mathfrak{q})^4\,\mathfrak{q}^2$.

which have a blue-red bulk intersection. This means that in order for the contribution of these two diagrams to survive in the $\mathfrak{q} \to 0$ limit, they must contribute with a trivial weight $(1-\mathfrak{q})^4\,\mathfrak{q}^0$. The factor of $(1-\mathfrak{q})^4$ is due to the fact that all of these diagrams have six boundary vertices, each of which has an associated boundary weight by our general rules. Note that in the limit $\mathfrak{q} \to 0$ this factor is trivialized, so in order to detect it we need to consult the other limit $\mathfrak{q} \to 1$.

The $\mathfrak{q} \to 1$ limit of the Schur index (4.10) tells us there should be 10 total chord diagrams, with four of them weighted by a factor of $(1-\mathfrak{q})^4\,\mathfrak{q}$, and one of them weighted by a factor of $(1-\mathfrak{q})^4\,\mathfrak{q}^2$. To determine these, we consider the remaining diagrams with $k = 3n = 6$ external vertices that we can construct using the rule to connect vertices in figure 5, that are not in figure 7. These are depicted in figure 8. By observation, we see that the first four diagrams all come with either one red-red intersection or one blue-blue intersection, with some number of trivially contributing blue-red bulk intersections. This suggests that red-red and blue-blue intersections should be counted with weight $\mathfrak{q}$, so that these four diagrams reproduce the index contribution $4\mathfrak{q}$. Finally, the fifth diagram is our maximally intersecting one, consisting of one of each type of bulk intersection, and so is assigned the bulk weight $\mathfrak{q}^{1+0+1} = \mathfrak{q}^2$. Summing all ten diagrams, we find the total contribution

$$(1-\mathfrak{q})^4\left(5 + 4\mathfrak{q} + \mathfrak{q}^2\right),\qquad(4.13)$$

which matches exactly the $SU(3)$ Schur half-index with the insertion of two fundamental Wilson lines (3.26).

Summarizing, we get the following set of intersection rules:

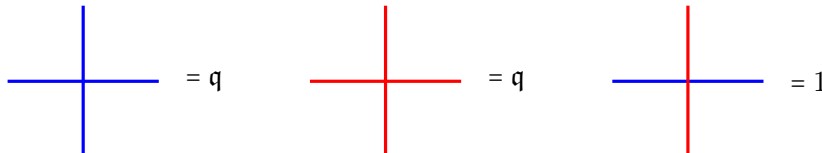

Figure 9: The bulk intersection rules for chord diagrams in $SU(3)$.

One can check explicitly that these rules reproduce the $SU(3)$ Schur indices for various values of $k$ (see (3.26) for the cases $k = 3, 6$).

In order to later make connection with the Wilson lines defined in section 3, it is useful to consider the opened chord diagrams. In the $SU(2)$ case of DSSYK chords, one marks any point on the thermal circle in between vertices and slices open the chord diagram at that point. The thermal circle can then be unfolded, with the convention that at the cutting point one declares to have no open chord and then moving to the right of the linearized diagram open chords decrease in height from left to right. As mentioned previously, for the $SU(3)$ case there is a restriction in where one can slice open the chord diagram.

Consider for example the two slicings depicted in figure 10 of the maximally intersecting $(1-q)^4 q^2$ chord diagram of figure 8. In the first, we slice the diagram between the nodes 6 and 1, and in the second we slice between the nodes 5 and 6. In the first slicing, all of the chords appear with the trivalent structure we introduced at the beginning: a blue chord opens, it closes to create a red chord, and then the red chord closes. In particular, all the arrows are directed from left to right, following the correct evolution of the diagram. Hence we say this is an *allowed* slicing. The second slicing is a *disallowed* slicing, since one of the red chords is incorrectly oriented from right to left. The issue with this slicing is that we cut open at a

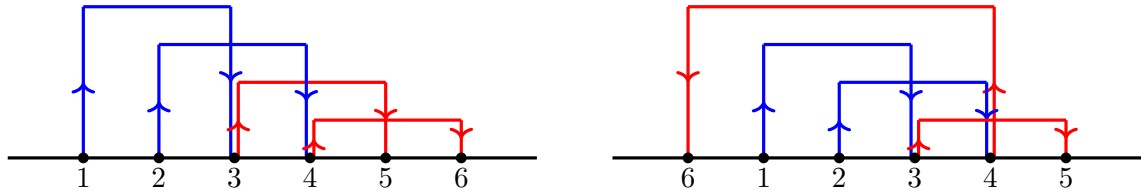

Figure 10: Two different slicings of the maximally intersecting chord diagram for $3k = 6$. The diagram on the left is an *allowed* slicing, where the circle has been cut between nodes 6 and 1. The diagram on the right is a *disallowed* slicing, where the circle has been cut between nodes 5 and 6.

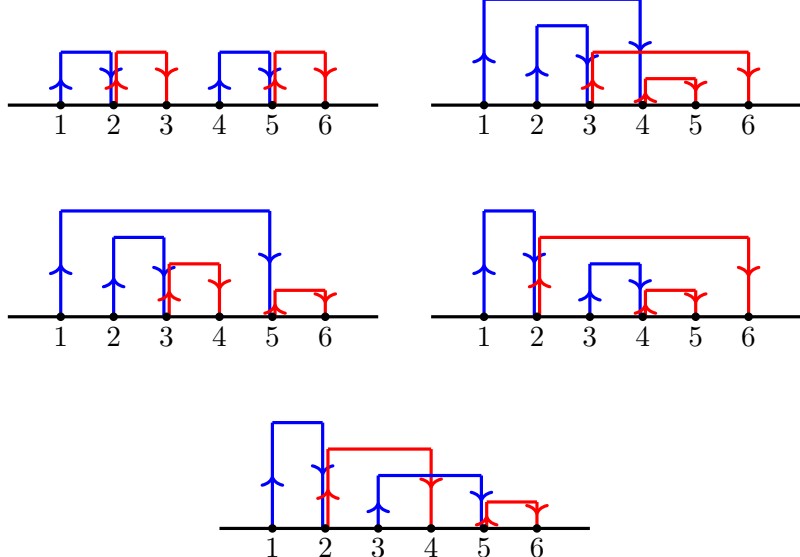

Figure 11: The five chord diagrams with intersection weight $\mathfrak{q}^0$ as open chord diagrams. The slice was taken between nodes 6 and 1.

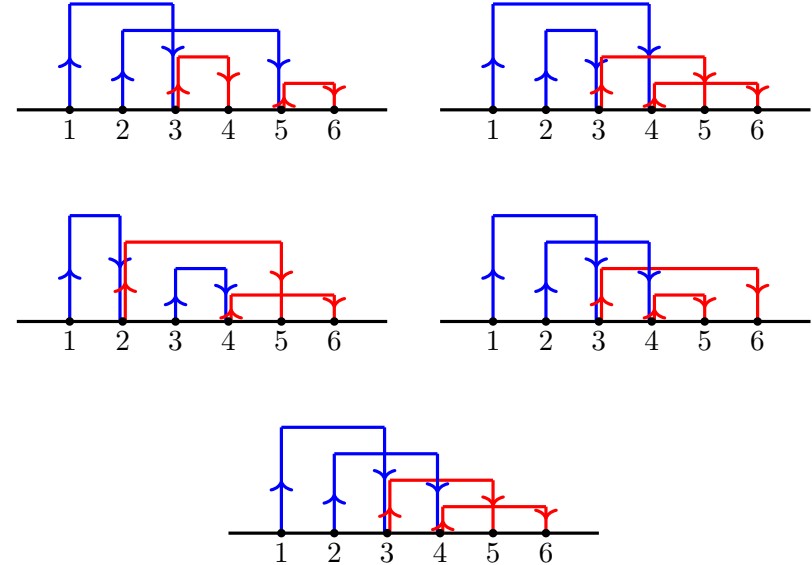

Figure 12: The five chord diagrams with non-trivial intersection weight, sliced open between nodes 6 and 1. The first four diagrams come with intersection weight $\mathfrak{q}$, and the final one comes with intersection weight $\mathfrak{q}^2$.

point where a red chord is already opened. This gives us a clear notion of when a slicing of a chord diagram is allowed

> *A chord diagram can only be sliced open where the number of open chords*
>
> *at the point of slicing is exactly 0,*

or equivalently

> *A chord diagram can only be sliced open at a point such that all chords*
> *have orientation from left to right in the opened diagram.*

With this convention the ten open chord diagrams corresponding to the closed diagrams of figures 7 and 8 are the ones depicted in figures 11 and 12 respectively.

## Hilbert Space Interpretation

We now wish to give a Hilbert space interpretation of the counting problem we have just defined, thus connecting it concretely to transfer matrices. As in the ordinary DSSYK model, we view an open chord diagram with $k$ vertices as generated by applying $k$ times the operator $T_{[1]}$ to the vacuum state $|0\rangle$ and then contracting with $\langle 0|$. At intermediate steps, we will have a state labeled by the set of chords that are opened. However, an interesting feature of the analogous problem for $SU(3)$ chords is that such a state is determined only by the number of blue chords $n_1$ and of red chords $n_2$, while their ordering does not matter. This is because swapping the order of blue and red chords might introduce extra intersections in a given diagram, however these are always of blue-red type that give a trivial contribution according to the rules in figure 9. This gives a tensor product Hilbert space with states $|n_1, n_2\rangle$ which we can choose to represent with all the blue chords on top of the red ones

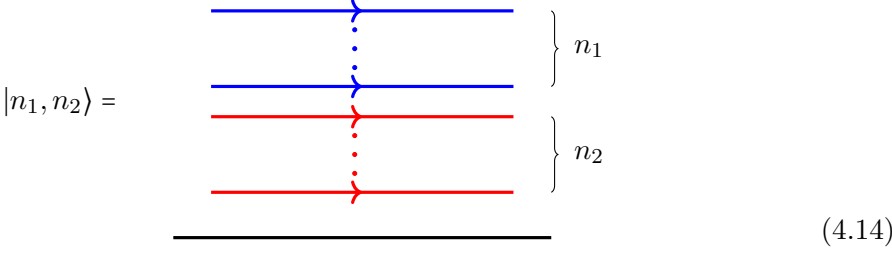

$$|n_1, n_2\rangle = \tag{4.14}$$

This Hilbert space corresponds to the one (3.5) of the decoupled $\mathfrak{q}$-oscillators.

Let us now consider the action of the transfer matrix that makes us evolve from such a state associated with a generic external point of a chord diagram to the next one. Given the rules for constructing a generalized chord diagram that we described in the previous section, there are three possibilities:

1. A blue chord might be created, so that we obtain the state $(1-\mathfrak{q})^{\frac{1}{2}}|n_1 + 1, n_2\rangle$.

2. A blue chord might be closed and a red chord created, so that we obtain the state $(1-\mathfrak{q})|n_1 - 1, n_2 + 1\rangle$. If we choose to close the $i$-th blue chord, then we will have $i-1$ blue-blue intersections and $n_2$ blue-red intersections, producing a factor of $\mathfrak{q}^{i-1}$.

3. A red chord might be closed, so that we obtain the state $(1-\mathfrak{q})^{\frac{1}{2}}|n_1, n_2-1\rangle$. If we choose to close the $j$-th red chord, then we will have $j-1$ red-red intersections. Hence, an additional factor of $\mathfrak{q}^{j-1}$ will be produced.

Summarizing, we find that the evolution in the open chords Hilbert space can be realized precisely with the fundamental Wilson line operator

$$
\begin{aligned}
T_{[1,0]}|n_1, n_2\rangle &= (1-\mathfrak{q})^{\frac{1}{2}}|n_1+1, n_2\rangle + (1-\mathfrak{q})\left(\sum_{i=1}^{n_1}\mathfrak{q}^{i-1}\right)|n_1-1, n_2+1\rangle + (1-\mathfrak{q})^{\frac{1}{2}}\left(\sum_{j=1}^{n_2}\mathfrak{q}^{j-1}\right)|n_1, n_2-1\rangle \\
&= (1-\mathfrak{q})^{\frac{1}{2}}|n_1+1, n_2\rangle + (1-\mathfrak{q})[n_1]_\mathfrak{q}|n_1-1, n_2+1\rangle + (1-\mathfrak{q})^{\frac{1}{2}}[n_2]_\mathfrak{q}|n_1, n_2-1\rangle \\
&= (1-\mathfrak{q})^{\frac{1}{2}}\left(a_1^\dagger + (1-\mathfrak{q})^{\frac{1}{2}}a_1 a_2^\dagger + a_2\right)|n_1, n_2\rangle,
\end{aligned}
\tag{4.15}
$$

where we use the creation and annihilation operators of two decoupled $\mathfrak{q}$-deformed harmonic oscillators

$$
\begin{aligned}
a_1^\dagger|n_1, n_2\rangle &= |n_1+1, n_2\rangle, & a_1|n_1, n_2\rangle &= [n_1]_\mathfrak{q}|n_1-1, n_2\rangle, \\
a_2^\dagger|n_1, n_2\rangle &= |n_1, n_2+1\rangle, & a_2|n_1, n_2\rangle &= [n_2]_\mathfrak{q}|n_1, n_2-1\rangle,
\end{aligned}
\tag{4.16}
$$

which satisfy the algebra

$$
[a_1, a_1^\dagger]_\mathfrak{q} = [a_2, a_2^\dagger]_\mathfrak{q} = 1, \qquad [a_1, a_2]_1 = [a_1, a_2^\dagger]_1 = 0.
\tag{4.17}
$$

We thus recover the system of two decoupled $\mathfrak{q}$-deformed oscillators (3.1) and the action of the operator $T_{[1,0]}$ associated to the fundamental Wilson line we introduced in (3.21).

We can intuitively think of the commutators (4.17) as associated with the intersection rules in figure 9 as follows. To a blue line we associate the operators $a_1^\dagger$, $a_1$ and to a red line we associate the operators $a_2^\dagger$, $a_2$. So from the intersection rules in figure 9 we get that

- blue-blue intersection implies $[a_1, a_1^\dagger]_\mathfrak{q} = 1$,

- red-red intersection implies $[a_2, a_2^\dagger]_\mathfrak{q} = 1$

- red-blue intersection implies $[a_1, a_2]_1 = [a_1^\dagger, a_2]_1 = 0$.

In this way we recover the $\mathfrak{q}$-oscillator algebra (4.17).

**Other Representations**

More bizarre chord counting problems arise from Wilson lines in other representations. The logic to take here is that we should think of the associated transfer matrices $T_\mathcal{R}$ as defining certain vertex rules. Each term in $T_\mathcal{R}$ labels a process that can happen on the thermal circle. To construct the chord diagrams with a fixed number of chords, one should apply all consistent

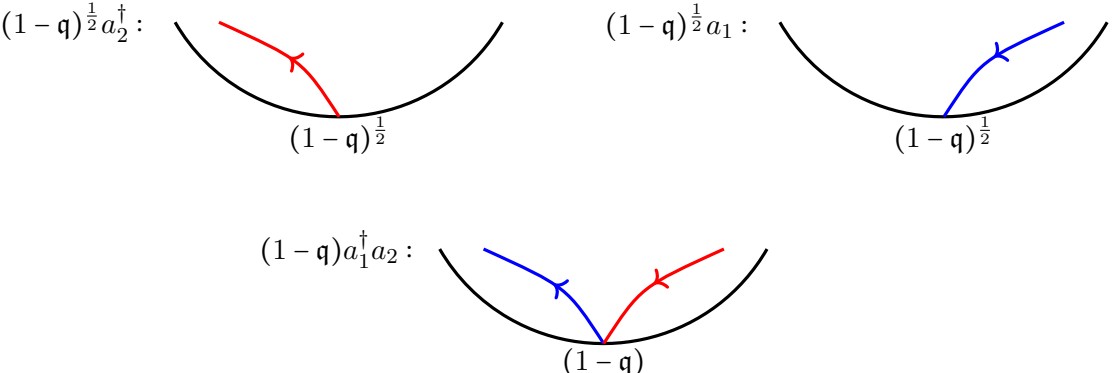

Figure 13: Chord vertex rules for the antifundamental transfer matrix $T_{[0,1]}$.

vertex rules to draw all possible chord diagrams, then sum over them, weighted appropriately by the boundary vertex factors and the number of chord intersections in the bulk, where the latter are the same as in figure 9 since the are only dictated by the $\mathfrak{q}$-oscillators algebra. Despite the chord counting problems for the fundamental and antifundamental representations are equivalent (up to an exchange of color) it is instructive to first demonstrate how the antifundamental works before going to higher weight representations.

For the antifundamental representation of $SU(3)$, we still have the two decoupled $\mathfrak{q}$-oscillators $a_i^\dagger$, $a_i$ for $i = 1, 2$, and the associated transfer matrix is given by

$$T_{[0,1]} = (1 - \mathfrak{q})^{\frac{1}{2}} \left( a_1 + (1 - \mathfrak{q})^{\frac{1}{2}} a_1^\dagger a_2 + a_2^\dagger \right). \tag{4.18}$$

In this case all the chords look as in the previous example of the fundamental Wilson line $T_{[1,0]}$, but with the colors of red and blue interchanged. Instead of drawing the entire chord diagram, we just write down the vertex rules. The antifundamental Wilson line has three terms, each of which has an associated external vertex that we depict in figure 13. Each diagram is to be read from right to left, so that newly created chords point towards the left, and annihilated chords enter from the right and are terminated on the circle boundary. We see that the $(1-\mathfrak{q})^{\frac{1}{2}} a_2^\dagger$ operator creates a red chord with boundary weight $(1-\mathfrak{q})^{\frac{1}{2}}$, $(1-\mathfrak{q})^{\frac{1}{2}} a_1$ annihilates a blue chord with boundary weight $(1-\mathfrak{q})^{\frac{1}{2}}$, and $(1-\mathfrak{q}) a_1^\dagger a_2$ annihilates a red chord and creates a blue chord with boundary weight $(1-\mathfrak{q})$. Each time one of these rules is applied, there must be a corresponding factor of $T_{[0,1]}$ in the vacuum expectation value, so we see that the first non-trivial non-zero VEV is $\langle T_{[0,1]}^3 \rangle$, where we have created a red chord with weight $(1-\mathfrak{q})^{\frac{1}{2}}$, annihilated it for a factor of $(1-\mathfrak{q})$ whilst creating a blue chord, and subsequently annihilated that blue chord with weight $(1-\mathfrak{q})^{\frac{1}{2}}$. Computing $\langle T_{[0,1]}^k \rangle$ is then a matter of fixing the number of nodes $k$ on the circle, and summing over all possible consistent applications of

the vertex rules, where each chord diagram is weighted by the boundary $(1 - \mathfrak{q})$ factors and the bulk $\mathfrak{q}^{\# \text{ intersections}}$ according to the rules in figure 9. This reproduces the results in (3.26).

For higher weight representations, we must use the tensor product decompositions into irreducible representations discussed in subsection 3.2.2. Here, we see the second advantage of our choice of normalization: the transfer matrices $T_{\mathcal{R}}$ obey the same tensor product decomposition rules as the Wilson lines $W_{\mathcal{R}}$. For example the decomposition $[1,0] \otimes [0,1] = [1,1] \oplus [0,0]$, gives us the Wilson line for the singlet representation

$$T_{[0,0]} = 1 \,, \tag{4.19}$$

and the one for the adjoint representation

$$T_{[1,1]} = - \mathfrak{q} + (1 - \mathfrak{q}) \left( a_1 a_2 + a_1^\dagger a_2^\dagger + a_1^\dagger a_1 + a_2^\dagger a_2 + (1 - \mathfrak{q})^{\frac{1}{2}} \left( (a_1^\dagger)^2 a_2 + a_1^\dagger a_2^2 + a_1^2 a_2^\dagger + a_1 (a_2^\dagger)^2 \right) \right)$$
$$+ (1 - \mathfrak{q})^2 \mathfrak{q} a_1^\dagger a_1 a_2^\dagger a_2 \,. \tag{4.20}$$

The chord diagrams for the singlet representation are all trivial, meaning there are no chords, no bulk intersections, and no boundary weights, and thus it obviously agrees with the index calculation of

$$\langle T_{[0,0]}^k \rangle = \mathbb{I}_{W_{[0,0]}^k}^{(3)} \,, \tag{4.21}$$

for all values of $k$. The chord diagrams for the adjoint Wilson line are instead less trivial, but they still reproduce

$$\langle T_{[1,1]}^k \rangle = \mathbb{I}_{W_{[1,1]}^k}^{(3)} \,. \tag{4.22}$$

The feature which sets the adjoint chord rules apart from the fundamental and antifundamental rules is that there are more choices of boundary vertices, depicted in figure 14. With the fundametal, there was always the fixed trivalent structure of creating a blue chord, annihilating it and creating a red chord, and then finally annihilating a red chord. Similarly with the antifundametal, there was again a trivalent structure, this time given by first creating a red chord, annihilating it and creating a blue chord, and then finally annihilating a blue chord. In the case of the adjoint representation however, one can have multi-valency structures of different types emerge that are consistent with the application of the boundary vertex rules, see figure 15 for two examples. To compute the VEVs of $T_{[1,1]}^k$, one must now sum over all consistent applications of the vertex rules with a fixed number $k$ of vertices on the circle. The combinatorics here becomes much more cumbersome than previously, as there are now more choices that one has available. Peculiarly, there is the choice to do nothing at the vertex, and acquire a factor of $-\mathfrak{q}$. Clearly doing the sum over all allowed chord diagrams with a fixed

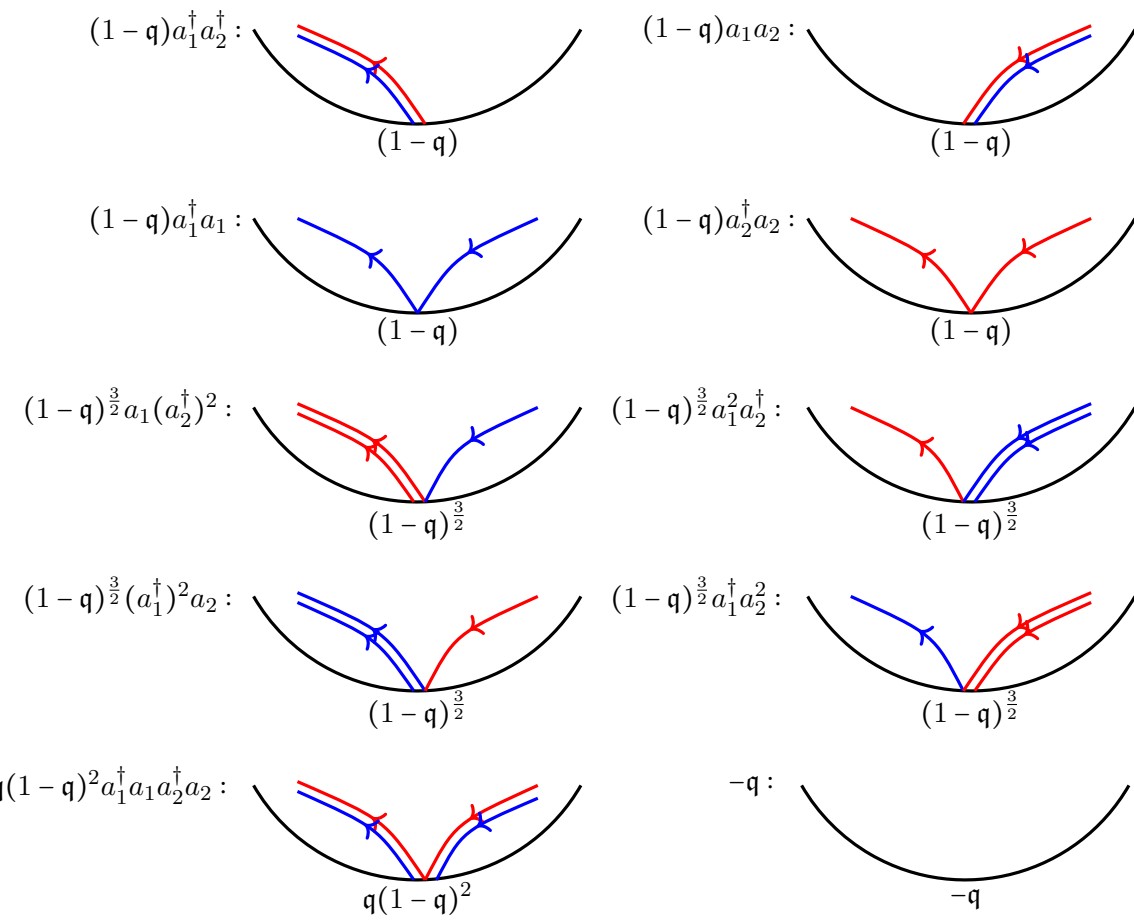

Figure 14: Chord vertex rules for the adjoint transfer matrix $T_{[1,1]}$ of $SU(3)$.

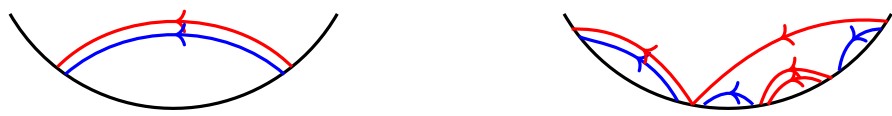

Figure 15: Examples of the different types of chord structures that appear in the transfer matrix VEVs $\langle T^2_{[1,1]} \rangle$ and $\langle T^5_{[1,1]} \rangle$ respectively.

number of external vertices is much more challenging in the case of the adjoint representation. However, this problem can be efficiently solved with the $\mathfrak{q}$-oscillators or with the Schur half-index. Indeed, using the Schur half-index, we can write down the closed form expression of such VEVs

$$\langle T^k_{[1,1]} \rangle = \mathbb{I}^{(3)}_{W^k_{[1,1]}} = \frac{(\mathfrak{q};\mathfrak{q})^2_\infty}{3!} \oint_{\mathbb{T}^2} \prod_{a=1}^2 \frac{\mathrm{d}z_a}{2\pi i z_a} \prod_{a<b}^3 \left( (z_a z_b^{-1})^{\pm 1} ; \mathfrak{q} \right)_\infty \prod_i \left( \chi_{[1,1]}(\vec{z}) \right)^k \Bigg|_{\prod_{a=1}^3 z_a = 1} . \quad (4.23)$$

As another example of Wilson line associated to a higher weight representation, we can consider the decomposition of $SU(3)$ representations $[1,0] \otimes [1,0] = [2,0] \oplus [0,1]$ to find

$$
\begin{aligned}
T_{[2,0]} = {}& (1-\mathfrak{q})^{\frac{1}{2}} \left( -(a_1 + a_2^\dagger)\mathfrak{q} + (1-\mathfrak{q})^{\frac{1}{2}} \left( a_1^\dagger a_1^\dagger + a_1^\dagger a_2 + a_2 a_2 \right) \right) \\
& + (1-\mathfrak{q})^{\frac{3}{2}} \left( a_1 a_1 a_2^\dagger a_2^\dagger + (1+\mathfrak{q}) \left( a_1 a_2^\dagger a_2 + a_1^\dagger a_1 a_2^\dagger \right) \right) .
\end{aligned}
\quad (4.24)
$$

One can extract chord vertex rules from this operator, which contain not only vertices that look like a doubled version of those of the fundamental representation depicted in figure 5, but also more involved vertices similar to those depicted in figure 14.

## 4.3 $SU(N)$ Chord Counting Problem

We shall now briefly discuss the chord counting problem for the fundamental transfer matrix $T_{[1,0,\dots,0]}$ of $SU(N)$. For other lowest weight representations $\mathcal{R}_r$ in (2.8) one can use the character to oscillator map explained in subsection 3.3 to obtain the transfer matrices and then construct the chord problems in a similar way. For higher weight representations one should consider again the decompositions of tensor product representations into irreducible representations.

For $SU(N)$, the chords come in $N-1$ different colors, labeled by $i = 1, \dots, N-1$. In the case of the fundamental representation, only $N$-tuples of external points can be joined simultaneously using these $N-1$ chords as shown in figure 16.

In analogy to the $SU(3)$ case, we can recast the combinatorial problem of counting these chord diagrams in the language of $N-1$ $\mathfrak{q}$-deformed harmonic oscillators. These obey the

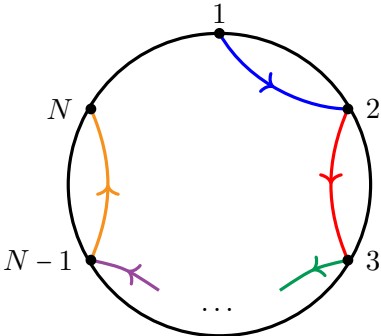

Figure 16: The minimal diagram associated to the fundamental representation of $SU(N)$ has $N$ external vertices joined simultaneously with $N-1$ types of chord.

algebra

$$[a_i, a_i^\dagger]_{\mathfrak{q}} = 1 \,,$$
$$[a_i, a_j^\dagger]_1 = [a_i, a_j]_1 = 0 \,, \qquad i \neq j \,, \tag{4.25}$$

capturing the fact that they are decoupled oscillators, in the sense that bulk intersections of two different colors are counted with weight $\mathfrak{q}^0 = 1$. Instead, the intersection of two chords of the same color has weight $\mathfrak{q}^1$. The fundamental $SU(N)$ transfer matrix is given by

$$T_{[1,0\ldots,0]} = (1-\mathfrak{q})^{\frac{1}{2}} \left( a_1^\dagger + (1-\mathfrak{q})^{\frac{1}{2}} \sum_{i=2}^{N-1} a_i^\dagger a_{i-1} + a_{N-1} \right) \,, \tag{4.26}$$

where the oscillators act on the chord Hilbert space as

$$a_i^\dagger |n_1, \ldots n_{N-1}\rangle = |n_1, \ldots, n_i + 1, \ldots n_{N-1}\rangle \,, \qquad a_i |n_1, \ldots n_{N-1}\rangle = [n_i]_{\mathfrak{q}} |n_1, \ldots, n_i - 1, \ldots n_{N-1}\rangle \,. \tag{4.27}$$

This is exactly the same oscillator algebra and transfer matrices discussed in section 3, and so we will not repeat the Hilbert space construction again here. From the form of the transfer matrix, we see that each vertex connected to a single chord, namely the $i = 1$ and the $i = N-1$ chords, has boundary weight $(1-\mathfrak{q})^{\frac{1}{2}}$, while the other vertices have boundary weight $1-\mathfrak{q}$.

The counting of chord diagrams with $k$ external vertices joined as depicted in figure 16 and with the bulk and boundary weights we have just discussed is solved by computing the VEV $\langle T_{[1,0,\cdots,0]}^k \rangle$, or equivalently by the Schur half-index with the insertion of $k$ fundamental Wilson lines

$$\langle T_{[1,0,\cdots,0]}^k \rangle = \mathbb{I}_{W_{[1,0,\cdots,0]}^k}^{(N)} = \frac{(\mathfrak{q};\mathfrak{q})_\infty^{N-1}}{N!} \oint_{\mathbb{T}^{N-1}} \prod_{a=1}^{N-1} \frac{\mathrm{d}z_a}{2\pi i z_a} \prod_{a<b}^N \left( (z_a z_b^{-1})^{\pm 1} ; \mathfrak{q} \right)_\infty \left( \chi_{[1,0,\cdots,0]}(\vec{z}) \right)^k \bigg|_{\prod_a z_a = 1} \,. \tag{4.28}$$

Such integrals can be explicitly evaluated, resulting in the desired solution of the counting problem. The generalization to other $SU(N)$ representations works similarly to the $SU(3)$

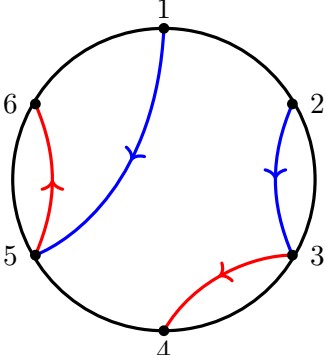 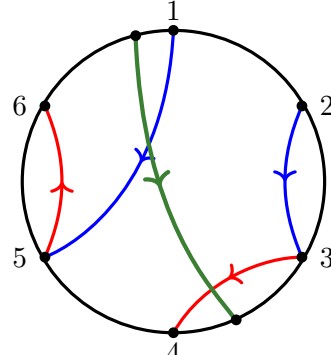

Figure 17: The figure on the left shows a generic diagram which contributes to $\langle T^6_{[1,0]} \rangle$ in the $SU(3)$ case. The diagram on the right is that same diagram, but with the insertion of a matter chord $M_1(\Delta_1)$ (in green) that starts between nodes 6 and 1, and ends between 3 and 4. This diagram contributes $\mathfrak{q}^{\Delta_1}$ to $\langle T^3_{[1,0]} \mathfrak{q}^{\Delta_1 \mathfrak{n}_1} T^3_{[1,0]} \rangle$.

case.

## 4.4 Dyonic Lines and Domain Walls From Matter Chords

The insertion of matter chords into the chord counting picture allows us to realize two different gauge theory objects, namely dyonic lines and 3d domain walls. The logic is that when one inserts a matter chord, corresponding to some operator of scaling dimension $\Delta$, one can use this to cut the chord diagram in two. Counting the number of open chords which cross the matter chords gives us a chordial realization of the number operator $\mathfrak{q}^{\mathfrak{n}}$. First, we explain how the matter chords work, and then we specialise to the case of dyonic lines and domain walls. For simplicity we will focus on the case of $SU(3)$ and on the chord diagrams arising from the fundamental transfer matrix $T_{[1,0]}$ but the same logic can be applied to the more general case.

### Matter Chords and Their $\mathfrak{q}$-oscillator Realization

In figure 17 left, we have a generic diagram that appears in the computation of the $\mathfrak{q}$-oscillator VEV $\langle 0,0|T^6_{[1,0]}|0,0 \rangle$ using the chords perspective. Suppose we want to measure the number of open blue chords between the nodes 3 and 4 in this diagram. Clearly there is one open blue chord between nodes 3 and 4, which starts at node 1 and ends at node 5. The other blue chord between nodes 2 and 3 has already been closed by the time we want to make this measurement. We would like to find some way of counting this one open blue chord and to translate it in the $\mathfrak{q}$-oscillator language.

The strategy is to introduce a new type of chord that we call a *matter chord* in analogy to the $SU(2)$ case of DSSYK. We denote this by $M_1(\Delta_1)$, where $\Delta_1$ is its conformal dimension,

and we draw it in green. In figure 17 right, we show how this matter chord should be placed in the chord diagram in order to solve our task of counting the number of open blue chords between nodes 3 and 4. To perform such counting, we also declare that it intersects blue chords with weight $\mathfrak{q}^{\Delta_1}$ and red chords with weight $\mathfrak{q}^0 = 1$. In this way the matter chord does not count the number of open red chords, only the number of open blue chords. We had to insert the matter chord starting just before node 1, since this is where the diagram starts, and between node 3 and 4, where we wanted to make our measurement. With these rules, the diagram in figure 17 right will correspond to $\mathfrak{q}^{\Delta_1}$, since it has only one green-blue intersection.

In order to translate the insertion of a matter chord in $\mathfrak{q}$-oscillators language, we associate to $M_1(\Delta_1)$ the operator $\mathfrak{q}^{\Delta_1 \mathfrak{n}_1}$. In other words, the diagram in figure 17 right is contributing to the $\mathfrak{q}$-oscillator VEV

$$\langle T^3_{[1,0]}\mathfrak{q}^{\Delta_1 \mathfrak{n}_1} T^3_{[1,0]}\mathfrak{q}^{\Delta_1 \mathfrak{n}_1}\rangle = \langle T^3_{[1,0]}\mathfrak{q}^{\Delta_1 \mathfrak{n}_1} T^3_{[1,0]}\rangle, \tag{4.29}$$

where we used that $\mathfrak{q}^{\Delta_1 \mathfrak{n}_1}|0,0\rangle = |0,0\rangle$. This VEV is indeed counting the number of diagrams with 6 external vertices grading the intersection of the ordinary chords with $\mathfrak{q}$ according to the usual rules explained in Subsection 4.2, but also grading with $\mathfrak{q}^{\Delta_1}$ the number of open blue chords after applying three times the transfer matrix, so between points 3 and 4. Hence, the diagram in figure 17 right will be contributing to this VEV with a term $\mathfrak{q}^{\Delta_1}$, as desired. More generally, if we want to grade diagrams with $k = k_1 + k_2$ external vertices by the number of open blue chords between vertex $k_2$ and $k_2 + 1$, we should consider the $\mathfrak{q}$-oscillators VEV

$$\langle T^{k_1}_{[1,0]}\mathfrak{q}^{\Delta_1 \mathfrak{n}_1} T^{k_2}_{[1,0]}\rangle. \tag{4.30}$$

We can similarly construct the matter chord for counting open red chords, which will be associated to the number operators of the second $\mathfrak{q}$-oscillator. We will denote this matter chord with $M_2(\Delta_2)$ and color it in orange. For this second type of matter chord, we choose trivial intersection weight $\mathfrak{q}^0=1$ between it and a blue chord, and intersection weight $\mathfrak{q}^{\Delta_2}$ with a red chord. Then this other matter chord will be realized with the $\mathfrak{q}$-oscillators by the operator $\mathfrak{q}^{\Delta_2 \mathfrak{n}_2}$, and if we want to count diagrams with $k = k_1 + k_2$ external vertices graded by the number of open red chords between vertex $k_2$ and $k_2 + 1$ we should consider the VEV

$$\langle T^{k_1}_{[1,0]}\mathfrak{q}^{\Delta_2 \mathfrak{n}_2} T^{k_2}_{[1,0]}\rangle. \tag{4.31}$$

In further generality, one can consider inserting both of these two matter chords, in such a way that they start before vertex 1 and end after any point $k_2$ of the diagram (again $k = k_1 + k_2$ is the number of external vertices). This will correspond to the VEV

$$\langle T^{k_1}_{[1,0]}\mathfrak{q}^{\Delta_1 \mathfrak{n}_1 + \Delta_2 \mathfrak{n}_2} T^{k_2}_{[1,0]}\rangle, \tag{4.32}$$

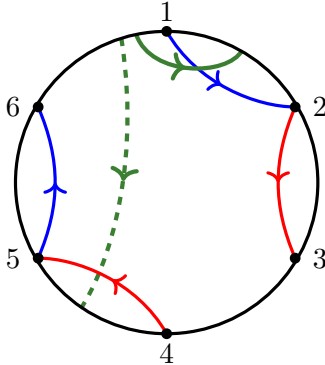

Figure 18: Example of a diagram used to count simultaneously the number of open blue chords between points 1 and 2, and 4 and 5. In terms of $\mathfrak{q}$, this diagram contributes $\mathfrak{q}^{\Delta_{1,1}}$ to the VEV $\langle T^2_{[1,0]}\mathfrak{q}^{\Delta_{1,2}\mathfrak{n}_1}T^3_{[1,0]}\mathfrak{q}^{\Delta_{1,1}\mathfrak{n}_1}T_{[1,0]}\rangle$, encoding the fact that there is one open blue chord between 1 and 2, and none between 4 and 5.

which is counting all chord diagrams with $k$ external vertices with an additional grading of $\mathfrak{q}^{\Delta_1}$ for each open blue chord after vertex $k_2$ and $\mathfrak{q}^{\Delta_2}$ for each open blue chord again after vertex $k_2$. Since the number operators $\mathfrak{n}_i$ all commute, their order inside the VEV is irrelevant and the intersection of different matter chords has trivial weight.

An important subtlety arises when considering the insertion of multiple chords of the same type. Let us consider for example the case in which we want to count simultaneously the number of open blue chords between the external points 1 and 2 as well as 4 and 5. For this purpose, we need to introduce two green matter chords $M_1(\Delta_{1,1})$ and $M_1(\Delta_{1,2})$ with different dimension, as depicted in figure 18. We still draw both of them in green since they have the property of having non-trivial intersection weight with the blue chord but trivial with the red chord, but we distinguish them with a solid and dashed line to stress that they have different dimensions and hence their intersections with blue chords contributes differently, with $\mathfrak{q}^{\Delta_1}$ and $\mathfrak{q}^{\Delta_2}$ respectively. The diagram in figure 18 will then contribute to the $\mathfrak{q}$-oscillators VEV

$$\langle T^2_{[1,0]}\mathfrak{q}^{\Delta_{1,2}\mathfrak{n}_1}T^3_{[1,0]}\mathfrak{q}^{\Delta_{1,1}\mathfrak{n}_1}T_{[1,0]}\rangle \tag{4.33}$$

with a term $\mathfrak{q}^{\Delta_{1,1}}$.

We finally note that the whole logic presented in this subsection can be applied in the same way to chord diagrams generated from transfer matrices $T_{\mathcal{R}}$ for any representation $\mathcal{R}$ and also to higher rank $SU(N)$. In particular for generic $SU(N)$ we will have $N-1$ different types of matter chords, which count the number of open chords of each of the $N-1$ types and which correspond to each of the $N-1$ number operators $\mathfrak{n}_i$.

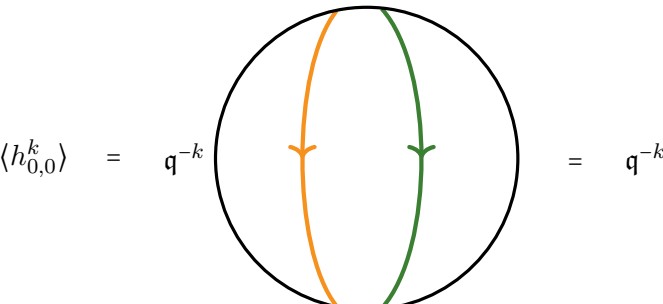

$$\langle h_{0,0}^k\rangle \quad = \quad \mathfrak{q}^{-k} \qquad\qquad\qquad = \quad \mathfrak{q}^{-k}$$

Figure 19: Evaluation of $\langle h_{0,0}^k\rangle = \langle \mathfrak{q}^{-k(\mathfrak{n}_1+\mathfrak{n}_2+1)}\rangle$ using chord diagrams. The green chord corresponds to counting the number of open blue chords that cross the diagram with intersection weight $\mathfrak{q}^{-k}$, and the orange chord counts the number of open red chords that cross the diagram with intersection weight $\mathfrak{q}^{-k}$. These correspond to the operator insertions of $\mathfrak{q}^{-k\mathfrak{n}_1}$ and $\mathfrak{q}^{-k\mathfrak{n}_2}$ respectively. Since there are no insertions of the transfer matrices $T_{\mathcal{R}}$, there are no open blue or red chords to cross the diagram, hence the matter chords give a trivial contribution.

**Chord Rules for Dyonic Lines $h_{n,m}$**

We recall from our discussion in section 3 that the representation of the dyonic lines $h_{n,m}$ of $\mathcal{A}_{\text{Schur}}$ in terms of $\mathfrak{q}$-deformed oscillators is of the form

$$h_{n,m} = \sum_j f_j(\vec{a}^\dagger, \vec{a})\mathfrak{q}^{\sum_i \Delta_{i,j}\mathfrak{n}_i}, \tag{4.34}$$

where $f_j$ are polynomials of the creation and annihilation operators, and $\Delta_{i,j}$ are integers. We would like to give an interpretation at the level of chord diagrams of the $\mathfrak{q}$-oscillator VEVs with the insertion of these operators on top of the transfer matrices $T_{\mathcal{R}}$.

Firstly, we treat each term in the sum over $j$ individually, and assign to each of them matter chords $M_i(\Delta_{i,j})$. As before these matter chords are taken to start right before node 1, and end in the interval corresponding to the location of the $h_{n,m}$ operator in the VEV. Next, we assign chord vertex rules to the coefficient functions $f_i$ in the analogous manner to how we constructed chord rules for generic transfer matrices in subsection 4.2. The chord diagrams constructed in this way will then be computing the desired VEV with the insertion of the $\mathfrak{q}$-oscillator representation of the dyonic line $h_{n,m}$.

As a first example, let us consider a VEV with the insertion of the operator $h_{0,0} = \mathfrak{q}^{-\mathfrak{n}_1-\mathfrak{n}_2-1}$

$$\langle h_{0,0}^k\rangle = \mathfrak{q}^{-k}\langle \mathfrak{q}^{-k\mathfrak{n}_1-k\mathfrak{n}_2}\rangle. \tag{4.35}$$

This is equivalent to decorating the single chord diagram with no external vertices (since we did not insert any transfer matrix in the VEV) with a green matter chord $M_1(\Delta_1 = -k)$ and an orange matter chord $M_2(\Delta_2 = -k)$, see figure 19. Since there is no blue nor red chord that

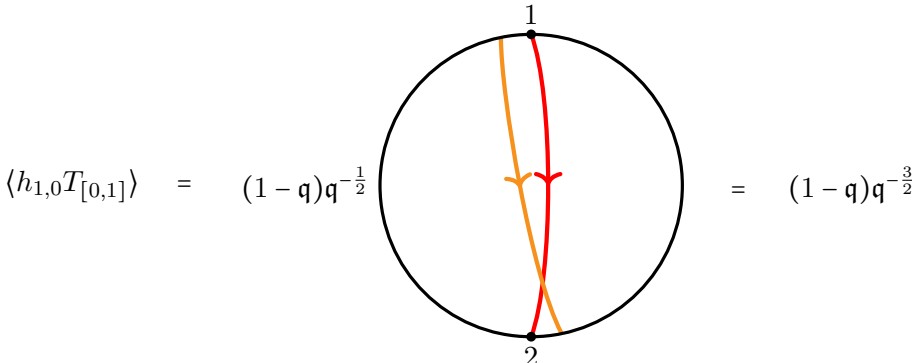

$$\langle h_{1,0} T_{[0,1]} \rangle \quad = \quad (1-\mathfrak{q})\mathfrak{q}^{-\frac{1}{2}} \qquad \qquad \qquad = \quad (1-\mathfrak{q})\mathfrak{q}^{-\frac{3}{2}}$$

Figure 20: Evaluation of $\langle h_{1,0} T_{[0,1]} \rangle$ using chord diagrams. The factor of $\mathfrak{q}^{-1}$ comes from the single bulk intersection of the orange matter chord with the red chord. We avoided drawing the green matter chord since this has trivially contributing intersection with the other chords.

can intersect with the matter chords, the result is simply

$$\langle h_{0,0}^k \rangle = \mathfrak{q}^{-k} \langle \mathfrak{q}^{-k\mathfrak{n}_1 - k\mathfrak{n}_2} \rangle = \mathfrak{q}^{-k} \, . \tag{4.36}$$

For a less trivial example, let us consider the VEV $\langle h_{1,0} T_{[0,1]} \rangle$. This can be computed using $\mathfrak{q}$-oscillators as follows:

$$\begin{aligned} \langle h_{1,0} T_{[0,1]} \rangle &= \langle 0 | (1-\mathfrak{q}) a_2 \mathfrak{q}^{-\mathfrak{n}_1 - \mathfrak{n}_2 - \frac{1}{2}} \left( a_2^\dagger + (1-\mathfrak{q})^{\frac{1}{2}} a_1^\dagger a_2 + a_1 \right) | 0 \rangle \\ &= (1-\mathfrak{q})\mathfrak{q}^{-\frac{3}{2}} \langle 0 | q^{-\mathfrak{n}_1 - \mathfrak{n}_2} a_2 a_2^\dagger | 0 \rangle \\ &= (1-\mathfrak{q})\mathfrak{q}^{-\frac{3}{2}} \, , \end{aligned} \tag{4.37}$$

where to go from the first to the second line we used (3.10) and that only terms with the same number of $a_i^\dagger$ and $a_i$ for any given $i$ contribute. The computation of this VEV using chord diagrams is represented in figure 20. In this case we only have one diagram, which is obtained by creating a red chord at point 1 using the $a_2^\dagger$ part of the transfer matrix $T_{[0,1]}$, closing an orange chord $M_2(\Delta_2 = -1)$ (which was created by adding for free the operator $\mathfrak{q}^{-\mathfrak{n}_2}$ on the very right of the VEV) using the $\mathfrak{q}^{-\mathfrak{n}_2}$ part of $h_{1,0}$ and finally closing the red chord using the $a_2$ part of $h_{1,0}$. Notice that in principle the operator $h_{1,0}$ has also a $\mathfrak{q}^{-\mathfrak{n}_1}$ which should give a green matter chord opened before 1 and closed after 1, however we avoided drawing it since it gives trivial contribution.

This whole structure generalizes in the natural way to $SU(N)$ chord diagrams, and can be used to realize any string of Wilson and dyonic lines inside a VEV.

### Matter 2-point Function and 3d $\mathcal{N} = 2$ Domain Walls

Given a matter chord $M$, one can compute the matter 2-point function, given by

$$G_{\vec{\Delta}}(\beta_1, \beta_2) = \langle e^{-\beta_1 T_{\mathcal{R}_L}} \mathfrak{q}^{\sum_j \Delta_j \mathfrak{n}_j} e^{-\beta_2 T_{\mathcal{R}_R}} \rangle \, , \tag{4.38}$$

where the $\mathfrak{q}^{\Delta_j}$ are the intersection weights for the $j = 1, \ldots N - 1$ different colored chords appearing in the $SU(N)$ chord counting problem. Note that one does not typically introduce dyonic lines into this definition.

By choosing the dimensions $\vec{\Delta}$ of the matter chords appropriately, one can relate the associated VEV to the index of a 3d $\mathcal{N} = 2$ domain wall inside the 4d $\mathcal{N} = 2$ $SU(N)$ SYM theory. Specifically, we take

$$\vec{\Delta} = (-\Delta, -2\Delta, \ldots, -(N-1)\Delta), \tag{4.39}$$

so the associated 2-point function becomes

$$G_{\vec{\Delta}}(\beta_1\beta_2) = \langle e^{-\beta_1 T_{\mathcal{R}_L}} \mathfrak{q}^{\Delta \sum_j j \mathfrak{n}_j} e^{-\beta_2 T_{\mathcal{R}_R}} \rangle. \tag{4.40}$$

We claim that this 2-point function can be computed by the integral

$$
\begin{aligned}
G_{\vec{\Delta}}(\beta_1, \beta_2) = {} & \frac{(\mathfrak{q}; \mathfrak{q})_\infty^{2(N-1)}}{(N!)^2} \oint_{\mathbb{T}^{N-1}} \prod_{a=1}^{N-1} \frac{\mathrm{d}z_a}{2\pi i z_a} \frac{\mathrm{d}u_a}{2\pi i u_a} \prod_{a<b}^{N} \left( (z_a z_b^{-1})^{\pm 1}; \mathfrak{q} \right)_\infty \left( (u_a u_b^{-1})^{\pm 1}; \mathfrak{q} \right)_\infty \\
& \times \frac{(\mathfrak{q}^{N\Delta}; \mathfrak{q})_\infty}{\prod_{a,b=1}^{N} (\mathfrak{q}^\Delta z_a u_b^{-1}; \mathfrak{q})_\infty} e^{-\beta_1 \chi_{\mathcal{R}_L}(\vec{z}) - \beta_2 \chi_{\overline{\mathcal{R}}_R}(\vec{u})} \Bigg|_{\prod_a z_a = \prod_a u_a = 1}.
\end{aligned}
\tag{4.41}
$$

One can check this explicitly to low orders in an expansion in $\beta_1$ and $\beta_2$, but in subsection 5.1 we will give a general proof.

This expression for the VEV corresponding to the insertion of matter chords with the dimensions (4.39) coincides with the Schur half-index of a configuration with two copies of the 4d $\mathcal{N} = 2$ $SU(N)$ SYM separated by a certain 3d $\mathcal{N} = 2$ domain wall. On the domain wall, the vector multiplets of the two copies of SYM are given Neumann boundary conditions, but we also have intrinsically 3d degrees of freedom that live on it. More precisely, we have a chiral multiplet $\Phi$ in the bifundamental representation of the two 4d $SU(N)$ gauge symmetries and a flipping field $F$, that is a gauge singlet chiral field with superpotential[15]

$$\mathcal{W} = F \det \Phi. \tag{4.42}$$

This superpotential preserves a $U(1)$ global symmetry under which $F$ has charge $-N$ and $\Phi$ has charge 1. Hence, the index can be refined with a fugacity $v = \mathfrak{q}^\Delta$ for such symmetry. Furthermore, we give R-charge 2 to $F$ and 0 to $\Phi$, again in compatibility with the superpotential interaction. On the boundary of the Schur half-index, the 3d fields are given Dirichlet boundary conditions for $F$ and Neumann for $\Phi$ [78]. If we further consider adding $k_1$ Wilson

---

[15]This domain has previously been considered in [74, 75]. A 5d/4d version was instead studied in [76] (see also [77]).

lines in a representation $\mathcal{R}_L$ in the left 4d theory and $k_2$ Wilson lines in a representation $\overline{\mathcal{R}}_R$ in the right 4d theory, then the Schur half-index of this system is (see Appendix A for more details)

$$
\begin{aligned}
\mathbb{I}^{(N)}_{W^{k_1}_{\mathcal{R}_L}, W^{k_2}_{\mathcal{R}_R}} &= \frac{(\mathfrak{q};\mathfrak{q})^{2(N-1)}_\infty}{(N!)^2} \oint_{\mathbb{T}^{N-1}} \prod_{a=1}^{N-1} \frac{\mathrm{d}z_a}{2\pi i z_a} \frac{\mathrm{d}u_a}{2\pi i u_a} \prod_{a<b}^{N} \left( \left( z_a z_b^{-1} \right)^{\pm 1} ; \mathfrak{q} \right)_\infty \left( \left( u_a u_b^{-1} \right)^{\pm 1} ; \mathfrak{q} \right)_\infty \\
&\times \frac{\left( \mathfrak{q}^{N\Delta}; \mathfrak{q} \right)_\infty}{\prod_{a,b=1}^{N} \left( \mathfrak{q}^\Delta z_a u_b^{-1}; \mathfrak{q} \right)_\infty} \left( \chi_{\mathcal{R}_L}(\vec{z}) \right)^{k_1} \left( \chi_{\mathcal{R}_R}(\vec{u}) \right)^{k_2} \Bigg|_{\prod_a z_a = \prod_a u_a = 1} .
\end{aligned}
\tag{4.43}
$$

This reproduces the order $\beta_1^{k_1} \beta_2^{k_2}$ in a double expansion in $\beta_1$, $\beta_2$ of the expression (4.41) for the matter 2-point function

$$
\mathbb{I}^{(N)}_{W^{k_1}_{\mathcal{R}_L}, W^{k_2}_{\mathcal{R}_R}} = \langle T^{k_1}_{\mathcal{R}_L} \mathfrak{q}^{-\sum_{j=1}^{N-1} j n_j} T^{k_2}_{\mathcal{R}_R} \rangle .
\tag{4.44}
$$

Hence, inserting the 3d domain wall corresponds to inserting the operator $\mathfrak{q}^{-\sum_{j=1}^{N-1} j n_j}$ in $\mathfrak{q}$-sscillator the VEV, or equivalently the matter chord of dimension (4.39) as in (4.40).

## 5 Spectral Problem for Wilson Lines and the Toda Chain

In section 3 we have argued for an identity (3.58) between the Schur half-indices with the insertion of Wilson lines and the VEVs of the corresponding transfer matrices

$$
\mathbb{I}^{(N)}_{W^{k_1}_{\mathcal{R}_1}, \cdots, W^{k_{N-1}}_{\mathcal{R}_{N-1}}} = \langle T^{k_1}_{\mathcal{R}_1} \cdots T^{k_{N-1}}_{\mathcal{R}_{N-1}} \rangle
\tag{5.1}
$$

We have given evidence for the validity of this equation by computing both sides explicitly for some cases and verifying that they match. In this section we will provide a general proof. We will also prove the identity (4.41) for the matter 2-point function that allowed us to connect it to the 3d $\mathcal{N} = 2$ domain wall.

We will do this by solving the eigenvalue problem for the transfer matrices $T_{\mathcal{R}_r}$. This will then allow us to show that their VEVs admit the same integral representation (2.4) that also computes the Schur half-indices. This analysis generalizes to $SU(N)$ a similar analysis that was done to compute the VEVs of the transfer matrix of the SYK model [20], which corresponds to our $N = 2$ case. As we will see, this diagonalization problem is strictly related to the relativistic open Toda chain [26]. Specifically, we will show that the commuting operators $T_{\mathcal{R}_r}$ coincide with the Hamiltonians $\mathsf{H}_r$ of the Toda chain, whose eigenfunctions are known to be given by a class of orthogonal polynomials called $\mathfrak{q}$-Whittaker polyomials (see e.g. [26–30]). We will also discuss the $\mathfrak{q} \to 1$ limit in which the operators $T_{\mathcal{R}_r}$ reduce to the Hamiltonians of the classical open Toda chain, again generalizing a similar result obtained in [22] for the SYK model.

## 5.1 Diagonalization and Relativistic Open Toda Chain

We begin by setting up the eigenvalue problem for the operators $T_{\mathcal{R}_r}$ that we introduced in subsection 3.3. These can be understood as implementing an evolution in the $\mathfrak{q}$-oscillators Hilbert space, as we discussed from the chord perspective in section 4. Let us focus for example on the fundamental representation

$$T_{[1,0,\cdots,0]} = (1 - \mathfrak{q})^{\frac{1}{2}} \left( a_1^\dagger + (1 - \mathfrak{q})^{\frac{1}{2}} \sum_{i=2}^{N-1} a_i^\dagger a_{i-1} + a_{N-1} \right). \tag{5.2}$$

We consider evolving the system by applying it several times and denote the state at the $a$-th step by $v_{n_1,\cdots,n_{N-1}}^{(a)}$. We then have a recursive relation that expresses the state at the step $a+1$ from the one at the step $a$ via $T_{[1,0,\cdots,0]}$

$$\begin{aligned}
v_{n_1,\cdots,n_{N-1}}^{(a+1)} &= \left( T_{[1,0,\cdots,0]} \, v^{(a)} \right)_{n_1,\cdots,n_{N-1}} \\
&= (1 - \mathfrak{q})^{\frac{1}{2}} v_{n_1-1,n_2,\cdots,n_{N-1}}^{(a)} + \sum_{i=1}^{N-2} (1 - \mathfrak{q}^{n_i+1}) v_{n_1,\cdots,n_i+1,n_{i+1}-1,\cdots,n_{N-1}}^{(a)} \\
&\quad + \frac{1 - \mathfrak{q}^{n_{N-1}+1}}{(1 - \mathfrak{q})^{\frac{1}{2}}} v_{n_1,\cdots,n_{N-2},n_{N-1}+1}^{(a)}.
\end{aligned} \tag{5.3}$$

This is because in order to have a state with $n_1,\cdots,n_{N-1}$ at the step $a+1$ we can only have the following possibilities for the state at the step $a$:

1. at step $a$ we have a state with $n_1 - 1,\cdots,n_{N-1}$ and we apply the creation operator of the first oscillator $a_1^\dagger$ with the prefactor $(1 - \mathfrak{q})^{\frac{1}{2}}$;

2. at step $a$ we have a state $n_1,\cdots,n_i + 1, n_{i+1} - 1,\cdots,n_{N-1}$ for $i = 1,\cdots,N-2$ and we apply $a_i a_{i+1}^\dagger$, with a prefactor $1 - \mathfrak{q}^{n_i+1}$;

3. at step $a$ we have a state $n_1,\cdots,n_{N-1} + 1$ and we apply $a_{N-1}$, with a prefactor $\frac{1-\mathfrak{q}^{n_{N-1}+1}}{(1-\mathfrak{q})^{\frac{1}{2}}}$.

In order to actually determine the state $v_{n_1,\cdots,n_{N-1}}^{(a)}$ at a generic step we need not only this recursive relation, but also an initial condition. For the chord diagrams of section 4 the initial condition is the vacuum state $v_{0,\cdots,0}^{(a)}$ but for the following analysis we can keep it generic.

Let us now consider the eigenvalue problem for the operator $T_{[1,0,\cdots,0]}$

$$T_{[1,0,\cdots,0]} v_{n_1,\cdots,n_{N-1}} = E_{[1,0,\cdots,0]} v_{n_1,\cdots,n_{N-1}}. \tag{5.4}$$

It is convenient to take

$$v_{n_1,\cdots,n_{N-1}} = (1 - \mathfrak{q})^{\frac{1}{2} \sum_i n_i} \mathcal{W}_{n_1,\cdots,n_{N-1}}. \tag{5.5}$$

Then the eigenvalue equation becomes

$$\mathcal{W}_{n_1-1,n_2,\cdots,n_{N-1}} + \sum_{i=1}^{N-2} \left(1 - \mathfrak{q}^{n_i+1}\right) \mathcal{W}_{n_1,\cdots,n_i+1,n_{i+1}-1,\cdots,n_{N-1}}$$
$$+ \left(1 - \mathfrak{q}^{n_{N-1}+1}\right) \mathcal{W}_{n_1,\cdots,n_{N-2},n_{N-1}+1} = E_{[1,0,\cdots,0]} \mathcal{W}_{n_1,\cdots,n_{N-1}} \,. \tag{5.6}$$

This relation generalizes to higher $N$ the one satisfied by the continuous $\mathfrak{q}$-Hermite polynomials, which is used in the case $N = 2$ in the context of the SYK model [20].

The l.h.s. of (5.6) corresponds precisely to the action of the first Hamiltonian of the relativistic open Toda chain [26] of type $\mathfrak{su}(N)$. To show this, we start from the Hamiltonian for type $\mathfrak{u}(N)$, which can be written as [29, 30] (see also Appendix B)

$$\mathsf{H}_1 = D_1 + \sum_{i=1}^{N-1} \left(1 - \mathfrak{q}^{p_{i+1}-p_i+1}\right) D_{i+1} \,, \tag{5.7}$$

where the difference operators $D_i$ act as

$$D_i f_{p_1,\cdots,p_N} = f_{p_1,\cdots,p_i+1,\cdots,p_N} \,. \tag{5.8}$$

In order to match this Hamiltonian with the l.h.s. of (5.6) we identify

$$n_i = p_{i+1} - p_i \,. \tag{5.9}$$

Note that this identification fixes all the $p_i$ up to their sum $\pi = \sum_{i=1}^{N} p_i$ which corresponds to the $\mathfrak{u}(1)$ part of $\mathfrak{u}(N)$. Moreover one can easily check that with the identification (5.9) the difference operators $D_i$ act on the parameters $n_i$ and $\pi$ as

$$D_i f_{n_1,\cdots,n_{N-1},\pi} = \begin{cases} f_{n_1-1,n_2,\cdots,n_{N-1},\pi+1} & i = 1 \,, \\ f_{n_1,\cdots,n_{i-1}+1,n_i-1,\cdots,n_{N-1},\pi+1} & i = 2,\cdots,N-1 \,, \\ f_{n_1,\cdots,n_{N-2},n_{N-1}+1,\pi+1} & i = N \,. \end{cases} \tag{5.10}$$

In particular, all the $D_i$ act in the same way on $\pi$ by shifting it by 1. This action is analogous to that of the last $\mathfrak{u}(N)$ Hamiltonian $\mathsf{H}_N = \prod_{i=1}^{N} D_i$

$$\mathsf{H}_N f_{n_1,\cdots,n_{N-1},\pi} = f_{n_1,\cdots,n_{N-1},\pi+N} \,. \tag{5.11}$$

Its $N$-th square root coincides with the single Hamiltonian for the $\mathfrak{u}(1)$ part of $\mathfrak{u}(N)$ and can be factored out of the first $\mathfrak{u}(N)$ Hamiltonian

$$\mathsf{H}_1 = \mathsf{H}_N^{\frac{1}{N}} \left(\tilde{D}_1 + \sum_{i=1}^{N-1} \left(1 - \mathfrak{q}^{p_{i+1}-p_i+1}\right) \tilde{D}_{i+1}\right) \,, \tag{5.12}$$

where we have defined the new difference operators $\tilde{D}_i$ acting only on the $n_i$ and not on $\pi$

$$\tilde{D}_i f_{n_1,\cdots,n_{N-1},\pi} = \begin{cases} f_{n_1-1,n_2,\cdots,n_{N-1},\pi} & i = 1 \,, \\ f_{n_1,\cdots,n_{i-1}+1,n_i-1,\cdots,n_{N-1},\pi} & i = 2,\cdots,N-1 \,, \\ f_{n_1,\cdots,n_{N-2},n_{N-1}+1,\pi} & i = N \,. \end{cases} \tag{5.13}$$

To obtain the $\mathfrak{su}(N)$ Hamiltonian we should trivialize the $\mathfrak{u}(1)$ part of $\mathfrak{u}(N)$. This is achieved by removing the overall $\mathsf{H}_N^{\frac{1}{N}}$ and setting $\pi = 0$. After the change of variables (5.9) we get

$$\mathsf{H}'_1 = \tilde{D}_1 + \sum_{i=1}^{N-1} \left(1 - \mathfrak{q}^{n_i+1}\right) \tilde{D}_{i+1} . \tag{5.14}$$

By our analysis we can see that the action of this operator coincides precisely with the l.h.s. of (5.6)

$$\mathsf{H}'_1 \mathcal{W}_{n_1,\cdots,n_{N-1}} = \mathcal{W}_{n_1-1,n_2,\cdots,n_{N-1}} + \sum_{i=1}^{N-2} \left(1 - \mathfrak{q}^{n_i+1}\right) \mathcal{W}_{n_1,\cdots,n_i+1,n_{i+1}-1,\cdots,n_{N-1}}$$
$$+ \left(1 - \mathfrak{q}^{n_{N-1}+1}\right) \mathcal{W}_{n_1,\cdots,n_{N-2},n_{N-1}+1} . \tag{5.15}$$

In summary, we have shown that the transfer matrix of the fundamental Wilson line coincides with the first Hamiltonian $\mathsf{H}'_1$ of the relativistic open Toda chain of type $\mathfrak{su}(N)$

$$\mathsf{H}'_1 = T_{[1,0,\cdots,0]} . \tag{5.16}$$

Remember that for generic $N$ all the $N-1$ operators $T_{\mathcal{R}_r}$ in the representations (2.8) commute. The open Toda chain of type $\mathfrak{su}(N)$ is also characterized by a set of $N-1$ commuting Hamiltonians $\mathsf{H}'_r$. We thus expect the transfer matrices $T_{\mathcal{R}_r}$ to be related to the Toda Hamiltonians

$$\mathsf{H}'_r = T_{\mathcal{R}_r} . \tag{5.17}$$

We will show this for the case of the anti-fundamental representation (the other cases work similarly but are more complicated). By similar manipulations to those that we did for the fundamental representation, the eigenvalue equation of the operator $T_{[0,\cdots,0,1]}$ can be rewritten as

$$\mathcal{W}_{n_1,\cdots,n_{N-1}-1} + \sum_{i=1}^{N-2} \left(1 - \mathfrak{q}^{n_{i+1}+1}\right) \mathcal{W}_{n_1,\cdots,n_i-1,n_{i+1}+1,\cdots,n_{N-1}}$$
$$+ \left(1 - \mathfrak{q}^{n_1+1}\right) \mathcal{W}_{n_1+1,n_2,\cdots,n_{N-1}-1} = E_{[0,\cdots,0,1]} \mathcal{W}_{n_1,\cdots,n_{N-1}} . \tag{5.18}$$

We would like to compare this with the last Toda Hamiltonian. The one for type $\mathfrak{u}(N)$ can be worked out from the expressions given in [29, 30] (see also Appendix B)

$$\mathsf{H}_{N-1} = \prod_{j=1}^{N-1} D_j + \sum_{i=1}^{N-1} \left(1 - \mathfrak{q}^{p_{i+1}-p_i+1}\right) \prod_{j \neq i}^{N} D_j . \tag{5.19}$$

Observe that each term contains $N-1$ difference operators, so we can factor out $\mathsf{H}_N^{\frac{N-1}{N}}$. Performing also the redefinition (5.9) we get the $\mathfrak{su}(N)$ Hamiltonian

$$\mathsf{H}'_{N-1} = \prod_{j=1}^{N-1} \tilde{D}_j + \sum_{i=1}^{N-1} \left(1 - \mathfrak{q}^{n_i+1}\right) \prod_{j \neq i}^{N} \tilde{D}_j . \tag{5.20}$$

What is left to do is to show that the combinations of difference operators that appear in this Hamiltonian act exactly as the l.h.s. of (5.18). This is indeed true since from (5.13) one can verify that

$$
\prod_{\substack{j \neq i}}^{N} \tilde{D}_j f_{n_1, \cdots, n_{N-1}, \pi} = \begin{cases} f_{n_1+1, n_2, \cdots, n_{N-1}, \pi} & i = 1 \,, \\ f_{n_1, \cdots, n_{i-1}-1, n_i+1, \cdots, n_{N-1}, \pi} & i = 2, \cdots, N-1 \,, \\ f_{n_1, \cdots, n_{N-2}, n_{N-1}-1, \pi} & i = N \,. \end{cases} \tag{5.21}
$$

Hence

$$
\mathsf{H}'_{N-1} \mathcal{W}_{n_1, \cdots, n_{N-1}} = \mathcal{W}_{n_1, \cdots, n_{N-1}-1} + \sum_{i=1}^{N-2} \left(1 - \mathsf{q}^{n_{i+1}+1}\right) \mathcal{W}_{n_1, \cdots, n_i-1, n_{i+1}+1, \cdots, n_{N-1}}
$$
$$
+ \left(1 - \mathsf{q}^{n_1+1}\right) \mathcal{W}_{n_1+1, n_2, \cdots, n_{N-1}-1} \,, \tag{5.22}
$$

which tells us that

$$
\mathsf{H}'_{N-1} = T_{[0, \cdots, 0, 1]} \,. \tag{5.23}
$$

Since we have a set of commuting operator $T_{\mathcal{R}_r}$, we can try to find a common basis of eigenfunctions, which should satisfy $N-1$ recursion relations similar to (5.6)-(5.18). It is known (see e.g. [26–30]) that a common basis of eigenfunctions of the relativistic open Toda Hamiltonians is given by the $\mathsf{q}$-*Whittaker polynomials* $\mathcal{W}_{\vec{n}}(\vec{z}; \mathsf{q})$, which correspond to the $t \to 0$ limit of the Macdonald polynomials (see Appendix B for some of their properties). In our case we have those of $\mathfrak{su}(N)$ type, which are functions of $N$ complex variables $z_1, \cdots, z_N$ with $|z_a| = 1$, i.e. defined on the unit circle, and subject to the constraint $\prod_a z_a = 1$. Moreover, the eigenvalues for each Hamiltonian are exactly the corresponding characters written in terms of the variables $\vec{z}$

$$
E_{\mathcal{R}} = \chi_{\mathcal{R}}(\vec{z}) \,. \tag{5.24}
$$

Since we have shown that the $T_{\mathcal{R}_r}$ operators coincide with the Toda Hamiltonians, this means that the $\mathsf{q}$-Whittaker polynomials solve our diagonalization problem

$$
T_{\mathcal{R}_r} \mathcal{W}_{\vec{n}}(\vec{z}; \mathsf{q}) = \mathsf{H}'_r \mathcal{W}_{\vec{n}}(\vec{z}; \mathsf{q}) = \chi_{\mathcal{R}}^{SU(N)}(\vec{z}) \mathcal{W}_{\vec{n}}(\vec{z}; \mathsf{q}) \,. \tag{5.25}
$$

The $\mathsf{q}$-Whittaker polynomials are orthogonal with respect to the measure

$$
\Delta(\vec{z}; \mathsf{q}) = \frac{(\mathsf{q}; \mathsf{q})_\infty^{N-1}}{N!} \prod_{a<b}^{N} \left( \left(z_a z_b^{-1}\right)^{\pm 1}; \mathsf{q} \right)_\infty \bigg|_{\prod_a z_a = 1} \,, \tag{5.26}
$$

which is exactly the same one we use in the Schur half-index (2.4). Explicitly we have

$$
\oint_{\mathbb{T}^{N-1}} \prod_{a=1}^{N-1} \frac{\mathrm{d}z_a}{2\pi i z_a} \Delta(\vec{z}; \mathsf{q}) \mathcal{W}_{\vec{n}}(\vec{z}; \mathsf{q}) \mathcal{W}_{\vec{m}}(\vec{z}; \mathsf{q}) = \delta_{\vec{n}, \vec{m}} \,. \tag{5.27}
$$

Thinking of the $\mathfrak{q}$-Whittaker polynomials as the inner product of the transfer matrix eigenstates and the open chord states $\mathcal{W}_{\vec{n}}(\vec{z}; \mathfrak{q}) = \langle \vec{n} | \vec{z} \rangle$, this gives us the completeness relation satisfied by the states $|\vec{z}\rangle$

$$\oint_{\mathbb{T}^{N-1}} \prod_{a=1}^{N-1} \frac{\mathrm{d}z_a}{2\pi i z_a} \Delta(\vec{z}; \mathfrak{q}) |\vec{z}\rangle \langle \vec{z}| = I \,, \tag{5.28}$$

which we can insert inside any VEV of the product of transfer matrices so to express them with an integral form that precisely coincides with that of the Schur half-index (2.4)

$$\begin{aligned}
\langle T_{\mathcal{R}_1}^{k_1} \cdots T_{\mathcal{R}_{N-1}}^{k_{N-1}} \rangle = \langle \vec{0} | T_{\mathcal{R}_1}^{k_1} \cdots T_{\mathcal{R}_{N-1}}^{k_{N-1}} | \vec{0} \rangle &= \oint_{\mathbb{T}^{N-1}} \prod_{a=1}^{N-1} \frac{\mathrm{d}z_a}{2\pi i z_a} \Delta(\vec{z}; q) \langle \vec{0} | \vec{z} \rangle \langle \vec{z} | T_{\mathcal{R}_1}^{k_1} \cdots T_{\mathcal{R}_{N-1}}^{k_{N-1}} | \vec{0} \rangle \\
&= \oint_{\mathbb{T}^{N-1}} \prod_{a=1}^{N-1} \frac{\mathrm{d}z_a}{2\pi i z_a} \Delta(\vec{z}; q) \prod_{r=1}^{N-1} \left( \chi_{\mathcal{R}_i}(\vec{z}) \right)^{k_r} \Bigg|_{\prod_a z_a = 1} \\
&= \mathbb{I}^{(N)}_{W_{\mathcal{R}_1}^{k_1}, \cdots, W_{\mathcal{R}_{N-1}}^{k_{N-1}}} \,,
\end{aligned} \tag{5.29}$$

where we used (5.25) and that $\langle \vec{0} | \vec{z} \rangle = 1$. This completes the proof of (5.1).

With a similar strategy we can also prove the integral form (4.41) of the matter 2-point function with the scaling dimension (4.39) that allowed us to make connection with the 3d $\mathcal{N} = 2$ domain wall inside the 4d $\mathcal{N} = 2$ $SU(N)$ SYM. Such 2-point function reads

$$G_{\vec{\Delta}}(\beta_1 \beta_2) = \langle \mathrm{e}^{-\beta_1 T_{\mathcal{R}_L}} \mathfrak{q}^{\Delta \sum_j j \mathfrak{n}_j} \mathrm{e}^{-\beta_2 T_{\mathcal{R}_R}} \rangle \,. \tag{5.30}$$

We first insert a complete set of open chord states (i.e. eigenstates of the number operators)

$$\begin{aligned}
G_{\vec{\Delta}}(\beta_1, \beta_2) &= \sum_{\vec{n} \in \mathbb{N}_0^{N-1}} \langle 0 | \mathrm{e}^{-\beta_1 T_{\mathcal{R}_L}} | \vec{n} \rangle \langle \vec{n} | \mathfrak{q}^{\Delta \sum_{j=1}^{N-1} j \mathfrak{n}_j} \mathrm{e}^{-\beta_2 T_{\mathcal{R}_R}} | 0 \rangle \\
&= \sum_{\vec{n} \in \mathbb{N}_0^{N-1}} \langle 0 | \mathrm{e}^{-\beta_1 T_{\mathcal{R}_L}} | \vec{n} \rangle \mathfrak{q}^{\Delta \sum_{j=1}^{N-1} j n_j} \langle \vec{n} | \mathrm{e}^{-\beta_2 T_{\mathcal{R}_R}} | 0 \rangle \,.
\end{aligned} \tag{5.31}$$

Next, we use which implies the completeness relation (5.28)[16]

$$\begin{aligned}
G_{\vec{\Delta}}(\beta_1, \beta_2) &= \frac{(\mathfrak{q}; \mathfrak{q})_\infty^{2(N-1)}}{(N!)^2} \oint_{\mathbb{T}^{N-1}} \prod_{a=1}^{N-1} \frac{\mathrm{d}z_a}{2\pi i z_a} \frac{\mathrm{d}u_a}{2\pi i u_a} \prod_{a<b}^{N} \left( (z_a z_b^{-1})^{\pm 1}; \mathfrak{q} \right)_\infty \left( (u_a u_b^{-1})^{\pm 1}; \mathfrak{q} \right)_\infty \\
&\quad \times \sum_{\vec{n} \in \mathbb{N}_0^{N-1}} \mathfrak{q}^{\Delta \sum_{j=1}^{N-1} j m_j} \mathcal{W}_{\vec{n}}(\vec{z}; \mathfrak{q}) \mathcal{W}_{\vec{n}}(\vec{u}^{-1}; \mathfrak{q}) \mathrm{e}^{-\beta_1 \chi_{\mathcal{R}_L}(\vec{z}) - \beta_2 \chi_{\overline{\mathcal{R}}_R}(\vec{u})} \Bigg|_{\prod_a z_a = \prod_a u_a = 1} .
\end{aligned} \tag{5.32}$$

We can now get rid of the sum over $\vec{n}$ by using the following property of the $\mathfrak{q}$-Whittaker polynomials (see Appendix B for a derivation):

$$\sum_{\vec{n} \in \mathbb{N}_0^{N-1}} \mathfrak{q}^{\Delta \sum_{j=1}^{N-1} j n_j} \mathcal{W}_{\vec{n}}(\vec{z}; \mathfrak{q}) \mathcal{W}_{\vec{n}}(\vec{u}^{-1}; \mathfrak{q}) \Bigg|_{\prod_a z_a = \prod_a u_a = 1} = \frac{(\mathfrak{q}^{N\Delta}; \mathfrak{q})_\infty}{\prod_{a,b=1}^{N} (\mathfrak{q}^{\Delta} z_a u_b^{-1}; \mathfrak{q})_\infty} \Bigg|_{\prod_a z_a = \prod_a u_a = 1} . \tag{5.33}$$

---

[16]We have performed a change of variables $\vec{u} \to \vec{u}^{-1}$ for later convenience.

Inserting this expression into the 2-point function, one recovers

$$
\begin{aligned}
G_{\vec{\Delta}}(\beta_1, \beta_2) = {} & \frac{(\mathfrak{q};\mathfrak{q})_\infty^{2(N-1)}}{(N!)^2} \oint_{\mathbb{T}^{N-1}} \prod_{a=1}^{N-1} \frac{\mathrm{d}z_a}{2\pi i z_a} \frac{\mathrm{d}u_a}{2\pi i u_a} \prod_{a<b}^{N} \left( (z_a z_b^{-1})^{\pm 1} ; \mathfrak{q} \right)_\infty \left( (u_a u_b^{-1})^{\pm 1} ; \mathfrak{q} \right)_\infty \\
& \times \frac{(\mathfrak{q}^{N\Delta};\mathfrak{q})_\infty}{\prod_{a,b=1}^{N} (\mathfrak{q}^{\Delta} z_a u_b^{-1};\mathfrak{q})_\infty} e^{-\beta_1 \chi_{\mathcal{R}_L}(\vec{z}) - \beta_2 \chi_{\overline{\mathcal{R}}_R}(\vec{u})} \Bigg|_{\prod_a z_a = \prod_a u_a = 1} .
\end{aligned}
\tag{5.34}
$$

## 5.2 Limit to the classical open Toda chain

The relativistic Toda chain reduces to the classical one in the limit $\mathfrak{q} \to 1$. Here we will explain how this limit should be taken at the level of the operators $T_{\mathcal{R}_r}$ in order to recover the Hamiltonians of the classical open Toda chain of type $\mathfrak{su}(N)$. Our analysis generalizes to any $N$ the one done in [22] for $N = 2$ in the context of the SYK model.

We start recalling that the Hamiltonian of the classical open Toda chain in the $\mathfrak{u}(N)$ case is given by

$$
H = \frac{1}{2} \sum_{a=1}^{N} P_a^2 + \sum_{a=1}^{N-1} e^{Q_a - Q_{a+1}} ,
\tag{5.35}
$$

where the coordinates satisfy the canonical commutations relations

$$
[Q_a, P_b] = i\delta_{ab} .
\tag{5.36}
$$

We can obtain the $\mathfrak{su}(N)$ Hamiltonian by performing the change of variables

$$
\begin{aligned}
Q_1 &= \eta + q_1 , & Q_a &= \eta + q_a - q_{a-1} , & a &= 2, \cdots, N-1 , & Q_N &= \eta - q_{N-1} , \\
P_1 &= \pi + p_1 , & P_a &= \pi + p_a - p_{a-1} , & a &= 2, \cdots, N-1 , & P_N &= \pi - p_{N-1} .
\end{aligned}
\tag{5.37}
$$

The original Hamiltonian then takes the form

$$
H = H_\pi + H' .
\tag{5.38}
$$

The $\mathfrak{u}(1)$ Hamiltonian

$$
H_\pi = \frac{N}{2} \pi^2
\tag{5.39}
$$

is written in terms of the variables $\eta$, $\pi$ which satisfy the commutation relation

$$
[\eta, \pi] = \frac{i}{N} .
\tag{5.40}
$$

The $\mathfrak{su}(N)$ Hamiltonian that we are interested in is instead

$$
H' = \frac{1}{2} \sum_{a,b=1}^{N-1} p_a C_{ab} p_b + \sum_{a=1}^{N-1} e^{\sum_{b=1}^{N-1} C_{ab} q_a} ,
\tag{5.41}
$$

where $C$ is the $\mathfrak{su}(N)$ Cartan matrix

$$C = \begin{pmatrix} 2 & -1 & 0 & \cdots & 0 \\ -1 & 2 & -1 & \cdots & 0 \\ 0 & -1 & 2 & \cdots & 0 \\ \vdots & \vdots & \vdots & \ddots & -1 \\ 0 & 0 & 0 & -1 & 2 \end{pmatrix} \tag{5.42}$$

and the variables $q_a$, $p_a$ satisfy the commutation relations

$$[q_a, p_b] = \frac{2N - (N-1)|a - b| - |N - a - b|}{2N} i . \tag{5.43}$$

This Hamiltonian can be recovered from any of the operators $T_{\mathcal{R}_r}$ as follows. We first perform the redefinition (a similar transformation was done for $SU(2)$ in [79])

$$a_i^\dagger = \frac{e^{ik_i\lambda}}{(1 - \mathfrak{q})^{\frac{1}{2}}} , \qquad a_i = \frac{e^{-ik_i\lambda}(1 - \mathfrak{q}^{n_i})}{(1 - \mathfrak{q})^{\frac{1}{2}}} , \qquad l_i = \lambda n_i , \tag{5.44}$$

where we also used $\mathfrak{q} = e^{-\lambda}$. Note that since the rescaled length operators $l_i$ are quantized, the momenta $k_i$ are periodic. One can check that the new operators $l_i$, $k_i$ satisfy the canonical commutation relations (up to a sign)

$$[l_i, k_j] = -i\delta_{ij} . \tag{5.45}$$

We also write the eigenvalues of the operators $T_{\mathcal{R}_r}$ as

$$E_{\mathcal{R}_r}(\vec{k}) = \chi_{\mathcal{R}_r}\left(e^{ik_1\lambda}, \cdots, e^{ik_{N-1}\lambda}\right) . \tag{5.46}$$

For convenience, we finally define

$$\tilde{T}_{\mathcal{R}_r} = T_{\mathcal{R}_r} - E_{\mathcal{R}_r}(\vec{0}) . \tag{5.47}$$

We then take the limit

$$\lambda \to 0 , \qquad l \to \infty , \qquad e^{-\tilde{l}} = e^{-l}\lambda^{-2} = \text{finite} . \tag{5.48}$$

One can check that the leading order in the $\lambda = -\log(\mathfrak{q})$ expansion of any of the $\tilde{T}_{\mathcal{R}_r}$ operators is

$$\tilde{T}_{\mathcal{R}_r} = -\left(\frac{1}{2}\sum_{i,j=1}^{N-1} k_i C_{ij} k_j + \sum_{i=1}^{N-1} e^{-\tilde{l}_i}\right)\lambda^2 + O\left(\lambda^3\right) . \tag{5.49}$$

This coincides with the $\mathfrak{su}(N)$ Toda Hamiltonian (5.41) up to the identification

$$k_i = p_i , \qquad \tilde{l}_i = -\sum_{j=1}^{N-1} C_{ij} q_j . \tag{5.50}$$

One can also check that the commutators (5.45) correctly imply the commutators (5.43) with this identification.

The classical open $\mathfrak{su}(N)$ Toda chain, as the relativistic one, is also known to have $N-1$ commuting conserved charges. Since we have shown in the previous section that the $N-1$ commuting operators $T_{\mathcal{R}_r}$ coincide with the Hamiltonians of the relativistic open Toda chain for any value of $\mathfrak{q}$, we expect to be able to recover all of the conserved charges of the classical Toda chain by taking the $\mathfrak{q} \to 1$ limit of the $T_{\mathcal{R}_r}$ operators.

Let us focus for example on $N = 3$, where we only have the operators $T_{[1,0]}$ and $T_{[0,1]}$ defined in (3.21). As we have just discussed, both of them reduce to the $\mathfrak{su}(3)$ Toda Hamiltonian to leading order $\lambda^2$ in the $\lambda = \log(\mathfrak{q})$ expansion. However, if we consider the limit of the operator $T_{[1,0]} - T_{[0,1]}$ then the order $\lambda^2$ vanishes and the first non-trivial contribution will be at order $\lambda^3$. This will give us an operator that is expected to commute with $H'$, since the original operators $T_{[1,0]}$ and $T_{[0,1]}$ were commuting with each other. We find

$$
\begin{aligned}
i\left(\tilde{T}_{[1,0]} - \tilde{T}_{[0,1]}\right) &= \left(k_1^2 k_2 - k_1 k_2^2 + k_2 e^{-\tilde{l}_1} - k_1 e^{-\tilde{l}_2}\right)\lambda^2 + O\left(\lambda^3\right) \\
&= \left(p_1^2 p_2 - p_1 p_2^2 + p_2 e^{2q_1 - q_2} - p_1 e^{-q_1 + 2q_2}\right)\lambda^2 + O\left(\lambda^3\right).
\end{aligned}
\tag{5.51}
$$

This coincides (up to a prefactor) with the second conserved charge of the open $\mathfrak{su}(3)$ Toda chain, which can be computed as $\frac{1}{3}\mathrm{Tr}(L^3)$ where $L$ is the Lax matrix. For generic $\mathfrak{u}(N)$, the Lax matrix is an $N \times N$ matrix that can be written as

$$
L = \begin{pmatrix}
P_1 & b_1 & 0 & 0 & \cdots & 0 \\
b_1 & P_2 & b_2 & 0 & \cdots & 0 \\
0 & b_2 & P_3 & b_3 & \cdots & 0 \\
0 & 0 & b_3 & P_4 & \cdots & 0 \\
\vdots & \vdots & \vdots & \vdots & \ddots & b_{N-1} \\
0 & 0 & 0 & 0 & b_{N-1} & P_N
\end{pmatrix},
\tag{5.52}
$$

where

$$
b_a = e^{\frac{Q_a - Q_{a-1}}{2}}, \qquad a = 1, \cdots, N-1.
\tag{5.53}
$$

Performing the redefinition (5.37) with $\pi = 0$ gives the $SU(N)$ Lax matrix, which in particular satisfies

$$
\mathrm{Tr}(L) = 0, \qquad \frac{1}{2}\mathrm{Tr}(L^2) = H'.
\tag{5.54}
$$

We can verify that the same logic works also for $N = 4$. As mentioned in our general discussion of subsection 3.3, the three operators $T_{[1,0,0]}$, $T_{[0,1,0]}$, $T_{[0,0,1]}$ all lead to leading

order to the Hamiltonian. The higher conserved charges can be obtained from the differences

$$i\left(2\tilde{T}_{[1,0,0]} - \tilde{T}_{[0,1,0]}\right) = i\left(\tilde{T}_{[0,1,0]} - 2\tilde{T}_{[0,0,1]}\right) + O\left(\lambda^4\right) = i\left(\tilde{T}_{[1,0,0]} - \tilde{T}_{[0,0,1]}\right) + O\left(\lambda^4\right)$$

$$= \frac{1}{3}\mathrm{Tr}(L^3)\lambda^3 + O\left(\lambda^4\right),$$

$$\tilde{T}_{[1,0,0]} - \tilde{T}_{[0,1,0]} + \tilde{T}_{[0,0,1]} = \left(\frac{1}{4}\mathrm{Tr}(L^4) - \frac{1}{8}\left(\mathrm{Tr}(L^2)\right)^2\right)\lambda^4 + O\left(\lambda^5\right). \tag{5.55}$$

# 6 Comments and Outlook

We have provided evidence for a connection between the Schur half-index for pure $\mathcal{N} = 2$ $SU(N)$ SYM theory with line operator insertions, and a generalized chord counting problem that equivalently has a description in terms of $\mathfrak{q}$-deformed harmonic oscillators, which in turn comes from the a $\mathfrak{q}$-Weyl realization that we derived. In as such this is a combinatorial or quantum mechanical recasting of the Schur index. There are various connections which are worthwhile exploring further, but have certain challenges, which we will summarize here.

**DSSYK Connection.** For $N = 2$ the chord diagrams were observed to be directly connected to certain moments of the Hamiltonian appearing in the partition function of the DSSYK model. For $N > 2$ there seems to not be an obvious connection to an SYK-like model, whose partition function would be computed in terms of the generalized chord counting relevant for the Schur index. For example, for $N = 3$ we find colored chords that are effectively trivalent (and for general $N$, there are $N - 1$ colors and an $N$-valent structure of vertices), the interpretation of which from an SYK-like perspective, where the chords would originate from Wick contractions, is less clear. Let us list the requirements for such a putative model (for the $SU(3)$ example for simplicity) it needs to: 1. incorporate color changing boundary vertices; it needs to give rise to multiple transfer matrices $T_{[1,0]}$ and $T_{[0,1]}$; 2. give rise to oriented chords (hinting at Dirac fermions with chemical potential turned on [80]), but with simple intersection rules (hinting at Majorana fermions). Clearly it would be very interesting to find a generalization of the DSSYK model whose partition functions are computed in terms of our generalized chords. This would in turn connect the Schur half-index to that model and provide a full-fledged extension of the proposal for $SU(2)$ SYM in [1].

**Connection to Liouville and Toda field theory.** There is a known connection between the double-scaling limit of the SYK model, and 2d Liouville theory on a Möbius strip with non-conformal boundary conditions. In fact, one can show directly (see Appendix H of [81]) that performing perturbation theory in the cosmological constant of Liouville on a Möbius strip gives rise to the same DSSYK chord diagrams. Motivated by the AGT correspondence [2]

and its higher rank extension [9], one can ask whether $A_{N-1}$ Toda theory on a Möbius strip is a natural candidate for the chord counting problems that give rise to the Schur half-index. By diagonalising the exponential term within the Toda field theory action, one can show that performing perturbation theory in the cosmological constant yields a chord counting problem with $N-1$ colors of chords, where the bulk intersection weights are given (in oscillator language) by

$$[a_i, a_j^\dagger]_{q^{C_{ij}}} = \delta_{ij} \,, \tag{6.1}$$

with $C_{ij}$ the entries of the $A_{N-1}$ Cartan matrix. Computing the partition functions of multi-colored chord counting constructions are open problems in the literature. Since the Cartan matrix is non-diagonal one needs to keep track of the ordering in which the chords appear, rendering the problem very difficult to solve. Nevertheless, one can compute such moments for low values of $k$. To the best of our knowledge, we cannot reconcile the Toda chord counting problem with the index chord counting problem. Moreover, there is no mechanism within the standard Toda Lagrangian that would give rise to the kind of color changing vertex rules we see on the boundary for the index counting problem. It would be very interesting to explore other proposals that may connect these two points of view.

**Holographic Interpretation.** In section 5.2 we showed that in the $\mathfrak{q} \to 1$ limit the relativistic Toda chain reduces to the classical Toda chain.[17] For the $SU(2)$ case this limit gives rise to Liouville quantum mechanics (LQM)

$$H = k^2 + e^{-\tilde{l}} \,. \tag{6.2}$$

The relevance of LQM to the SYK model was first discussed in [82], and its holographic interpretation in JT gravity was explained in [22, 83]. The phase space of pure JT gravity is two-dimensional: it can be described using the (renormalised) distance between the two asymptotic boundaries of $\mathrm{AdS}_2$, $\tilde{l}$ and its canonical conjugate momentum $k$ and their dynamics is governed by the Hamiltonian (6.2).

There are two notable checks of this correspondence. First, the JT gravity partition function [23, 84–86] on the disk topology matches a survival amplitude in LQM:

$$
\begin{aligned}
Z_{\text{JT gravity}}(\beta) &= \langle 0 | e^{-\beta H} | 0 \rangle_{\text{LQM}} \\
&= \int_0^\infty dE \, \rho(E) e^{-\beta E} = \frac{e^{\pi^2/\beta}}{4\sqrt{\pi}\beta^{3/2}} \,,
\end{aligned}
\tag{6.3}
$$

---

[17]For the precise double scaling limit, see (5.48).

where the state $|0\rangle$ is the zero chord state localized at $\tilde{l} = -\infty$,[18] and where the density of states is

$$\rho(E) = \frac{\sinh(2\pi\sqrt{E})}{4\pi^2}, \tag{6.4}$$

which can be obtained both from a gravity computation, but also by taking the $\mathfrak{q} \to 1$ limit of density of states in (5.26).

Second, an even more detailed check is the matching between the Hartle-Hawking wave function of the gravity side [83, 86] and the Euclidean evolution of the zero chord state:

$$\begin{aligned}
\Psi_{\text{JT gravity}}(\beta/2, \tilde{l}) &= \langle \tilde{l}|e^{-\beta H/2}|0\rangle_{\text{LQM}} \\
&= \int_0^\infty dE\, \rho(E)\, e^{-\beta E/2}\, 2^{3/2} K_{i2\sqrt{E}}\left(2e^{-\tilde{l}/2}\right),
\end{aligned} \tag{6.5}$$

where $K_\nu(z)$ is the modified Bessel function of the second kind. In fact the first match in (6.4) is a consequence of the second, as $Z_{\text{JT}}(\beta) = \int d\tilde{l}\, |\Psi_{\text{JT}}(\beta/2, \tilde{l})|^2$.

JT gravity can be recast in the first order formalism as $SL(2, \mathbb{R})$ BF theory [87–92]. This rewriting suggests a generalization to $SL(N, \mathbb{R})$ BF theory as a higher spin gravity [93]. In the $SL(2, \mathbb{R})$ BF theory the formulas for the partition function and the wave function acquire group theoretical meanings [89, 90]: $\rho(E)dE = d\mu(s)$ with $E = s^2$ is the Plancherel measure with $s$ labeling the principal series representations, the Liouville Hamiltonian is the Casimir, $\tilde{\ell}$ is the hyperbolic Gauss-Euler coordinate on $SL(2, \mathbb{R})$, and the Bessel function is a matrix element of the principal series representation.[19] It would be very interesting to perform the analogous $SL(N, \mathbb{R})$ BF theory computations. We expect that the partition function would match $\langle 0|e^{-\beta H_{\text{Toda}}}|0\rangle$ and the wave function, which now is labelled by multiple wormhole "lengths" appropriate for higher spin gravity, would match $\langle \tilde{l}_i|e^{-\beta/2 H_{\text{Toda}}}|0\rangle$ in the classical Toda chain. The corresponding formulas can be extracted from section 5.

## Acknowledgements

We thank Andrea Antinucci, Pieter Bomans, Justin Kulp, Andy Neitzke, Nikita Nekrasov, Gabriel Wong, and Jingxiang Wu for discussions. OL, MM, SSN are supported in part by STFC grant ST/X000761/1. The work of SSN is also supported in part by the EPSRC Open Fellowship (Schafer-Nameki) EP/X01276X/1. SSN thanks the KITP for hospitality during the course of this work. This research was supported in part by grant NSF PHY-2309135 to the Kavli Institute for Theoretical Physics (KITP).

---

[18]Due to the infinite shift between the $l$ and $\tilde{l}$ variables.

[19]The holographic boundary conditions fix the dependence on the other Gauss-Euler coordinates.

# A  Generalities on Supersymmetric Indices

In this appendix we review the basic ingredients to compute the Schur full and half-indices of 4d $\mathcal{N} = 2$ theories, possibly with the insertion of 3d $\mathcal{N} = 2$ domain walls.

We start by considering the supersymmetric index of 4d $\mathcal{N} = 1$ theories [65, 94, 95] (see [96, 97] for reviews). This can be defined as a refined Witten index of the theory quantized on $S^3 \times \mathbb{R}$

$$\mathcal{I}_{\mathcal{N}=1} = \mathrm{Tr}\, (-1)^F \mathfrak{p}^{j_1 + j_2 + \frac{R_0}{2}} \mathfrak{q}^{j_2 - j_1 + \frac{R_0}{2}} \prod_i f_i^{T_i} \,. \tag{A.1}$$

In this expression, $F$ is the fermion number, $j_1$ and $j_2$ are the Cartan generators of the isometry group $\mathfrak{so}(4) \cong \mathfrak{su}(2)_1 \oplus \mathfrak{su}(2)_2$ of $S^3$, $R_0$ is the Cartan generator of the $U(1)_{R_0}$ R-symmetry, and $T_i$ are the generators of global symmetries which commute with one of the supercharges, which is usually taken to be $\tilde{Q}_{\dot{-}}$. Hence, all the charges that appear in the index (A.1) as the exponents of the fugacities $\mathfrak{p}$, $\mathfrak{q}$, $f_i$ commute with the supercharges $\tilde{Q}_{\dot{-}}$. The trace is taken over states that obey the shortening condition of being annihilated by $\tilde{Q}_{\dot{-}}$, which is equivalent to the condition

$$\tilde{\delta}_{\dot{-}} = E - 2j_2 - \frac{3}{2} R_0 = 0 \,, \tag{A.2}$$

provided that $U(1)_{R_0}$ is taken to be the superconformal R-symmetry.

We can then consider 4d $\mathcal{N} = 2$ SCFTs, which posses an $SU(2)_R \times U(1)_r$ R-symmetry. These can be thought of as $\mathcal{N} = 1$ theories with a $U(1)_{\mathfrak{t}}$ global symmetry that is the commutant of the $\mathcal{N} = 1$ R-symmetry inside the $\mathcal{N} = 2$ R-symmetry. One usually defines (we follow the conventions of [16])

$$\mathcal{I}_{\mathcal{N}=2} = \mathrm{Tr}\, (-1)^F \mathfrak{p}^{j_1 + j_2 - r} \mathfrak{q}^{j_2 - j_1 - r} \mathfrak{t}^{R+r} \prod_i f_i^{T_i} \,, \tag{A.3}$$

where now the trace is taken over states annihilated by $\tilde{Q}_{1\dot{-}}$, which thus obey the condition

$$\tilde{\delta}_{1\dot{-}} = E - 2j_2 - 2R + r = 0 \,. \tag{A.4}$$

One can consider limits of the superconformal index of 4d $\mathcal{N} = 2$ SCFTs that preserve additional supersymmetries, that is that count states annihilated by more than one supercharge [16]. For example, the Schur index is obtained by taking the limit

$$\text{Schur limit:} \quad \mathfrak{t} \to \mathfrak{q} \,. \tag{A.5}$$

Indeed, it turns out that in this limit all the charges that appear in the index commute not only with $\tilde{Q}_{1\dot{-}}$, but also with $Q_{1+}$ and thus obey the additional shortening condition

$$\delta_{1+} = E + 2j_1 - 2R - r = 0 \,, \tag{A.6}$$

which combined with (A.4) gives

$$E = -j_1 + j_2 + 2R, \qquad r = j_1 + j_2.$$

(A.7)

Plugging (A.5)-(A.7) inside (A.3) we find that the Schur index is given by

$$\mathcal{I} = \text{Tr}\,(-1)^F \mathfrak{q}^{j_2-j_1+R},$$

(A.8)

where we recall that the trace is taken over all the states annihilated by $Q_{1+}$ and $\tilde{Q}_{1\dot{-}}$, which by the state/operator map correspond to the so called Schur operators. Observe that the Schur index does not depend on $r$, hence one can actually define the Schur index also for non-conformal theories that lack a $U(1)_r$ symmetry. An example is the 4d $\mathcal{N} = 2$ SYM, for which $U(1)_r$ is anomalous.

For Lagrangian gauge theories, these indices can be computed by enumerating all the possible operators constructed from the fundamental fields and that satisfy the shortening conditions, and selecting only those that are gauge invariant. In the case of the Schur index, the contribution of $\mathcal{N} = 2$ half-hypermultiplets and vector multiplets is given in terms of the following single letter indices:

$$i_{\frac{1}{2}H}(\mathfrak{q}) = \frac{\mathfrak{q}^{\frac{1}{2}}}{1-\mathfrak{q}}, \qquad i_V(\mathfrak{q}) = -\frac{2\mathfrak{q}}{1-\mathfrak{q}}.$$

(A.9)

The index contributions of all possible operators that can be constructed from these fields is obtained by taking the plethystic exponential of the single letter indices, which is defined as

$$\text{PE}\,[f(x)] = \exp\left[\sum_{k=1}^{\infty} \frac{1}{k} f(x^k)\right].$$

(A.10)

Finally, to select only gauge invariant operators one has to integrate over all gauge fugacities with the appropriate Vandermonde determinant measure $\Delta(\vec{z})$. For a gauge theory with gauge algebra $\mathfrak{g}$ and hypermultiplets in representations $\mathcal{R}_n$ this looks as follows:

$$\mathcal{I} = \oint_{\mathbb{T}^{\text{rk}\,\mathfrak{g}}} \prod_{a=1}^{\text{rk}\,\mathfrak{g}} \frac{dz_a}{2\pi i z_a} \Delta(\vec{z}) \text{PE}\left[i_V(\mathfrak{q})\chi_{\text{adj}}(\vec{z}) + \sum_n \left(i_{\frac{1}{2}H}(\mathfrak{q})f_n\chi_{\mathcal{R}_n}(\vec{z}) + i_{\frac{1}{2}H}(\mathfrak{q})f_n^{-1}\chi_{\overline{\mathcal{R}}_n}(\vec{z})\right)\right],$$

(A.11)

where we used the fact that a hyper is made of two half-hypers in conjugate representations under the gauge group and rotated by a flavor symmetry of fugacity $f_n$. This integral can be recast in terms of $\mathfrak{q}$-Pochhammer symbols $(x;\mathfrak{q})_\infty = \prod_{k=0}^{\infty}(1-x\mathfrak{q}^k)$ using that

$$\text{PE}\left[\frac{x}{1-\mathfrak{q}}\right] = \frac{1}{(x;\mathfrak{q})_\infty}.$$

(A.12)

The index (A.11) then becomes

$$\mathcal{I} = \frac{(\mathfrak{q};\mathfrak{q})_\infty^{\mathrm{rk}\,\mathfrak{g}}}{|W_\mathfrak{g}|} \oint_{\mathbb{T}^{\mathrm{rk}\,\mathfrak{g}}} \prod_{a=1}^{\mathrm{rk}\,\mathfrak{g}} \frac{\mathrm{d}z_a}{2\pi i z_a} \frac{\prod_\alpha (\vec{z}^\alpha;\mathfrak{q})_\infty (\mathfrak{q}\vec{z}^\alpha;\mathfrak{q})_\infty}{\prod_n \prod_{\rho_n} \left(\mathfrak{q}^{\frac{1}{2}} f_n \vec{z}^{\rho_n};\mathfrak{q}\right)_\infty \left(\mathfrak{q}^{\frac{1}{2}} f_n^{-1} \vec{z}^{-\rho_n};\mathfrak{q}\right)_\infty} \, , \qquad (A.13)$$

where $|W_\mathfrak{g}|$ is the order of the Weyl group of $\mathfrak{g}$, $\alpha$ are the roots of $\mathfrak{g}$ and $\rho_n$ are the weights of the representation $\mathcal{R}_n$. We also used the short-hand notation $\vec{z}^\alpha = \prod_{a=1}^{\mathrm{rk}\,\mathfrak{g}} z_a^{\alpha_a}$.

Thanks to the additional supersymmetry preserved by the Schur index, one can decorate it with the insertion of $\frac{1}{2}$-BPS line operators [18]. One can either insert a full or a half line. In the case of a half Wilson line in a representation $\mathcal{R}$ one simply inserts in the integrand of (A.13) the corresponding character

$$\mathcal{I} = \frac{(\mathfrak{q};\mathfrak{q})_\infty^{\mathrm{rk}\,\mathfrak{g}}}{|W_\mathfrak{g}|} \oint_{\mathbb{T}^{\mathrm{rk}\,\mathfrak{g}}} \prod_{a=1}^{\mathrm{rk}\,\mathfrak{g}} \frac{\mathrm{d}z_a}{2\pi i z_a} \frac{\prod_\alpha (\vec{z}^\alpha;\mathfrak{q})_\infty (\mathfrak{q}\vec{z}^\alpha;\mathfrak{q})_\infty}{\prod_n \prod_{\rho_n} \left(\mathfrak{q}^{\frac{1}{2}} f_n \vec{z}^{\rho_n};\mathfrak{q}\right)_\infty \left(\mathfrak{q}^{\frac{1}{2}} f_n^{-1} \vec{z}^{-\rho_n};\mathfrak{q}\right)_\infty} \chi_\mathcal{R}(\vec{z}) \, , \qquad (A.14)$$

while a full Wilson line amounts to adding two half Wilson lines in conjugate representations

$$\mathcal{I} = \frac{(\mathfrak{q};\mathfrak{q})_\infty^{2\mathrm{rk}\,\mathfrak{g}}}{|W_\mathfrak{g}|} \oint_{\mathbb{T}^{\mathrm{rk}\,\mathfrak{g}}} \prod_{a=1}^{\mathrm{rk}\,\mathfrak{g}} \frac{\mathrm{d}z_a}{2\pi i z_a} \frac{\prod_\alpha (\vec{z}^\alpha;\mathfrak{q})_\infty (\mathfrak{q}\vec{z}^\alpha;\mathfrak{q})_\infty}{\prod_n \prod_{\rho_n} \left(\mathfrak{q}^{\frac{1}{2}} f_n \vec{z}^{\rho_n};\mathfrak{q}\right)_\infty \left(\mathfrak{q}^{\frac{1}{2}} f_n^{-1} \vec{z}^{-\rho_n};\mathfrak{q}\right)_\infty} \chi_\mathcal{R}(\vec{z}) \chi_{\overline{\mathcal{R}}}(\vec{z}) \, . \qquad (A.15)$$

In the main text we are particularly concerned with the Schur half-index [18]. This is defined as the index of the 4d $\mathcal{N} = 2$ theory in the presence of a 3d boundary that preserves half of the supersymmetries

$$\mathbb{I} = \mathrm{Tr}_\partial (-1)^F \mathfrak{q}^{j_2 - j_1 + R} \, . \qquad (A.16)$$

Due to the presence of the boundary, one has to give boundary conditions to the 4d bulk fields. In order to describe these, it is useful to decompose the $\mathcal{N} = 2$ multiplets into $\mathcal{N} = 1$ ones. In particular, the $\mathcal{N} = 2$ vector decomposes into an $\mathcal{N} = 1$ vector and an $\mathcal{N} = 1$ chiral in the adjoint representation of the gauge group, while the $\mathcal{N} = 2$ hyper decomposes into two $\mathcal{N} = 1$ chirals in conjugate representations. In the present context, the boundary conditions for the $\mathcal{N} = 2$ vector that we are interested in assign Neumann boundary conditions to the $\mathcal{N} = 1$ vector and Dirichlet to the $\mathcal{N} = 1$ adjoint chiral. For the $\mathcal{N} = 2$ hyper instead, one usually assigns Neumann to one of the two $\mathcal{N} = 1$ chirals and Dirichlet to the other. Summarizing

$$\begin{aligned}
\mathcal{N} = 2 \text{ vector} \quad &\to \quad \begin{cases} \mathcal{N} = 1 \text{ vector: Neumann} \\ \mathcal{N} = 1 \text{ adjoint chiral: Dirichlet} \end{cases} \\
\mathcal{N} = 2 \text{ hyper in rep. } \mathcal{R} \quad &\to \quad \begin{cases} \mathcal{N} = 1 \text{ chiral in rep. } \mathcal{R}\text{: Neumann} \\ \mathcal{N} = 1 \text{ chiral in rep. } \overline{\mathcal{R}}\text{: Dirichlet} \end{cases}
\end{aligned} \qquad (A.17)$$

Notice that this in particular implies that Wilson lines are dynamical on the boundary. Only the fields that are given Neumann boundary conditions will contribute to the half index, so

that its integrand is roughly the square root of that of (A.13). Decorating the half index with half Wilson lines (full lines are not allowed for the half index), one has the integral expression

$$\mathcal{I} = \frac{(\mathfrak{q};\mathfrak{q})_\infty^{\mathrm{rk}\,\mathfrak{g}}}{|W_\mathfrak{g}|} \oint_{\mathbb{T}^{\mathrm{rk}\,\mathfrak{g}}} \prod_{a=1}^{\mathrm{rk}\,\mathfrak{g}} \frac{\mathrm{d}z_a}{2\pi i z_a} \frac{\prod_\alpha (\vec{z}^\alpha;\mathfrak{q})_\infty}{\prod_n \prod_{\rho_n} \left(\mathfrak{q}^{\frac{1}{2}} v_n \vec{z}^{\rho_n};\mathfrak{q}\right)_\infty} \chi_\mathcal{R}(\vec{z}) . \tag{A.18}$$

Specializing this expression to the 4d $\mathcal{N} = 2$ $SU(N)$ SYM with insertion of various half Wilson lines one finds the integral (2.4), where in particular we parametrized the $SU(N)$ fugacities using $U(N)$ fugacities $z_1, \cdots, z_N$ subject to the tracelessness condition $\prod_{a=1}^N z_a = 1$.

One can consider more general configurations where a 3d domain wall with intrinsic 3d degrees of freedom is inserted in the 4d theory [3,34,98]. This 3d domain wall does not fully live on the 3d boundary of the half index. Rather, the domain wall intersects the boundary of the half index on a 2d sub-boundary. In particular in subsection 4.4 we focused on the situation in which only 3d $\mathcal{N} = 2$ chiral multiplets lived on this domain wall. To determine their contribution to the 4d Schur half-index, we first start from the one to the 3d index [99–103][20]

$$\mathcal{Z}_{\mathrm{chir}} = \frac{\left(\mathfrak{q}^{1-R_\chi} v^{-1};\mathfrak{q}\right)_\infty}{\left(\mathfrak{q}^{R_\chi} v;\mathfrak{q}\right)_\infty} , \tag{A.19}$$

where $R_\chi$ is the R-charge of the chiral and $v$ is the fugacity for a possible $U(1)$ global symmetry that acts on it. However, this 3d $\mathcal{N} = 2$ chiral living on the domain wall should be assigned either Neumann or Dirichlet boundary conditions on the boundary of the half-index. Depending on this choice we will have that some of the fields of the chiral multiplet will not contribute to the index, resulting in the two possible contributions [78]

$$\mathcal{Z}_{\mathrm{chir}}^{\mathrm{Neu}} = \frac{1}{\left(\mathfrak{q}^{R_\chi} v;\mathfrak{q}\right)_\infty} , \qquad \mathcal{Z}_{\mathrm{chir}}^{\mathrm{Dir}} = \left(\mathfrak{q}^{1-R_\chi} v^{-1};\mathfrak{q}\right)_\infty . \tag{A.20}$$

# B    Properties of Orthogonal Polynomials

In this appendix we review a few relevant properties of orthogonal polynomials [104] in the conventions we used in the main text.

The most general class of such polynomials are the Macdonald polynomials [27]. We will focus on the ones of type $\mathfrak{u}(N)$ first, and then discuss their specialization corrresponding to those of type $\mathfrak{su}(N)$. The Macdonald polynomials are functions of $N$ variables $x_a$ for $a = 1, \cdots, N$ as well as the deformation parameters $\mathfrak{q}$, $\mathfrak{t}$, and they are labelled by ordered partitions of $N$ which we denote by $\vec{\lambda} = (\lambda_1, \lambda_2, \cdots, \lambda_N)$ with $\lambda_1 \geq \lambda_2 \geq \cdots \geq \lambda_N \geq 0$. They are

---

[20] All magnetic fluxes are turned off in this expression.

defined as

$$\mathcal{P}_{\vec{\lambda}}(\vec{x}:\mathfrak{q},\mathfrak{t}) = \mathcal{N}_{\vec{\lambda}}(\mathfrak{q},\mathfrak{t})\left(m_{\vec{\lambda}}(\vec{x}) + \sum_{\vec{\mu}<\vec{\lambda}} u_{\vec{\lambda}\vec{\mu}}(\mathfrak{q},\mathfrak{t})m_{\vec{\mu}}(\vec{x})\right). \tag{B.1}$$

In this expression, the dominance order of partitions is defined as

$$\vec{\mu} \leq \vec{\lambda} \quad \Longleftrightarrow \quad \sum_{a=1}^{k}\mu_a \leq \sum_{a=1}^{k}\lambda_k \text{ for } 1 \leq k \leq N-1 \text{ and } \sum_{a=1}^{N}\mu_a = \sum_{a=1}^{N}\lambda_a \tag{B.2}$$

and $\vec{\mu} < \vec{\lambda}$ if one of the inequalities is strict. Moreover, we defined the monomial symmetric functions as

$$m_{\vec{\lambda}}(\vec{x}) = \sum_{\sigma \in S'_N}\prod_{a=1}^{N} x_a^{\sigma(\lambda_a)} \tag{B.3}$$

with $S'_N$ denoting the distinct permutations of $\vec{\lambda} = (\lambda_1, \lambda_2, \cdots, \lambda_N)$, and compared to [27, 104] we introduced the normalization factor

$$\mathcal{N}_{\vec{\lambda}}(\mathfrak{q},\mathfrak{t}) = z_{\vec{\lambda}}\prod_{a=1}^{N}\frac{1-\mathfrak{q}^{\lambda_a}}{1-\mathfrak{t}^{\lambda_a}}, \qquad z_{\vec{\lambda}} = \prod_{r\geq 1}r^{m_r}m_r!, \tag{B.4}$$

where we re-expressed the partition as $\vec{\lambda} = (N^{l_N}, \cdots, 2^{l_2}, 1^{l_1})$. The coefficients $u_{\vec{\lambda}\vec{\mu}}(\mathfrak{q},\mathfrak{t})$ do not admit a closed form expression, but they can be determined algorithmically (see e.g. [105] for a `Mathematica` package).

Tha Macdonald polynomials are orthogonal

$$\oint_{\mathbb{T}^N}\prod_{a=1}^{N}\frac{\mathrm{d}z_a}{2\pi i z_a}\Delta(\vec{z};\mathfrak{q},\mathfrak{t})\mathcal{P}_{\vec{\lambda}}(\vec{z};\mathfrak{q},\mathfrak{t})\mathcal{P}_{\vec{\nu}}(\vec{z};\mathfrak{q},\mathfrak{t}) = \delta_{\vec{\lambda},\vec{\nu}} \tag{B.5}$$

with respect to the integration measure

$$\Delta(\vec{z};\mathfrak{q},\mathfrak{t}) = \frac{1}{N!}\left(\frac{(\mathfrak{q};\mathfrak{q})_\infty}{(\mathfrak{t};\mathfrak{q})_\infty}\right)^N \prod_{a<b}^{N}\frac{\left((z_a z_b^{-1})^{\pm 1};\mathfrak{q}\right)_\infty}{\left(\mathfrak{t}(z_a z_b^{-1})^{\pm 1};\mathfrak{q}\right)_\infty}. \tag{B.6}$$

They also satisfy various other remarkable properties. One of these is the Cauchy identity

$$\sum_{\vec{\lambda}}\mathcal{P}_{\vec{\lambda}}(\vec{x};\mathfrak{q},\mathfrak{t})\mathcal{P}_{\vec{\lambda}}(\vec{y};\mathfrak{q},\mathfrak{t}) = \prod_{a,b=1}^{N}\frac{(\mathfrak{t}x_a y_b;\mathfrak{q})_\infty}{(x_a y_b;\mathfrak{q})_\infty}. \tag{B.7}$$

Moreover, they are homogeneous in the variables $x_a$ with degree $|\lambda| = \lambda_1 + \cdots + \lambda_N$

$$\mathcal{P}_{\vec{\lambda}}(v x_1, \cdots, v x_N;\mathfrak{q},\mathfrak{t}) = v^{|\lambda|}\mathcal{P}_{\vec{\lambda}}(v x_1, \cdots, v x_N;\mathfrak{q},\mathfrak{t}) \tag{B.8}$$

and possess the shift property

$$\mathcal{P}_{(\lambda_1,\cdots,\lambda_N)}(\vec{x};\mathfrak{q},\mathfrak{t}) = X\,\mathcal{P}_{(\lambda_1-1,\cdots,\lambda_N-1)}(\vec{x};\mathfrak{q},\mathfrak{t}), \qquad X \equiv \prod_{a=1}^{N}x_a. \tag{B.9}$$

One can also label the Macdonald polynomials not with partitions $\vec{\lambda}$ but with a basis corresponding to Dynkin labels of $\mathfrak{u}(N)$

$$\vec{n} = (n_1, \cdots, n_N) = (\lambda_1 - \lambda_2, \lambda_2 - \lambda_3, \cdots, \lambda_{N-1} - \lambda_N, \lambda_N) \,. \tag{B.10}$$

In the main text we used Dynkin labels rather than partitions.

The Macdonald polynomials are eigenfunctions of the Macdonald-Ruijsenaars difference operators. We will not give here the explicit expression of these operators in full generality, but only in the $\mathfrak{t} \to 0$ limit. This is because in this limit the Macdonald polynomials reduce to the $\mathfrak{q}$–Whittaker polynomials that we used in the main text

$$\mathcal{W}_{\vec{\lambda}}(\vec{x}; \mathfrak{q}) = \mathcal{P}_{\vec{\lambda}}(\vec{x}; \mathfrak{q}, \mathfrak{t} = 0) \,. \tag{B.11}$$

In this limit the Macdonald-Ruijsenaars difference operators coincide with the Hamiltonians of the relativistic open Toda chain [26], which for type $\mathfrak{u}(N)$ can be written as (see e.g. [29,30])

$$\mathsf{H}_r = \sum_{I_r} X_{i_1}^{1-\delta_{i_1,1}} X_{i_2}^{1-\delta_{i_2-i_1,1}} \cdots X_{i_r}^{1-\delta_{i_r-i_{r-1},1}} D_{i_1} \cdots D_{i_r} \,, \quad r = 1, \cdots, N \,, \tag{B.12}$$

where the sum is over ordered subsets $I_r = \{i_1 < i_2, \cdots, i_r\}$ of $1, 2, \cdots, N$ and we defined

$$X_a = 1 - \mathfrak{q}^{p_{a+1}-p_a+1} \,, \quad D_a f_{p_1,\cdots,p_N} = f_{p_1,\cdots,p_{a-1},p_a+1,p_{a+1},\cdots,p_N} \,, \tag{B.13}$$

with the action of $D_a$ being expressed in the Dynkin label basis (B.10).

For our purposes, we need the analogous properties (B.5)-(B.7)-(B.8)-(B.9) but for the $\mathfrak{q}$-Whittaker polynomials. These are easily obtained by just considering the $\mathfrak{t} \to 0$ limit. We find that the $\mathfrak{q}$-Whittaker polynomials are orthogonal

$$\oint_{\mathbb{T}^N} \prod_{a=1}^N \frac{\mathrm{d}z_a}{2\pi i z_a} \Delta(\vec{z}; \mathfrak{q}) \mathcal{W}_{\vec{\lambda}}(\vec{z}; \mathfrak{q}) \mathcal{W}_{\vec{\nu}}(\vec{z}; \mathfrak{q}) = \delta_{\vec{\lambda},\vec{\nu}} \,, \tag{B.14}$$

with respect to the integration measure

$$\Delta(\vec{z}; \mathfrak{q}) = \frac{(\mathfrak{q}; \mathfrak{q})_\infty^N}{N!} \prod_{a<b}^N \left( \left(z_a z_b^{-1}\right)^{\pm 1}; \mathfrak{q} \right)_\infty \,, \tag{B.15}$$

they satisfy the Cauchy identity

$$\sum_{\vec{\lambda}} \mathcal{W}_{\vec{\lambda}}(\vec{x}; \mathfrak{q}) \mathcal{W}_{\vec{\lambda}}(\vec{y}; \mathfrak{q}) = \prod_{a,b=1}^N \frac{1}{(x_a y_b; \mathfrak{q})_\infty} \,, \tag{B.16}$$

they are homogeneous

$$\mathcal{W}_{\vec{\lambda}}(v\,x_1, \cdots, v\,x_N; \mathfrak{q}) = v^{|\lambda|} \mathcal{W}_{\vec{\lambda}}(x_1, \cdots, x_N; \mathfrak{q}) \,, \tag{B.17}$$

and they possess the shift property

$$\mathcal{W}_{(\lambda_1,\cdots,\lambda_N)}(\vec{x};\mathfrak{q}) = X\,\mathcal{W}_{(\lambda_1-1,\cdots,\lambda_N-1)}(\vec{x};\mathfrak{q})\,, \qquad X \equiv \prod_{a=1}^{N} x_a\,. \tag{B.18}$$

As mentioned at the beginning of this appendix, so far we have focused on the orthogonal polynomials of type $\mathfrak{u}(N)$. However, in the main text we mainly used those of type $\mathfrak{su}(N)$. The latter can be obtained from the former by just setting

$$\mathfrak{su}(N)\ \text{condition:}\quad X = \prod_{a=1}^{N} x_a = 1\,, \quad \lambda_N = 0\,. \tag{B.19}$$

In the following and in the main text we denote by $\mathcal{W}_{\vec{n}}(\vec{x};\mathfrak{q})$ the $\mathfrak{q}$-Whittaker polynomials of type $\mathfrak{su}(N)$ and in the Dynkin label basis $\vec{n} = (n_1,\cdots,n_{N-1}) \in \mathbb{N}_0^{N-1}$.

We would like to conclude this appendix by providing a proof of the relation (5.33) for the $\mathfrak{q}$-Whittaker polynomials of type $\mathfrak{su}(N)$

$$\sum_{\vec{m}\in\mathbb{N}_0^{N-1}} v^{\sum_{j=1}^{N-1} j m_j}\mathcal{W}_{\vec{m}}(\vec{x};\mathfrak{q})\mathcal{W}_{\vec{m}}(\vec{y};\mathfrak{q})\Bigg|_{\prod_a x_a=\prod_a y_a=1} = \frac{\left(v^N;\mathfrak{q}\right)_\infty}{\prod_{a,b=1}^N \left(v\,x_a y_b;\mathfrak{q}\right)_\infty}\Bigg|_{\prod_a x_a=\prod_a y_a=1}\,, \tag{B.20}$$

where (5.33) is obtained by taking $v = \mathfrak{q}^\Delta$. For this, we start from the Cauchy identity (B.16) for type $\mathfrak{u}(N)$ $\mathfrak{q}$-Whittaker and rewrite the l.h.s using (B.18) for $\lambda_N$ times

$$\sum_{\lambda_N\geq 0} \mathcal{W}_{(\lambda_N)}(X;\mathfrak{q})\mathcal{W}_{(\lambda_N)}(Y;\mathfrak{q})$$

$$\times \sum_{\substack{\lambda_1,\cdots,\lambda_{N-1}\\ \lambda_1\geq\cdots\lambda_{N-1}\geq\lambda_N}} \mathcal{W}_{(\lambda_1-\lambda_N,\cdots,\lambda_{N-1}-\lambda_N,0)}(\vec{x};\mathfrak{q})\mathcal{W}_{(\lambda_1-\lambda_N,\cdots,\lambda_{N-1}-\lambda_N,0)}(\vec{y};\mathfrak{q}) = \prod_{a,b=1}^N \frac{1}{(x_a y_b;\mathfrak{q})_\infty}\,, \tag{B.21}$$

where we defined $X = \prod_{a=1}^N x_a$ and $Y = \prod_{a=1}^N y_a$, and we used the fact that the $\mathfrak{u}(1)$ $\mathfrak{q}$-Whittaker polynomials are simply $\mathcal{W}_{(\lambda_N)}(X;\mathfrak{q}) = X^{\lambda_N}$. Next, we perform the redefinition in the sum on the second line

$$\lambda_i \to \lambda_i + \lambda_N\,, \qquad i = 1,\cdots,N-1\,, \tag{B.22}$$

so to make it independent from the first sum over $\lambda_N$

$$\sum_{\lambda_N\geq 0} \mathcal{W}_{(\lambda_N)}(X;\mathfrak{q})\mathcal{W}_{(\lambda_N)}(Y;\mathfrak{q})$$

$$\times \sum_{\substack{\lambda_1,\cdots,\lambda_{N-1}\\ \lambda_1\geq\cdots\lambda_{N-1}\geq 0}} \mathcal{W}_{(\lambda_1,\cdots,\lambda_{N-1},0)}(\vec{x};\mathfrak{q})\mathcal{W}_{(\lambda_1,\cdots,\lambda_{N-1},0)}(\vec{y};\mathfrak{q}) = \prod_{a,b=1}^N \frac{1}{(x_a y_b;\mathfrak{q})_\infty}\,. \tag{B.23}$$

This allows us to evaluate the sum over $\lambda_N$ of the $\mathfrak{u}(1)$ $\mathfrak{q}$-Whittaker polynomials using the Cauchy identity for $N = 1$

$$\sum_{\substack{\lambda_1,\cdots,\lambda_{N-1}\\ \lambda_1\geq\cdots\lambda_{N-1}\geq 0}} \mathcal{W}_{(\lambda_1,\cdots,\lambda_{N-1},0)}(\vec{x};\mathfrak{q})\mathcal{W}_{(\lambda_1,\cdots,\lambda_{N-1},0)}(\vec{y};\mathfrak{q}) = \frac{(X\,Y;\mathfrak{q})_\infty}{\prod_{a,b=1}^N (x_a y_b;\mathfrak{q})_\infty}\,. \tag{B.24}$$

We now perform the rescaling

$$x_a \to v\,x_a\,, \qquad a = 1, \cdots, N\,, \tag{B.25}$$

which after using the homogeneity property (B.17) gives us

$$\sum_{\substack{\lambda_1,\cdots,\lambda_{N-1} \\ \lambda_1 \geq \cdots \lambda_{N-1} \geq 0}} v^{|\lambda|} \mathcal{W}_{(\lambda_1,\cdots,\lambda_{N-1},0)}(\vec{x};\mathfrak{q}) \mathcal{W}_{(\lambda_1,\cdots,\lambda_{N-1},0)}(\vec{y};\mathfrak{q}) = \frac{\left(v^N X\,Y;\mathfrak{q}\right)_\infty}{\prod_{a,b=1}^N \left(v\,x_a y_b;\mathfrak{q}\right)_\infty}\,. \tag{B.26}$$

The desired result (B.20) is obtained by using the map (B.10) from partitions to Dynkin labels and imposing the $\mathfrak{su}(N)$ conditions $X = Y = 1$ on the fugacities.

# C   Deriving $\mathcal{A}_{\text{Schur}}$ Identities

## C.1   Constructing Commutation Relations for $U_{i,\pm}$

In this appendix we explain the strategy to derive the relations (2.27)-(2.28) for the product of pairs of the $U_{i,\pm}$ in the $SU(3)$ case. We focus in particular on the first equation in (2.27) for the case $i = 1$ and $j = 3$. All the other relations are derived similarly.

By using the definitions (2.25) we first note that

$$U_{1,+}U_{3,+} = \frac{\mathfrak{q}}{\left(1 - V_2 V_1^{-1}\right)\left(1 - V_3 V_1^{-1}\right)} U_1 \frac{\mathfrak{q}}{\left(1 - V_1 V_3^{-1}\right)\left(1 - V_2 V_3^{-1}\right)} U_3$$

$$= \frac{\mathfrak{q}^2}{\left(1 - V_2 V_1^{-1}\right)\left(1 - V_3 V_1^{-1}\right)\left(1 - \mathfrak{q} V_1 V_3^{-1}\right)\left(1 - V_2 V_3^{-1}\right)} U_1 U_3\,, \tag{C.1}$$

where we commuted $U_1$ across the rational function of the $V_i$ using the $\mathfrak{q}$-Weyl algebra (2.24). Similarly

$$U_{3,+}U_{1,+} = \frac{\mathfrak{q}}{\left(1 - V_1 V_3^{-1}\right)\left(1 - V_2 V_3^{-1}\right)} U_3 \frac{\mathfrak{q}}{\left(1 - V_2 V_1^{-1}\right)\left(1 - V_3 V_1^{-1}\right)} U_1$$

$$= \frac{\mathfrak{q}^2}{\left(1 - V_1 V_3^{-1}\right)\left(1 - V_2 V_3^{-1}\right)\left(1 - V_2 V_1^{-1}\right)\left(1 - \mathfrak{q} V_3 V_1^{-1}\right)} U_3 U_1\,. \tag{C.2}$$

By comparing these two expressions and using that $U_1$ and $U_3$ commute, we get the desired result

$$U_{1,+}U_{3,+} = \frac{\left(1 - \mathfrak{q} V_3 V_1^{-1}\right)\left(1 - V_1 V_3^{-1}\right)}{\left(1 - V_3 V_1^{-1}\right)\left(1 - \mathfrak{q} V_1 V_3^{-1}\right)} U_{3,+}U_{1,+} = \frac{V_1 - \mathfrak{q} V_3}{\mathfrak{q} V_1 - V_3} U_{3,+}U_{1,+}\,. \tag{C.3}$$

The other results can be derived analogously.

## C.2   Other Dyonic Lines

In section 2.2 we explained how to express some of the dyonic lines of the $SU(N)$ SYM, which we denoted $h_{n,m}$ in (2.20), in terms of the $\mathfrak{q}$-Weyl algebra variables $V_i$, $U_{i,\pm}$. In this section we

introduce other possible dyonic lines that one could consider and show that they are actually not independent from the $h_{n,m}$. For $SU(3)$ there are no other dyonic lines, so the $h_{n,m}$ are enough to cover all of them.

As we did in section 2.2 for the $h_{n,m}$, we start from the $U(N)$ dyonic lines with minimal magnetic fluxes

$$\mathcal{F}_+ = (0, \cdots, 0, 1), \qquad \mathcal{F}_- = (-1, 0, \cdots, 0), \tag{C.4}$$

These break the gauge group as $U(N) \to U(1) \times U(N-1)$, so we can then dress the line operators with characters either for the $U(1)$ or the $U(N)$ part. In section 2.2 we only considered dressing by the $U(1)$ part, which led us to define the $U(N)$ dyonic lines $H_{\pm 1,n}$ in (2.18). However, we can also consider the $U(N)$ dyonic lines with dressing for the $U(N-1)$ part

$$\tilde{H}_{\pm 1,n} = \sum_{i=1}^{N} \mathfrak{q}^{\pm \frac{n}{2}} \left( \sum_{j \neq i}^{N} V_j^{\mathrm{sgn}(n)} \right)^{|n|} U_{i,\pm}. \tag{C.5}$$

Note that for positive $n$ we dress by powers of the fundamental representation of the residual $U(N-1)$, while for negative $n$ we have the antifundamental representation. In both cases this gives an electric charge of $q_e = n$ to the dyonic line.

From the $U(N)$ operators we can then build $SU(N)$ dyonic lines with the minimal $SU(N)$ magnetic flux

$$\mathcal{F} = (-1, 0, \cdots, 0, 1). \tag{C.6}$$

Some are the $h_{n,m}$ we defined in (2.20), but here we want to consider also

$$\tilde{h}_{n,m} = \tilde{H}_{1,n} \tilde{H}_{-1,m}. \tag{C.7}$$

Similarly to the $h_{n,m}$, these also have charge $q_e = n + m$ under the electric 1-form symmetry.

We will now show how to re-express the dyonic lines $\tilde{h}_{n,m}$ in terms of the dyonic lines $h_{n,m}$ and the Wilson lines $W_{\mathcal{R}_r}$. We will give this proof for generic $SU(N)$, and use its conclusion to argue that we have indeed constructed all dyonic line operators for $SU(3)$. First, let us prove a small result concerning commutators of the $U(N)$ dyonic lines with the fundamental Wilson line that will be useful later

$$\left[ H_{\pm 1,n}, W_{[1,0,\ldots,0]} \right] = \pm (\mathfrak{q}^{\frac{1}{2}} - \mathfrak{q}^{-\frac{1}{2}}) H_{\pm 1,n+1}. \tag{C.8}$$

To prove this, we use the definition of the lines, expand it, and simplify using the $\mathfrak{q}$-Weyl

algebra (2.24)

$$[H_{\pm 1,n}, W_{[1,0,\ldots,0]}] = \sum_{i=1}^{N}\sum_{j=1}^{N} \mathfrak{q}^{\pm\frac{n}{2}} V_i^n U_{i,\pm} V_j - \sum_{i=1}^{N}\sum_{j=1}^{N} \mathfrak{q}^{\pm\frac{n}{2}} V_j V_i^n U_{i,\pm}$$

$$= \sum_{i=1}^{N} \mathfrak{q}^{\pm\frac{n+2}{2}} V_i^{n+1} U_{i,\pm} + \sum_{i\neq j=1}^{N} \mathfrak{q}^{\pm\frac{n}{2}} V_i^n V_j U_{i,\pm} - \sum_{i=1}^{N} \mathfrak{q}^{\pm\frac{n}{2}} V_i^{n+1} U_{i,\pm} - \sum_{i\neq j=1}^{N} \mathfrak{q}^{\pm\frac{n}{2}} V_i^n V_j U_{i,\pm}$$

$$= \mathfrak{q}^{\pm\frac{1}{2}}\sum_{i=1}^{N} \mathfrak{q}^{\pm\frac{n+1}{2}} V_i^{n+1} U_{i,\pm} - \mathfrak{q}^{\mp\frac{1}{2}}\sum_{i=1}^{N} \mathfrak{q}^{\pm\frac{n+1}{2}} V_i^{n+1} U_{i,\pm}$$

$$= \pm(\mathfrak{q}^{\frac{1}{2}} - \mathfrak{q}^{-\frac{1}{2}})H_{\pm 1,n+1}\,. \tag{C.9}$$

A similar proof for the commutator with the antifundamental Wilson line can be constructed, yielding the result

$$[H_{\pm 1,n}, W_{[0,\ldots,0,1]}] = \mp(\mathfrak{q}^{\frac{1}{2}} - \mathfrak{q}^{-\frac{1}{2}})H_{\pm 1,n-1}\,. \tag{C.10}$$

Now we are ready to construct the general proof that we can explicitly express the dyonic lines $\tilde{h}_{n,m}$ in terms of the lines $h_{n,m}$ and Wilson lines $W_{\mathcal{R}_r}$. Let us consider first the case $n, m \geq 0$ where we get the simple expression

$$\tilde{H}_{\pm 1,n} = \sum_{i=1}^{N} \mathfrak{q}^{\pm\frac{n}{2}} \left(\sum_{j\neq i}^{N} V_j\right)^n U_{i,\pm}\,. \tag{C.11}$$

We start by using the identity

$$\sum_{j\neq i}^{N} V_j = W_{[1,0,\ldots,0]} - V_i\,. \tag{C.12}$$

From here, it is simple to rewrite $\tilde{H}_{\pm 1,n}$ in terms of $H_{\pm 1,n}$ and the fundamental Wilson line as

$$\tilde{H}_{\pm 1,n} = \sum_{i=1}^{N} \mathfrak{q}^{\pm\frac{n}{2}} (W_{[1,0,\ldots,0]} - V_i)^n U_{i,\pm}$$

$$= \sum_{i=1}^{N} \binom{n}{k} \mathfrak{q}^{\pm\frac{n}{2}} \left(\sum_{k=0}^{n} W_{[1,0,\ldots,0]}^k (-1)^{n-k} V_i^{n-k}\right) U_{i,\pm}$$

$$= \sum_{k=0}^{n} \binom{n}{k} \mathfrak{q}^{\pm\frac{k}{2}} W_{[1,0,\ldots,0]}^k (-1)^{n-k} \left(\sum_{i=1}^{N} \mathfrak{q}^{\pm\frac{n-k}{2}} V_i^{n-k} U_{i,\pm}\right)$$

$$= \sum_{k=0}^{n} \binom{n}{k} \mathfrak{q}^{\pm\frac{k}{2}} W_{[1,0,\ldots,0]}^k (-1)^{n-k} H_{\pm 1,n-k}\,. \tag{C.13}$$

We can now use these expressions to write down the full $\tilde{h}_{n,m}$ as

$$\tilde{h}_{n,m} = \sum_{k=0}^{n}\sum_{l=0}^{m} \binom{n}{k}\binom{m}{l} \mathfrak{q}^{\frac{k}{2}} W_{[1,0,\ldots,0]}^k (-1)^{n-k} H_{1,n-k} \mathfrak{q}^{-\frac{l}{2}} W_{[1,0,\ldots,0]}^l (-1)^{m-l} H_{-1,m-l}$$

$$= \sum_{k=0}^{n}\sum_{l=0}^{m} \binom{n}{k}\binom{m}{l} (-1)^{n+m-k-l} \mathfrak{q}^{\frac{k-l}{2}} W_{[1,0,\ldots,0]}^k H_{1,n-k} W_{[1,0,\ldots,0]}^l H_{-1,m-l}\,. \tag{C.14}$$

Since we want this expression in terms of $h_{n,m}$ and not simply $H_{\pm 1,n}$, we need to commute the factors $W_{[1,0]}^l$ to the left. In order to do this, we make use of two key facts: the first being (C.8) that we proved earlier, and the second being the following expression

$$AB^l = \sum_{p=0}^{l} \binom{l}{p} B^{l-p} \mathrm{ad}_B^p(A) \,, \tag{C.15}$$

where we define the nested commutator

$$\mathrm{ad}_B^p(A) = [[...[[A,B],B],...,]B] \,, \tag{C.16}$$

with $p$ commutators. Combining these two facts we see

$$H_{1,n-k} W_{[1,0]}^l = \sum_{p=0}^{l} \binom{l}{p} W_{[1,0]}^{l-p} (\mathfrak{q}^{\frac{1}{2}} - \mathfrak{q}^{-\frac{1}{2}})^p H_{1,n-k+p} \,. \tag{C.17}$$

Inserting this expression into $\tilde{h}_{n,m}$ and using the definition of $h_{n,m}$ in terms of $H_{\pm 1,n}$, we arrive at our final expression

$$\tilde{h}_{n,m} = \sum_{k=0}^{n} \sum_{l=0}^{n} \sum_{p=0}^{p} \binom{n}{k}\binom{m}{l}\binom{l}{p} (-1)^{n+m-k-l} \mathfrak{q}^{\frac{k-l}{2}} (\mathfrak{q}^{\frac{1}{2}} - \mathfrak{q}^{-\frac{1}{2}})^p W_{[1,0]}^{k+l-p} h_{n-k+p,m-l} \,. \tag{C.18}$$

One can similarly deal with the case of the dyonic lines $\tilde{h}_{-n,-m}$ with $n,m \geq 0$. Notice indeed that

$$\sum_{j \neq i}^{N} V_j^{-1} = W_{[0,...,0,1]} - V_i^{-1} \,, \tag{C.19}$$

meaning we can write

$$\begin{aligned}
\tilde{H}_{\pm 1,-n} &= \sum_{i=1}^{N} \mathfrak{q}^{\pm \frac{(-n)}{2}} (W_{[0,...,0,1]} - V_i^{-1})^n U_{i,\pm} \\
&= \sum_{i=1}^{N} \sum_{k=0}^{n} \binom{n}{k} \mathfrak{q}^{\pm \frac{(-n)}{2}} W_{[0,...,0,1]}^k (-1)^{n-k} (V_i^{-1})^{n-k} U_{i,\pm} \\
&= \sum_{k=0}^{n} \binom{n}{k} (-1)^{n-k} \mathfrak{q}^{\mp \frac{k}{2}} W_{[0,...,0,1]}^k H_{\pm 1,k-n} \,.
\end{aligned} \tag{C.20}$$

Now we make use of the commutator (C.10) and the expression (C.15) to write

$$\begin{aligned}
\tilde{h}_{-n,-m} &= \sum_{k=0}^{n} \sum_{l=0}^{m} \binom{n}{k}\binom{m}{l} (-1)^{n+m-k-l} \mathfrak{q}^{\frac{-k+l}{2}} W_{[0,...,0,1]}^k H_{1,k-n} W_{[0,...,0,1]}^l H_{-1,l-m} \\
&= \sum_{k=0}^{n} \sum_{l=0}^{m} \sum_{p=0}^{l} \binom{n}{k}\binom{m}{l}\binom{l}{p} (-1)^{n+m-k-l-p} \mathfrak{q}^{\frac{-k+l}{2}} (\mathfrak{q}^{\frac{1}{2}} - \mathfrak{q}^{-\frac{1}{2}})^p W_{[0,...,0,1]}^{k+l-p} H_{1,k-n-p} H_{-1,l-m} \\
&= \sum_{k=0}^{n} \sum_{l=0}^{m} \sum_{p=0}^{l} \binom{n}{k}\binom{m}{l}\binom{l}{p} (-1)^{n+m-k-l-p} \mathfrak{q}^{\frac{-k+l}{2}} (\mathfrak{q}^{\frac{1}{2}} - \mathfrak{q}^{-\frac{1}{2}})^p W_{[0,...,0,1]}^{k+l-p} h_{k-n-p,l-m} \,.
\end{aligned} \tag{C.21}$$

Hence we have expressed the dyonic lines $\tilde{h}_{-n,-m}$ in terms of dyonic lines $h_{n,m}$ and the anti-fundamental Wilson line $W_{[0,...,0,1]}$. For dyonic lines with mixed positive and negative indices $\tilde{h}_{n,-m}$ and $\tilde{h}_{-n,m}$, one can use the above formulas to derive similar closed form expressions for them in terms of the lines $h_{n,m}$ and the Wilson lines $W_{[1,0,...,0]}$ and $W_{[0,...,0,1]}$. This shows that indeed the $\tilde{h}_{n,m}$ are not independent from the $h_{n,m}$.

We should note that one could also consider $SU(N)$ dyonic lines obtained by multiplying $U(N)$ lines of different type, namely $H_{1,n}\tilde{H}_{-1,m}$ and $\tilde{H}_{1,n}H_{-1,m}$. However, thanks to (C.13)-(C.20), one can still re-express these in terms of the $h_{n,m}$ and the Wilson lines only. For $SU(3)$, the dyonic lines considered in this appendix are all the ones of minimal magnetic flux that one can construct. Hence, for $SU(3)$ it is enough to focus on the $h_{n,m}$ as we did in the main text.

## C.3   Proof of the Recurrence Relations for $h_{n,m}$

Here we prove the first of the recurrence relations (2.30). The proofs for the second equation in (2.30) and for the relations in (2.31) work similarly. Before we dive straight into the proof, it is pertinent to collect some small facts that we will use later.

**Fact 1:**   The Wilson line $W_{[n+m-2,0]}$ is written in terms of the Weyl character formula, which can be expressed in the following way:

$$\chi_{[m+n-2,0]}(x,y,z) = \det\begin{pmatrix} x^{m+n} & y^{m+n} & z^{m+n} \\ x & y & z \\ 1 & 1 & 1 \end{pmatrix} \det\begin{pmatrix} x^2 & y^2 & z^2 \\ x & y & z \\ 1 & 1 & 1 \end{pmatrix}^{-1} , \qquad \text{(C.22)}$$

where we recognise the second determinant as the familiar Vandermonde determinant

$$\det\begin{pmatrix} x^2 & y^2 & z^2 \\ x & y & z \\ 1 & 1 & 1 \end{pmatrix} = (y-x)(z-x)(z-y) . \qquad \text{(C.23)}$$

**Fact 2:**   The following equation holds true provided that $\prod_{i=1}^3 V_i = 1$:

$$V_i^n V_j^m V_k = V_i^{n-1} V_j^{m-1} , \qquad i \neq j \neq k , \qquad \text{(C.24)}$$

upon insertion of the $SU(3)$ identity $V_1 V_2 V_3 = 1$. This can be easily proven by exhaustion. For example, if $k = 3$ we have

$$V_i^n V_j^m V_3 = V_i^n V_j^m V_i^{-1} V_j^{-1} = V_i^{n-1} V_j^{m-1} , \qquad \text{(C.25)}$$

as required. Another example would be $i = 3$, $j = 2$, $k = 1$

$$V_3^n V_2^m V_1 = V_1^{-n} V_2^{-n} V_2^m V_1 = V_1^{1-n} V_2^{m-n} = V_3^{n-1} V_2^{m-1} . \qquad \text{(C.26)}$$

One can show that (C.24) holds for all possible choices of $i, j, k = 1, 2, 3$ with $i \neq j \neq k$ in a similar way.

We are now ready to prove the following statement:

$$h_{n,m} W_{[1,0]} - \mathfrak{q}^{\frac{1}{2}} h_{n+1,m} - \mathfrak{q}^{-\frac{1}{2}} h_{n,m+1} - h_{n-1,m-1} = -\mathfrak{q}^{\frac{n+m}{2}} W_{[n+m-2,0]} . \tag{C.27}$$

To do so, we will start with the easier task of matching the terms that have non-trivial $U_{i,\pm}$ dependence on either side. Since the Wilson line is a function of $V_i$ only, there are no terms with a non-trivial $U_{i,\pm}$ dependence on the right hand side. On the left hand side, upon substituting the definition (2.20) of $h_{m,n}$, the non-zero $U_{i\pm}$ dependent terms are

$$\mathfrak{q}^{\frac{n-m}{2}} \left( \sum_{i \neq j}^{3} V_i^n U_{i,+} V_j^m U_{j,-} \right) (V_1 + V_2 + V_3) - \mathfrak{q}^{\frac{n-m+2}{2}} \sum_{i \neq j}^{3} V_i^{n+1} U_{i,+} V_j^m U_{j,-}$$
$$- \mathfrak{q}^{\frac{n-m-2}{2}} \sum_{i \neq j}^{3} V_i^n U_{i,+} V_j^{m+1} U_{j,-} - \mathfrak{q}^{\frac{n-m}{2}} \sum_{i \neq j}^{3} V_i^{n-1} U_{i,+} V_j^{m-1} U_{j,-} . \tag{C.28}$$

Note that we sum over $i \neq j$, since terms with $i = j$ will fuse into functions of the $V_i$'s according to (2.28). Since the products $U_{i,+} U_{j,-}$ with $i \neq j$ are independent, their $V_i$ dependent coefficients must vanish for each of them. Let us consider a generic term of the form

$$(\mathfrak{q}^{\frac{n-m}{2}} V_i^n U_{i,+} V_j^m U_{j,-})(V_i + V_j + V_k) - \mathfrak{q}^{\frac{n-m+2}{2}} V_i^{n+1} U_{i,+} V_j^m U_{j,-}$$
$$- \mathfrak{q}^{\frac{n-m-2}{2}} V_i^n U_{i,+} V_j^{m+1} U_{j,-} - \mathfrak{q}^{\frac{n-m}{2}} V_i^{n-1} U_{i,+} V_j^{m-n} U_{j,-} , \tag{C.29}$$

where $i \neq j \neq k$. Of course under this condition $(V_i + V_j + V_k) = (V_1 + V_2 + V_3)$. Applying the commutation rules for the $\mathfrak{q}$-Weyl algebra, this reduces to

$$\mathfrak{q}^{\frac{n-m}{2}} (V_i^n V_j^m V_k - V_i^{n-1} V_j^{m-1}) U_{i,+} U_{j,-} . \tag{C.30}$$

Now we apply **Fact 2** and conclude the coefficient of $U_{i,+} U_{j,-}$ is zero. Thus we have shown that all $U_{i,\pm}$ dependent terms vanish in (C.27).

Next, we will show that the $U_{i,\pm}$ independent terms on the left hand side can be rearranged into the form of the Weyl character (C.22) with $x = V_1$, $y = V_2$, and $z = V_3$. To start, we first need to rearrange all terms into the form $f_i(V_1, V_2, V_3) U_{i,+} U_{i,-}$. Doing so results in

$$\mathfrak{q}^{\frac{n-m}{2}} (\mathfrak{q}^m V_1^{n+m} (V_2 + V_3) - \mathfrak{q}^{m+1} V_1^{m+n+1} - \mathfrak{q}^{m-1} V_1^{m+n-2}) U_{1,+} U_{1,-}$$
$$+ \mathfrak{q}^{\frac{n-m}{2}} (\mathfrak{q}^m V_2^{n+m} (V_1 + V_3) - \mathfrak{q}^{m+1} V_2^{m+n+1} - \mathfrak{q}^{m-1} V_2^{m+n-2}) U_{2,+} U_{2,-} \tag{C.31}$$
$$+ \mathfrak{q}^{\frac{n-m}{2}} (\mathfrak{q}^m V_3^{n+m} (V_1 + V_2) - \mathfrak{q}^{m+1} V_3^{m+n+1} - \mathfrak{q}^{m-1} V_3^{m+n-2}) U_{3,+} U_{3,-} .$$

Next, we apply (2.28) so to get

$$
\mathfrak{q}^{\frac{n+m}{2}}(\mathfrak{q}V_1^{n+m+1}+\mathfrak{q}^{-1}V_1^{n+m-2}-V_1^{n+m}(V_2+V_3))\frac{\mathfrak{q}V_1^2V_2V_3}{(V_1-V_2)(V_1-V_3)(\mathfrak{q}V_1-V_2)(\mathfrak{q}V_1-V_3)}
$$
$$
+\mathfrak{q}^{\frac{n+m}{2}}(\mathfrak{q}V_2^{n+m+1}+\mathfrak{q}^{-1}V_2^{n+m-2}-V_2^{n+m}(V_1+V_3))\frac{\mathfrak{q}V_2^2V_1V_3}{(V_2-V_1)(V_2-V_3)(\mathfrak{q}V_2-V_1)(\mathfrak{q}V_2-V_3)}
$$
$$
+\mathfrak{q}^{\frac{n+m}{2}}(\mathfrak{q}V_3^{n+m+1}+\mathfrak{q}^{-1}V_3^{n+m-2}-V_3^{n+m}(V_1+V_2))\frac{\mathfrak{q}V_3^2V_1V_2}{(V_3-V_1)(V_3-V_1)(\mathfrak{q}V_3-V_1)(\mathfrak{q}V_3-V_2)}\,.
$$
$$\tag{C.32}$$

From here, we multiply and divide the first term by $(V_3 - V_2)$, the second by $(V_1 - V_3)$, and the last by $(V_2 - V_1)$. This allows us to factorise out the common denominator

$$
\frac{1}{(V_2-V_1)(V_3-V_1)(V_3-V_2)}\,,
\tag{C.33}
$$

which we recognise as the inverse of the Vandermonde determinant that appears in the Weyl character formula of **Fact 1**. The next task is to recover the form of the numerator of the Weyl character formula. Indeed, we can verify that after stripping the expression of the Vandermonde determinant, we are left with

$$
\mathfrak{q}^{\frac{n+m}{2}}(\mathfrak{q}V_1^{n+m+1}+\mathfrak{q}^{-1}V_1^{n+m-2}-V_1^{n+m}(V_2+V_3))\frac{\mathfrak{q}V_1^2V_2V_3(V_3-V_2)}{(\mathfrak{q}V_1-V_2)(\mathfrak{q}V_1-V_3)}
$$
$$
+\mathfrak{q}^{\frac{n+m}{2}}(\mathfrak{q}V_2^{n+m+1}+\mathfrak{q}^{-1}V_2^{n+m-2}-V_2^{n+m}(V_1+V_3))\frac{\mathfrak{q}V_2^2V_1V_3(V_1-V_3)}{(\mathfrak{q}V_2-V_1)(\mathfrak{q}V_2-V_3)}
\tag{C.34}
$$
$$
+\mathfrak{q}^{\frac{n+m}{2}}(\mathfrak{q}V_3^{n+m+1}+\mathfrak{q}^{-1}V_3^{n+m-2}-V_3^{n+m}(V_1+V_2))\frac{\mathfrak{q}V_3^2V_1V_2(V_2-V_1)}{(\mathfrak{q}V_3-V_1)(\mathfrak{q}V_3-V_2)}\,,
$$

which upon inserting $V_3 = (V_1 V_2)^{-1}$, simplifies to

$$
\mathfrak{q}^{\frac{n+m}{2}}(V_1^{n+m}-V_1^{-n-m+2}V_2^{-n-m+1}+V_1^{-n-m+1}V_2^{-n-m+2}-V_1^{n+m+1}V_2^2-V_2^{n+m}+V_1^2V_2^{n+m})\,.\tag{C.35}
$$

Lastly, we can rewrite this expression as

$$
-\mathfrak{q}^{\frac{n+m}{2}}\det\begin{pmatrix}V_1^{m+n} & V_2^{m+n} & V_1^{-n-m}V_2^{-n-m}\\ V_1 & V_2 & V_3\\ 1 & 1 & 1\end{pmatrix}\,,
\tag{C.36}
$$

allowing us to reconstruct the full Weyl character (C.22). This completes the proof of (C.27).

## C.4 Proof of the Dyonic Line Algebraic Relations

In this appendix we provide the proof of the first relation in (2.36). The proofs of the other relations in (2.36)-(2.37) work similarly.

Using the explicit form of $h_{n,m}$ in terms of the $\mathfrak{q}$-Weyl algebra, we have

$$
h_{n,0}=\mathfrak{q}^{\frac{n}{2}}\sum_{i,j}V_i^nU_{i+}U_{j-}=\mathfrak{q}^{\frac{n}{2}}V_1^n\sum_jU_{1+}U_{j-}+\mathfrak{q}^{\frac{n}{2}}V_2^n\sum_jU_{2+}U_{j-}+\mathfrak{q}^{\frac{n}{2}}V_3^n\sum_jU_{3+}U_{j-}\,,\tag{C.37}
$$

and similarly

$$h_{n,1} = \mathfrak{q}^{\frac{n-1}{2}} \sum_{i,j} V_i^n U_{i+} V_j U_{j-} = \mathfrak{q}^{\frac{n-1}{2}} V_1^n \sum_j U_{1+} V_j U_{j-} + \mathfrak{q}^{\frac{n-1}{2}} V_2^n \sum_j U_{2+} V_j U_{j-}$$
$$+ \mathfrak{q}^{\frac{n-1}{2}} V_3^n \sum_j U_{3+} V_j U_{j-} \,. \tag{C.38}$$

Our goal is to now verify the following three claims:

$$\mathfrak{q}^{\frac{n+2}{2}} V_1^n \sum_j U_{1+} U_{j-} h_{0,1} = \mathfrak{q}^{\frac{n-1}{2}} V_1^n \sum_j U_{1+} V_j U_{j-} h_{0,0} \,,$$
$$\mathfrak{q}^{\frac{n+2}{2}} V_2^n \sum_j U_{2+} U_{j-} h_{0,1} = \mathfrak{q}^{\frac{n-1}{2}} V_2^n \sum_j U_{2+} V_j U_{j-} h_{0,0} \,, \tag{C.39}$$
$$\mathfrak{q}^{\frac{n+2}{2}} V_3^n \sum_j U_{3+} U_{j-} h_{0,1} = \mathfrak{q}^{\frac{n-1}{2}} V_3^n \sum_j U_{3+} V_j U_{j-} h_{0,0} \,.$$

Since $V_i^n$ factors out of the above equations, we see that the above equalities are in fact independent of $n$. Thus we only have to prove that (C.39) holds true for $n = 0$. To prove these statements, one should first expand $h_{0,1}$ and $h_{0,0}$ using their $\mathfrak{q}$-Weyl definitions (2.20), then proceed to commute all of the $V_i$ to the left using the $\mathfrak{q}$-Weyl algebra (2.26), and order the $U_{i,\pm}$ according to the standard ordering $U_{1,+}^{p_1} U_{1,-}^{q_1} U_{2,+}^{p_2} U_{2,-}^{q_2} U_{3,+}^{p_3} U_{3,-}^{q_3}$, where the $p_i$, $q_i$ are some positive integer powers. Fusing the $U_{i,+} U_{i,-}$ using (2.28) and re-ordering the expression such that the rational functions of the $V_i$'s always appear on the left of the expressions, one can then verify that the two sides of the equations agree. Indeed, one can perform this substantial exercise to verify these equalities are indeed true for $n = 0$, and so we have proven the claim.

## C.5 Fusion of Bulk and Boundary Lines

In this section we provide more details for the derivation of the relations in (2.44) and (2.48) encoding the fusion of certain bulk and boundary lines. In particular, we will focus on the action of the dyonic lines $h_{1,0}$ and $h_{0,1}$ on the trivial boundary Wilson line $|1^\partial\rangle$, which is the vacuum of the Hilbert space $\mathcal{H}_{\text{Schur}}$. The equations for the action of $h_{-1,0}$ and $h_{0,-1}$ are proven analogously.

The first identity that we would like to prove is

$$h_{1,0}|1^\partial\rangle = \mathfrak{q}^{\frac{3}{2}} |W_{[1,0]}^\partial\rangle = \mathfrak{q}^{\frac{3}{2}} W_{[1,0]}^\partial |1^\partial\rangle \,. \tag{C.40}$$

For this, we first write out the definition of $h_{1,0}$ in terms of the variables $V_i$ and $U_{i,\pm}$

$$h_{1,0} = H_{1,1} H_{-1,0} = \mathfrak{q}^{\frac{1}{2}} \left( \sum_{i=1}^3 V_i U_{i,+} \right) \left( \sum_{j=1}^3 U_{j,-} \right) \,, \tag{C.41}$$

which when written in terms of the $\mathfrak{q}$-Weyl variables $V_i$, $U_i$ using (2.12), becomes

$$h_{1,0} = \mathfrak{q}^{\frac{3}{2}} \sum_{i,j=1}^{3} V_i \frac{1}{\prod_{k\neq i}(1 - V_k V_i^{-1})} U_i \frac{1}{\prod_{l\neq j}(V_j V_l^{-1} - 1)} U_j^{-1} \,. \tag{C.42}$$

The goal is now to commute the $U_i$ to the right hand side of this expression, where we can then safely apply the action of boundary fusion. Employing the $\mathfrak{q}$-Weyl algebra, we see that

$$h_{1,0} = \mathfrak{q}^{\frac{3}{2}} \sum_{i,j=1}^{3} V_i \frac{1}{\prod_{k\neq i}(1 - V_k V_i^{-1})} \frac{1}{\prod_{l\neq j}(\mathfrak{q}^{\delta_{ij}-\delta_{il}} V_j V_l^{-1} - 1)} U_i U_j^{-1} \,. \tag{C.43}$$

Acting on the vacuum then gives

$$h_{1,0}|1^\partial\rangle = \mathfrak{q}^{\frac{3}{2}} \sum_{i,j=1}^{3} V_i \frac{1}{\prod_{k\neq i}(1 - V_k V_i^{-1})} \frac{1}{\prod_{l\neq j}(\mathfrak{q}^{\delta_{ij}-\delta_{il}} V_j V_l^{-1} - 1)} |1^\partial\rangle \,, \tag{C.44}$$

where we used the property that fusing to the boundary is implemented on the $\mathfrak{q}$-Weyl variables $U_i$ by $U_i \to 1$. It is now simply a task of verifying the above rational function of $\vec{V}$ simplifies to the form of the fundamental $SU(3)$ Wilson line

$$\sum_{i,j=1}^{3} V_i \frac{1}{\prod_{k\neq i}(1 - V_k V_i^{-1})} \frac{1}{\prod_{l\neq j}(\mathfrak{q}^{\delta_{ij}-\delta_{il}} V_j V_l^{-1} - 1)} = V_1 + V_2 + V_3 \,. \tag{C.45}$$

Since this is a function of only $V_i$, all the variables in this function are commuting, and so it is just an algebraic exercise to verify that indeed the above equation is true.

Next, we wish to prove the statement

$$h_{0,1}|1^\partial\rangle = 0 \,. \tag{C.46}$$

Proceeding in the same manner, we expand the dyonic line explicitly in terms of the $\mathfrak{q}$-Weyl algebra variables

$$\begin{aligned} h_{0,1} = H_{1,0} H_{-1,1} &= \mathfrak{q}^{-\frac{1}{2}} \left( \sum_{i=1}^{3} U_{i,+} \right) \left( \sum_{j=1}^{3} V_j U_{j,-} \right) \\ &= \mathfrak{q}^{\frac{1}{2}} \sum_{i,j=1}^{3} \frac{1}{\prod_{k\neq i}(1 - V_k V_i^{-1})} U_i V_j \frac{1}{\prod_{l\neq j}(V_j V_l^{-1} - 1)} U_j^{-1} \,, \,. \end{aligned} \tag{C.47}$$

Commuting $U_i$ through the expression and acting on the ground state gives

$$h_{0,1}|1^\partial\rangle = \mathfrak{q}^{\frac{1}{2}} \sum_{i,j=1}^{3} \frac{1}{\prod_{k\neq i}(1 - V_k V_i^{-1})} \mathfrak{q}^{\delta_{ij}} V_j \frac{1}{\prod_{l\neq j}(\mathfrak{q}^{\delta_{ij}-\delta_{il}} V_j V_l^{-1} - 1)} |1^\partial\rangle \,. \tag{C.48}$$

Finally, one can verify the somewhat non-trivial looking identity

$$\sum_{i,j=1}^{3} \frac{1}{\prod_{k\neq i}(1 - V_k V_i^{-1})} \mathfrak{q}^{\delta_{ij}} V_j \frac{1}{\prod_{l\neq j}(\mathfrak{q}^{\delta_{ij}-\delta_{il}} V_j V_l^{-1} - 1)} = 0 \,, \tag{C.49}$$

thus proving the dyonic line $h_{0,1}$ annihilates the ground state of the Schur Hilbert space.

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
