# Peer review of "Schur Connections: Chord Counting, Line Operators, and Indices"

_SciPost Physics_

## Round 1 · Referee Report · Anonymous (Referee 1) · 2025-9-19

The referee discloses that the following generative AI tools have been used in the preparation of this report:
After I prepared a report in a text editor, I let ChatGPT 5 polish the whole report. Then I checked and edited the content.
Strengths
- Makes a number of intriguing connections between different subjects in mathematical physics.
- Derivations are detailed.
- Some claims are proved in general settings.
Weaknesses
- The connections made may be mere coincidences.
- Contains typos and grammatical errors.
Report
At present, it is unclear whether these connections are mere mathematical coincidences or the first indications of a deeper physical structure. Nonetheless, the range of concrete correspondences uncovered by the authors meets two of the journal’s criteria:
• providing a novel and synergetic link between different research areas, and
• opening a new pathway in an existing or a new research direction, with clear potential for multi-pronged follow-up work.
However, the current version falls short of meeting two other essential criteria:
• being written in a clear and intelligible way, free of unnecessary jargon, ambiguities and misrepresentations, and
• providing citations to relevant literature in a way that is as representative and complete as possible.
In addition, the manuscript contains numerous typographical and grammatical errors, which hinder readability. Some specific issues are:
• In (2.5), the character in terms of $z_a$ should be simply $\sum_{a=1}^N z_a$. It seems the authors have confused (2.5) with (3.54), which itself contains a typo (see below).
• In (3.3), $n_{i-1}$ should be $n_i - 1$.
• In (3.54), the factor should be $v_i v_{i-1}^{-1}$, not $v_i v_{i+1}^{-1}$.
• Four lines below (4.29), the phrase “so between points 3 and 4” is unclear and should be clarified.
• Substituting (4.39) into (4.38) produces a minus sign in front of \Delta in (4.40), which is currently missing.
• In (4.40), a comma is required between $\beta_1$ and $\beta_2$.
• In (4.40), there is a $\Delta$ in the exponent of $q$, but no corresponding $\Delta$ in (4.44) or in the expression involving $q$ below (4.44). Consistency needs to be checked.
Requested changes
- Correct the typos and grammatical errors listed.
- Carefully review the manuscript once again to eliminate further typographical errors and imprecise statements not listed.
- At the beginning of the paragraph containing (4.42), it is claimed that (4.41) coincides with a Schur half-index decorated by a domain wall. Some explanation is provided, but without appropriate references or a more detailed derivation, the claim cannot be verified. The physical interpretation of $\beta_1$ and $\beta_2$ in the domain wall setting should also be clarified.
Recommendation
Ask for minor revision

---

## Editorial Decision

unknown